# Selective Sampling and Imitation Learning via Online Regression

**Ayush Sekhari**[1][*]    **Karthik Sridharan**[2]    **Wen Sun**[2]    **Runzhe Wu**[2][†]

[1]MIT, [2]Cornell University

## Abstract

We consider the problem of Imitation Learning (IL) by actively querying noisy expert for feedback. While imitation learning has been empirically successful, much of prior work assumes access to noiseless expert feedback which is not practical in many applications. In fact, when one only has access to noisy expert feedback, algorithms that rely on purely offline data (non-interactive IL) can be shown to need a prohibitively large number of samples to be successful. In contrast, in this work, we provide an interactive algorithm for IL that uses selective sampling to actively query the noisy expert for feedback. Our contributions are twofold: First, we provide a new selective sampling algorithm that works with general function classes and multiple actions, and obtains the best-known bounds for the regret and the number of queries. Next, we extend this analysis to the problem of IL with noisy expert feedback and provide a new IL algorithm that makes limited queries.

Our algorithm for selective sampling leverages function approximation, and relies on an online regression oracle w.r.t. the given model class to predict actions, and to decide whether to query the expert for its label. On the theoretical side, the regret bound of our algorithm is upper bounded by the regret of the online regression oracle, while the query complexity additionally depends on the eluder dimension of the model class. We complement this with a lower bound that demonstrates that our results are tight. We extend our selective sampling algorithm for IL with general function approximation and provide bounds on both the regret and the number of queries made to the noisy expert. A key novelty here is that our regret and query complexity bounds only depend on the number of times the optimal policy (and not the noisy expert, or the learner) go to states that have a small margin.

## 1 Introduction

From the classic supervised learning setting to the more complex problems like interactive Imitation Learning (IL) [Ross et al., 2011], high-quality labels or supervision is often expensive and hard to obtain. Thus, one wishes to develop algorithms that do not require a label for every data sample presented during the learning process. Active learning or selective sampling is a learning paradigm that is designed to reduce query complexity by only querying for labels at selected data points, and has been extensively studied in both theory and practice [Agarwal, 2013, Dekel et al., 2012, Hanneke and Yang, 2021, Zhu and Nowak, 2022, Cesa-Bianchi et al., 2005, Hanneke and Yang, 2015].

In this work, we study selective sampling and its application to interactive Imitation Learning [Ross et al., 2011]. Our goal is to design algorithms that can leverage general function approximation and online regression oracles to achieve small regret on predicting the correct labels, and at the

---

[*]Authors are listed in alphabetical order of their last names.

[†]Emails: `sekhari@mit.edu`, `{ks999, ws455, rw646}@cornell.edu`

37th Conference on Neural Information Processing Systems (NeurIPS 2023).

same time minimize the number of expert queries made (query complexity). Towards this goal, we first study selective sampling which is an online active learning framework, and provide regret and query complexity bounds for general function classes (used to model the experts). Our key results in selective sampling are obtained by developing a connection between the regret of the online regression oracles and the regret of predicting the correct labels. Additionally, we bound the query complexity using the eluder dimension [Russo and Van Roy, 2013] of the underlying function class used to model the expert. We complement our results with a lower bound indicating that a dependence on an eluder dimension like complexity measure is unavoidable in the query complexity in the worst case. In particular, we provide lower bounds in terms of the star number of the function class—a quantity closely related to the eluder dimension. Our new selective sampling algorithm, called SAGE, can operate under fairly general modeling assumptions, loss functions, and allows for multiple labels (i.e., multi-class classification).

We then extend our selective sampling algorithm to the interactive IL framework proposed by Ross et al. [2011] to reduce the query complexity. While the DAgger algorithm proposed by Ross et al. [2011] has been extensively used in various robotics applications (e.g., Ross et al. [2013], Pan et al. [2018]), it often requires a large number of expert queries. There have been some efforts on reducing the expert query complexity by leveraging ideas from active learning (e.g., Laskey et al. [2016], Brantley et al. [2020]), however, these prior attempts do not have theoretical guarantees on bounding expert's query complexity. In this work, we provide the first provably correct algorithm for interactive IL with general function classes, called RAVIOLI, which not only achieves strong regret bounds in terms of maximizing the underlying reward functions, but also enjoys a small query complexity. Furthermore, we note that RAVIOLI operates under significantly weaker assumptions as compared to the prior works, like Ross et al. [2011], on interactive IL. In particular, we only assume access to a noisy expert, as compared to the prior works that assume that the expert is noiseless. In fact, for the noisy setting, we show that one can not even hope to learn from purely offline expert demonstrations unless one has exponentially in horizon $H$ many samples. Such a strong separation does not hold in the noiseless setting.

Our bounds depend on the margin of the noisy expert, which intuitively quantifies the confidence level of the expert. In particular, the margin is large for states where the expert is very confident in terms of providing the correct labels, while on the other hand, the margin is small on the states where the expert is less confident and subsequently provides more noisy labels as feedback. Such kind of margin condition was missing in prior works, like Ross et al. [2011], which assumes that the expert can provide confident labels everywhere. Additionally, we note that our margin assumption is quite mild as we only assume that the expert has a large margin under the states that could be visited by the noiseless expert (however, the states visited by the learner, or by following the noisy expert, may not have a small margin).

We then extend our results to the multiple expert setting where the learner has access to $M$ many experts/teachers who may have different expertise at different parts of the state space. In particular, there is no expert who can singlehandedly perform well on the underlying environment, but an aggregation of their policies can lead to good performance. Such an assumption holds in various applications and has been recently explored in continuous control tasks like robotics and discrete tasks like chess and Minigrid [Beliaev et al., 2022]. Similar to the single expert setting, we model the expertise of the experts in multiple expert setting using the concept of margins. Different experts have different margin functions, capturing the fact that experts may have different expertise at different parts of the state space. Prior work from Cheng et al. [2020] also considers multiple experts in IL and provides meaningful regret bounds, however, their assumption on the experts is much stronger than us: they assume that for any state, there at least exists one expert who can achieve high reward-to-go if the expert took over the control starting from this state till the end of the episode. Furthermore, Cheng et al. [2020] considers the setting where one can also query for the reward signals, whereas we do not require access to any reward signals.

## 2 Contributions and Overview of Results

### 2.1 Selective Sampling

Online selective sampling models the interaction between a learner and an adversary over $T$ rounds. At the beginning of each round of the interaction, the adversary presents a context $x_t$ to the learner. After receiving the context, the learner makes a prediction $\hat{y}_t \in [K]$, where $K$ denotes the number of

actions. Then, the learner needs to make a choice of whether or not to query an *expert* who is assumed to have some knowledge about the true label for all the presented contexts. The experts knowledge about the true label is modeled via the ground truth modeling function $f^\star$, which is assumed to belong to a given function class $\mathcal{F}$ but is unknown to the learner. If the learner decides to query for the label, then the expert will return a noisy label $y_t$ sampled using $f^\star$. If the learner does not query, then the learner does not receive any feedback in this round. The learner makes an update based on the latest information it has, and moves on to the next round of the interaction. The goal of the learner is to compete with the expert policy $\pi^\star$, that is defined using the experts model $f^\star$. In the selective sampling setting, we are concerned with two things: the total regret of the learner w.r.t. the policy $\pi^\star$, and the number of expert queries that the learner makes. Our key contributions are as follows:

- We provide a new selective sampling algorithm (Algorithm 1) that relies on an online regression oracle w.r.t. $\mathcal{F}$ (where $\mathcal{F}$ is the given model class) to make predictions and to decide whether to query for labels. Our algorithm can handle multiple actions, adversarial contexts, arbitrary model class $\mathcal{F}$, and fairly general modeling assumptions (that we discuss in more detail in Section 3), and enjoys the following regret bound and query complexity:

$$\mathrm{Reg}_T = \widetilde{\mathcal{O}}\left(\inf_\varepsilon\left\{\varepsilon T_\varepsilon + \frac{\mathrm{Reg}(\mathcal{F};T)}{\varepsilon}\right\}\right) \quad \text{and} \quad N_T = \widetilde{\mathcal{O}}\left(\inf_\varepsilon\left\{T_\varepsilon + \frac{\mathrm{Reg}(\mathcal{F};T)\cdot\mathfrak{E}(\mathcal{F},\varepsilon;f^\star)}{\varepsilon^2}\right\}\right).$$
(1)

  where $\mathrm{Reg}(\mathcal{F};T)$ denotes the regret bound for the online regression oracle on $\mathcal{F}$, $\mathfrak{E}(\mathcal{F},\varepsilon;f^\star)$ denotes the eluder dimension of $\mathcal{F}$, and $T_\varepsilon$ denotes the number of rounds at which the margin of the experts predictions is smaller than $\varepsilon$ (the exact notion of margin is defined in Section 3).

- We show via a lower bound that, without additional assumptions, the dependence on the eluder dimension in the query complexity bound (1) is unavoidable if we desire a regret bound of the form (1), even when $T_\varepsilon = 0$. The details are located in Section 3.2.

- For the stochastic setting, where the context $\{x_t\}_{t\le T}$ are sampled i.i.d. from a fixed unknown distribution, we provide an alternate algorithm (Algorithm 3) that enjoys the same regret bound as (1) but whose query complexity scales with the disagreement coefficient of $\mathcal{F}$ instead of the eluder dimension (Theorem 2). Since the disagreement coefficient is always smaller than the eluder dimension, Theorem 2 yields an improvement in the query complexity.

## 2.2 Imitation Learning

We then move to the more challenging Imitation Learning (IL) setting, where the learner operates in an episodic finite horizon Markov Decision Process (MDP), and can query a noisy expert for feedback (i.e. the expert action) on the states that it visits. The interaction proceeds in $T$ episodes of length $H$ each. In episode $t$, at each time step $h \in [H]$ and on the state $x_{t,h}$, the learner chooses an action $\widehat{y}_{t,h}$ and transitions to state $x_{t,h+1}$. However, the learner does not receive any reward signal. Instead, the learner can actively choose to query an *expert* who has some knowledge about the correct action to be taken on $x_{t,h}$, and gives back noisy feedback $y_{t,h}$ about this action. Similar to the selective sampling setting, the experts knowledge about the true label is modeled via the ground truth modeling function $f_h^\star$, which is assumed to belong to a given function class $\mathcal{F}_h$ but is unknown to the learner. The goal of the learner is to compete with the optimal policy $\pi^\star$ of the (noiseless) expert. Our key contributions in IL are:

- In Section 4, we first demonstrate an exponential separation in terms of task horizon $H$ in the sample complexity, for learning via offline expert demonstration only vs interactive querying of experts, when the feedback from the expert is noisy.

- We then provide a general IL algorithm (in Algorithm 2) that relies on online regression oracles w.r.t. $\{\mathcal{F}_h\}_{h\le H}$ to predict actions, and to decide whether to query for labels. Similar to the selective sampling setting, the regret bound for our algorithm scales with the regret of the online regression oracles, and the query complexity bound has an additional dependence on the eluder dimension. Furthermore, our algorithm can handle multiple actions, adversarially changing dynamics, arbitrary model class $\mathcal{F}$, and fairly general modeling assumptions.

- A key difference from our results in selective sampling is that the term $T_\varepsilon$ that appears in our regret and query complexity bounds in IL denote the number of time steps in which the expert

policy $\pi^\star$ has a small margin (instead of the number of time steps when the learner's policy has a small margin). In fact, the learner and the expert trajectories could be completely different from each other, and we only pay in the margin term if the expert trajectory at that time step would have a low margin. See Section 4 for the exact definition of margin.

- In Section 4.1, we provide extensions to our algorithm when the learner can query $M$ experts at each round. In particular, we do not assume that any of the experts is singlehandedly optimal for the entire state space, but that there exist aggregation functions of these experts' predictions that perform well in practice, and with which we compete.

## 3   Selective Sampling

In the problem of selective sampling, on every round $t$, nature (or an adversary) produces a context $x_t$. The learner then receives this context and predicts a label $\widehat{y_t} \in [K]$ for that context. The learner also computes a query condition $Z_t \in \{0, 1\}$ for that context. If $Z_t = 1$, the learner requests for label $y_t \in [K]$ corresponding to the $x_t$, and if not, the learner receives no feedback on the label for that round. Let $\mathcal{F}$ be a model class such that each model $f \in \mathcal{F}$ maps contexts $x$ to scores $f(x) \in \mathbb{R}^K$. In this work we assume that while contexts can be chosen arbitrarily, the label $y_t$ corresponding to a context $x_t$ is drawn from a distribution over labels specified by the score $f^\star(x_t)$ where $f^\star \in \mathcal{F}$ is a fixed model unknown to the learner. We assume that a link function $\phi : \mathbb{R}^K \mapsto \Delta(K)$ maps scores to distributions and assume that the noisy label $y_t$ is sampled as

$$y_t \sim \phi(f^\star(x_t)). \tag{2}$$

In this work, we assume that the link function $\phi(v) = \nabla\Phi(v)$ for some $\Phi : \mathbb{R}^K \mapsto \mathbb{R}$ (see Agarwal [2013] for more details) which satisfies the following assumption.

**Assumption 1.** *The function $\Phi$ is $\lambda$-strongly-convex and $\gamma$-smooth, i.e. for all $u, u' \in \mathbb{R}^K$,*

$$\frac{\lambda}{2}\|u' - u\|_2^2 \le \Phi(u') - \Phi(u) - \langle \nabla\Phi(u), u' - u \rangle \le \frac{\gamma}{2}\|u' - u\|_2^2.$$

Our main contribution in this section is a selective sampling algorithm that uses online non-parametric regression w.r.t. the model class $\mathcal{F}$ as a black box. Specifically, define the loss function corresponding to the link function $\phi$ as $\ell_\phi(v, y) = \Phi(v) - v[y]$ where $v \in \mathbb{R}^K$ and $y \in [K]$. We assume that the learner has access to an online regression oracle for the loss $\ell_\phi$ (which is a convex loss) w.r.t. the class $\mathcal{F}$, that for any sequence $\{(x_1, y_1), \ldots, (x_T, y_T)\}$ guarantees the regret bound

$$\sum_{s=1}^{T} \ell_\phi(f_s(x_s), y_s) - \inf_{f \in \mathcal{F}} \sum_{s=1}^{T} \ell_\phi(f(x_s), y_s) \le \mathrm{Reg}^{\ell_\phi}(\mathcal{F}; T). \tag{3}$$

When $\phi$ is identity (under which the models in $\mathcal{F}$ directly map to distributions over the labels), then $\ell_\phi$ denotes the standard square loss, and we need a bound on the regret w.r.t. the square loss, denoted by $\mathrm{Reg}^{\mathrm{sq}}(\mathcal{F}; T)$. When $\phi$ is the Boltzman distribution mapping (given by $\Phi$ being the softmax function) then $\ell_\phi$ is the logistic loss, and we need an online logistic regression oracle for $\mathcal{F}$. Minimax rates for the regret bound in (3) are well known:

- *Square-loss regression:* Rakhlin and Sridharan [2014] characterized the minimax rates for online square loss regression in terms of the offset sequential Rademacher complexity of $\mathcal{F}$, which for example, leads to regret bound $\mathrm{Reg}^{\mathrm{sq}}(\mathcal{F}; T) = O(\log|\mathcal{F}|)$ for finite function classes $\mathcal{F}$, and $\mathrm{Reg}^{\mathrm{sq}}(\mathcal{F}; T) = O(d\log(T))$ when $\mathcal{F}$ is a $d$-dimensional linear class. More examples can be found in Rakhlin and Sridharan [2014, Section 4]. We refer the readers to Krishnamurthy et al. [2017], Foster et al. [2018a] for efficient implementations.

- *Logistic-loss regression:* When $\mathcal{F}$ is finite, we have the regret bound $\mathrm{Reg}(\mathcal{F}; T) \le O(\log|\mathcal{F}|)$ [Cesa-Bianchi and Lugosi, 2006, Chapter 9]. For learning linear predictors, there exists efficient improper learner with regret bound $\mathrm{Reg}(\mathcal{F}; T) \le O(d\log|T|)$ [Foster et al., 2018b]. More examples can be found in Foster et al. [2018b, Section 7] and Rakhlin and Sridharan [2015].

When one deals with complex model classes $\mathcal{F}$ such that the labeling concept class corresponding to $\mathcal{F}$ could possibly have infinite VC dimension (like it is typically the case), then one needs to naturally

rely on a margin-based analysis [Tsybakov, 2004, Shalev-Shwartz and Ben-David, 2014, Dekel et al., 2012]. For $p \in \mathbb{R}^K$, we use the following well-known notion of margin for multiclass settings[3]:

$$\texttt{Margin}(p) = \phi(p)[k^\star] - \max_{k' \neq k^\star} \phi(p)[k'] \qquad \text{where} \quad k^\star \in \operatorname*{argmax}_k \phi(p)[k], \qquad (4)$$

A key quantity that appears in our results is the number of $x_t$'s that fall within an $\varepsilon$ margin region,

$$T_\varepsilon = \sum_{t=1}^{T} \mathbf{1}\{\texttt{Margin}(f^\star(x_t)) \leq \varepsilon\}.$$

$T_\varepsilon$ denotes the number of times where even the Bayes optimal classifier is confused about the correct label on $x_t$, and has confidence less than $\varepsilon$. The algorithm relies on an online regression oracle mentioned above to produce the predictor $f_t$ at every round. The predicted label $\widehat{y}_t = \texttt{SelectAction}(f_t(x_t)) = \operatorname{argmax}_k \phi(f_t(x_t))[k]$ is picked based on the score $f_t(x_t)$ (where $\widehat{y}_t$ is the label with the largest score). The learner updates the regression oracle on only those rounds in which it makes a query. Our main algorithm for selective sampling is provided in Algorithm 1.[4]

---

**Algorithm 1** **S**elective **SA**mplin**G** with **E**xpert Feedback (SAGE)

---

**Input:** Parameters $\delta, \gamma, \lambda, T$, function class $\mathcal{F}$, and online regression oracle Oracle w.r.t $\ell_\phi$.
1: Set $\Psi_\delta^{\ell_\phi}(\mathcal{F}, T) = \frac{4}{\lambda} \operatorname{Reg}^{\ell_\phi}(\mathcal{F}; T) + \frac{112}{\lambda^2} \log(4 \log^2(T)/\delta)$, Compute $f_1 \leftarrow \texttt{Oracle}_1(\varnothing)$.
2: **for** $t = 1$ to $T$ **do**
3:     Nature chooses $x_t$.
4:     Learner plays the action $\widehat{y}_t = \texttt{SelectAction}(f_t(x_t))$.
5:     Learner computes

$$\Delta_t(x_t) := \max_{f \in \mathcal{F}} \|f(x_t) - f_t(x_t)\| \quad \text{s.t.} \quad \sum_{s=1}^{t-1} Z_s \|f(x_s) - f_s(x_s)\|^2 \leq \Psi_\delta^{\ell_\phi}(\mathcal{F}, T). \qquad (5)$$

6:     Learner decides whether to query: $Z_t = \mathbf{1}\{\texttt{Margin}(f_t(x_t)) \leq 2\gamma\Delta_t(x_t)\}$.
7:     **if** $Z_t = 1$ **then**
8:         Learner queries the label $y_t$ on $x_t$.
9:         $f_{t+1} \leftarrow \texttt{Oracle}_t(\{x_t, y_t\})$.
10:     **else**
11:         $f_{t+1} \leftarrow f_t$.

---

Our goal in this work is twofold: Firstly, we would like Algorithm 2 to have a low regret w.r.t. the optimal model $f^\star$, defined as

$$\operatorname{Reg}_T = \sum_{t=1}^{T} \mathbf{1}\{\widehat{y}_t \neq y_t\} - \sum_{t=1}^{T} \mathbf{1}\{\texttt{SelectAction}(f^\star(x_t)) \neq y_t\}$$

Simultaneously, we also aim to make as few label queries $N_T = \sum_{t=1}^{T} Z_t$ as possible. Before delving into our results, we first recall the following variant of eluder-dimension [Russo and Van Roy, 2013, Foster et al., 2020, Zhu and Nowak, 2022].

**Definition 1** (Scale-sensitive eluder dimension (normed version)). *Fix any $f^\star \in \mathcal{F}$, and define $\widetilde{\mathfrak{E}}(\mathcal{F}, \beta; f^\star)$ to be the length of the longest sequence of contexts $x_1, x_2, \ldots x_m$ such that for all $i$, there exists $f_i \in \mathcal{F}$ such that*

$$\|f_i(x_i) - f^\star(x_i)\| > \beta, \quad \text{and} \quad \sum_{j < i} \|f_i(x_j) - f^\star(x_j)\|^2 \leq \beta^2.$$

*The value function eluder dimension is defined as $\mathfrak{E}(\mathcal{F}, \beta'; f^\star) = \sup_{\beta \geq \beta'} \widetilde{\mathfrak{E}}(\mathcal{F}, \beta; f^\star)$.*

Bounds on the eluder dimension for various function classes are well known, e.g. when $\mathcal{F}$ is finite, $\mathfrak{E}(\mathcal{F}, \beta'; f^\star) \leq |\mathcal{F}| - 1$, and when $\mathcal{F}$ is the set of $d$-dimensional function with bounded norm, then $\mathfrak{E}(\mathcal{F}, \beta'; f^\star) = O(d)$. We refer the reader to Russo and Van Roy [2013], Mou et al. [2020], Li et al. [2022] for more examples. The following theorem is our main result for selective sampling:

---

[3]Throughout the paper, we assume that the ties in argmax or argmin are broken arbitrarily, but consistently.
[4]Unless explicitly specified, the action set is given by $\mathcal{A} = [K] = \{1, \ldots, K\}$ where $K \geq 2$.

**Theorem 1.** *Let $\delta \in (0,1)$. Under the modeling assumptions above (in (2), (3) and (4)), with probability at least $1 - \delta$, Algorithm 1 obtains the regret bound*

$$\mathrm{Reg}_T = \widetilde{\mathcal{O}}\left( \inf_\varepsilon \left\{ \varepsilon T_\varepsilon + \frac{\gamma^2}{\lambda \varepsilon} \mathrm{Reg}^{\ell_\phi}(\mathcal{F};T) + \frac{\gamma^2}{\lambda^2 \varepsilon} \log(1/\delta) \right\} \right), \qquad and,$$

$$N_T = \widetilde{\mathcal{O}}\left( \inf_\varepsilon \left\{ T_\varepsilon + \frac{\gamma^2}{\lambda \varepsilon^2} \cdot \mathrm{Reg}^{\ell_\phi}(\mathcal{F};T) \cdot \mathfrak{E}(\mathcal{F}, \varepsilon/4\gamma; f^\star) + \frac{\gamma^2}{\lambda^2 \varepsilon^2} \log(1/\delta) \right\} \right).$$

A few points are in order:

- It must be noted that for most settings we consider, as an example if model class $\mathcal{F}$ is finite, one typically has that $\mathrm{Reg}(\mathcal{F};T) \le \log|\mathcal{F}|$. Thus, in the case where one has a hard margin condition i.e. $T_{\varepsilon_0} = 0$ for some $\varepsilon_0 > 0$, we get $\mathrm{Reg}_T \le O\left( \frac{\log|\mathcal{F}|}{\varepsilon_0} \right)$ and $N_T \le O\left( \frac{\mathfrak{E}(\mathcal{F}, \varepsilon; f^\star) \log|\mathcal{F}|}{\varepsilon_0^2} \right)$.

- Our regret bound does not depend on the eluder dimension. However, the query complexity bound has a dependence on eluder dimension. Thus, for function classes for which the eluder dimension is large, the regret bound is still optimal while the number of label queries may be large.

### 3.1  Selective Sampling in the Stochastic Setting

So far we assumed that the contexts $\{x_t\}_{t \ge 0}$ could be chosen in a possibly adversarial fashion, and thus our bound on the number of label queries scales with the eluder dimension. However, it turns out that if the contexts are drawn i.i.d. from some (unknown) distribution $\mu$, then one can improve the query complexity to scale with the value function disagreement coefficient of $\mathcal{F}$ (defined below) which is always smaller than the eluder dimension (Lemma 6).

**Definition 2** (Scale sensitive disagreement coefficient (normed version), Foster et al. [2020]). *Let $\mathcal{F} \subseteq \{\mathcal{X} \mapsto \mathbb{R}^K\}$. For any $f^\star \in \mathcal{F}$, and $\beta_0, \varepsilon_0 > 0$, the value function disagreement coefficient $\theta^{\mathrm{val}}(\mathcal{F}, \varepsilon_0, \beta_0; f^\star)$ is defined as*

$$\sup_\mu \sup_{\beta > \beta_0, \varepsilon > \varepsilon_0} \left\{ \frac{\varepsilon^2}{\beta^2} \cdot \mathrm{Pr}_{x \sim \mu}(\exists f \in \mathcal{F} \mid \|f(x) - f^\star(x)\| > \varepsilon, \|f - f^\star\|_\mu \le \beta) \right\} \vee 1$$

*where $\|f\|_\mu = \sqrt{\mathbb{E}_{x \sim \mu}[\|f(x)\|^2]}$.*

The key idea that gives us the above improvement, of replacing the eluder dimension by disagreement coefficient in the query complexity bound, is to use epoching for the query condition, while still using an online regression oracle to make predictions. The exact algorithm is given in Appendix E.4.

**Theorem 2.** *Let $\delta \in (0,1)$, and consider the modeling assumptions in (2), (3) and (4). Furthermore, suppose that $x_t$ is sampled i.i.d. from $\mu$, where $\mu$ is a fixed distribution. Then, with probability at least $1 - \delta$, Algorithm 3 obtains the bounds*[5]

$$\mathrm{Reg}_T = \widetilde{\mathcal{O}}\left( \inf_\varepsilon \left\{ \varepsilon T_\varepsilon + \frac{\gamma^2}{\lambda \varepsilon} \mathrm{Reg}^{\ell_\phi}(\mathcal{F};T) \right\} \right), \qquad and,$$

$$N_T = \widetilde{\mathcal{O}}\left( \inf_\varepsilon \left\{ T_\varepsilon + \frac{\gamma^2}{\lambda \varepsilon^2} \cdot \mathrm{Reg}^{\ell_\phi}(\mathcal{F};T) \cdot \theta^{\mathrm{val}}\left( \mathcal{F}, \varepsilon/8\gamma, \mathrm{Reg}^{\ell_\phi}(\mathcal{F};T)/T; f^\star \right) \right\} \right).$$

We note that Algorithm 3 automatically adapts to Tsybakov noise condition with respect to $\mu$.

**Corollary 1** (Tsybakov noise condition, Tsybakov [2004]). *Suppose there exists constants $c, \rho \ge 0$ s.t. $\mathrm{Pr}_{x \sim \mu}(\mathtt{Margin}(f^\star(x)) \le \varepsilon) \le c\varepsilon^\rho$ for all $\varepsilon \in (0,1)$, and consider the same modeling assumptions as in Theorem 2. Then, with probability at least $1 - \delta$, Algorithm 3 obtains the bound*

$$\mathrm{Reg}_T = \widetilde{\mathcal{O}}\left( \left( \mathrm{Reg}^{\ell_\phi}(\mathcal{F};T) \right)^{\frac{\rho+1}{\rho+2}} \cdot (T)^{\frac{1}{\rho+2}} \right), \qquad and,$$

$$N_T = \widetilde{\mathcal{O}}\left( \left( \mathrm{Reg}^{\ell_\phi}(\mathcal{F};T) \cdot \theta^{\mathrm{val}}\left( \mathcal{F}, \varepsilon/8\gamma, \mathrm{Reg}^{\ell_\phi}(\mathcal{F};T)/T; f^\star \right) \right)^{\frac{\rho}{\rho+2}} \cdot T^{\frac{2}{\rho+2}} \right).$$

*where the $\widetilde{\mathcal{O}}(\cdot)$ notation hides poly-logarithmic factors of $\gamma, \lambda, c, \rho$ and $\log(T/\delta)$.*

A detailed comparison of our results with the relevant prior works is given in Appendix E.

---

[5] In the rest of the paper, the notation $\widetilde{\mathcal{O}}$ hides additive $\log(1/\delta)$-factors which, for constant $\delta$ and in all the results, are asymptotically dominated by the other terms presented in the displayed bounds.

## 3.2 Lower Bounds (Binary Action Case)

We supplement the above upper bound with a lower bound in terms of the star number of $\mathcal{F}$ (defined below). The star number is bounded from above by the eluder dimension which appears in our upper bounds (Lemma 6). While star number may not be lower bounded by eluder dimension in general, for many commonly considered classes, star number is of the same order as the eluder dimension [Foster et al., 2020]. For the sake of a clean presentation, we restrict our lower bound to the binary actions case, although one can easily extend the lower bound to the multiple actions case.

**Definition 3** (scale-sensitive star number). *For any $\zeta \in (0,1)$ and $\beta \in (0,\zeta/2)$, define $\mathfrak{s}^{\mathrm{val}}(\mathcal{F},\zeta,\beta)$ as the largest $m$ such that there exists target function $f^\star \in \mathcal{F}$ and sequence $x_1,\ldots,x_m \in \mathcal{X}$ s.t. $\forall i \in [m]$, $|f^\star(x_i)| > \zeta$, $\exists f_i \in \mathcal{F}$ s.t.,*

$$(\mathbf{1}) \sum_{\substack{j \neq i}} (f_i(x_j) - f^\star(x_j))^2 < \beta^2 \quad (\mathbf{2}) |f_i(x_i)| > \zeta/2 \text{ and } f_i(x_i)f^\star(x_i) < 0 \quad (\mathbf{3}) |f_i(x_i) - f^\star(x_i)| \leq 2\zeta$$

The below theorem provides a lower bound on number of queries, in terms of star number for any algorithm that guarantees a non-trivial regret bound.

**Theorem 3.** *Given a function class $\mathcal{F}$ and some desired margin $\zeta > 0$, define $\beta \in (0,\zeta/2)$ be the largest number such that $\beta^2 \leq \min\{\zeta^2/\mathfrak{s}^{\mathrm{val}}(\mathcal{F},\zeta,\beta), \zeta^2/16\}$. Then, for any algorithm that guarantees regret bound of $\mathbb{E}[\mathrm{Reg}_T] \leq 64\frac{\zeta T}{\mathfrak{s}^{\mathrm{val}}(\mathcal{F},\zeta,\beta)}$ on all instances with margin $\zeta/2$, there exists a distribution $\mu$ over $\mathcal{X}$ and a target function $f^\star \in \mathcal{F}$ with margin[6] $\zeta$ such that the number of queries $N_T$ made by the algorithm on that instance in $T$ rounds of interaction satisfy*

$$\mathbb{E}[N_T] = \Omega\left(\frac{\mathfrak{s}^{\mathrm{val}}(\mathcal{F},\zeta,\beta)}{40\zeta^2}\right).$$

The above lower bound demonstrates that for any algorithm that has a sublinear regret guarantee, a dependence on an additional complexity measure like the star number (or the eluder dimension) is unavoidable in the number of queries in the worst case. This suggests that our upper bound cannot be further improved beyond the discrepancy between the star number and eluder dimension. The following corrolary illustrates the above lower bound.

**Corollary 2.** *There exists a class $\mathcal{F}$ with $|\mathcal{F}| = \sqrt{T}$, and $\mathfrak{s}^{\mathrm{val}}(\mathcal{F},\zeta,\beta) = O(\sqrt{T})$ for any $\beta = O(1)$ and $\zeta = O(1)$, such that any algorithm that makes less than $\sqrt{T}$ number of label queries, will have a regret of at least $\mathbb{E}[\mathrm{Reg}_T] \geq \sqrt{T}$ on some instance with margin $\zeta$.*

## 4 Imitation Learning ($H > 1$) with Selective Queries to an Expert

The problem of Imitation Learning (IL) consists of learning policies in MDPs when one has access to an expert (aka the teacher) that can make suggestions on which actions to take at a given state. IL has enjoyed tremendous empirical success, and various different interaction models have been considered. In the simplest IL setting, studied under the umbrella of offline RL [Levine et al., 2020] or Behavior Cloning [Ross and Bagnell, 2010, Torabi et al., 2018], the learner is given an offline dataset of trajectories (state and action pairs) from an expert and aims to output a well-performing policy. Here, the learner is not allowed any interaction with the expert, and can only rely on the provided dataset of expert demonstrations for learning. A much stronger IL setting is the one where the learner can interact with the expert, and rely on its feedback on states that it reaches by executing its own policies.

In their seminal work, Ross et al. [2011] proposed a framework for interactive imitation learning via reduction to online learning and classification tasks. This has been extensively studied in the IL literature (e.g., Ross and Bagnell [2014], Sun et al. [2017], Cheng and Boots [2018]). The algorithm DAgger from [Ross et al., 2011] has enjoyed great empirical success. On the theoretical side, however, performance guarantees for DAgger only hold under the assumption that, when queried, the expert makes action suggestions from a very good policy $\pi^\star$ that we would like to compete with. However, in practice, human demonstrators are far from being optimal and suggestions from experts should be modeled as noisy suggestions that only correlate with $\pi^\star$. It turns out that IL where one only has

---

[6]When $\mathcal{A} = \{1,2\}$, recall that $\mathtt{Margin}(f(x)) = |\mathrm{Pr}(y = 2 \mid f(x)) - \mathrm{Pr}(y = 1 \mid f(x))| = |f(x)|$.

access to noisy expert suggestions is drastically different from the noiseless setting. For instance, in the sequel, we show that there can be an exponential separation in terms of the dependence on horizon $H$ in the sample complexity of learning purely from offline demonstration vs learning with online interactions.

Formally, we consider interactive IL in an episodic finite horizon Markov Decision Process (MDP), where the learner can query a noisy expert for feedback (i.e., action) on the states that it visits. The game proceeds in $T$ episodes. In each episode $t$, the nature picks the initial state $x_{t,1}$ for $h = 1$; then for every time step $h \in [H]$, the learner proposes an action $\hat{y}_{t,h} \in [K]$ given the current state $x_{t,h}$; then the system proceeds by selecting the next state $x_{t;h+1} \leftarrow \mathbb{T}_{t,h}(x_{t,h}, \hat{y}_{t,h})$, where $\mathbb{T}_{t,h} : \mathcal{X} \times \mathcal{Y} \mapsto \mathcal{X}$ denotes the deterministic dynamics at timestep $h$ of round $t$ and is unknown to the learner. The learner then decides whether to query the expert for feedback. If the learner queries, it receives a recommended action from the expert, and otherwise the learner does not receive any additional information. The game moves on to the next time step $h + 1$, and moves to the next episode $t + 1$ when it reaches to time step $H$ in the current episode. We now describe the expert model. With $f_h^\star$ being the underlying score function at time step $h$, the expert feedback is sampled from a distribution $\phi(f_h^\star(x)) \in \Delta(K)$, with $\phi : \mathbb{R}^K \mapsto \mathbb{R}^K$ being some link function (e.g., $\phi(p)[i] \propto \exp(p[i])$). The goal of the leaner is to perform as well as the Bayes optimal policy[7] defined as $\pi_h^\star(x) := \mathrm{argmax}_{a \in [K]} \phi(f_h^\star(x))$. In particular, the learner aims to find a sequence of policies $\{\pi_t\}_{t \leq T}$ that have a small cumulative regret defined w.r.t. some (unknown) reward function under possibly adversarial (and unknown) transition dynamics $\{\mathbb{T}_{t,h}\}_{h \leq H, t \leq T}$. At the same time, the learner wants to minimize the number of queries made to the expert. Formally, we consider counterfactual regret defined as

$$\mathrm{Reg}_T = \sum_{t=1}^{T} \sum_{h=1}^{H} r(x_{t,h}^{\pi^\star}, \pi_h^\star(x_{t,h}^{\pi^\star})) - \sum_{t=1}^{T} \sum_{h=1}^{T} r(x_{t,h}, \hat{y}_{t,h})$$

where $x_{t,h}$ are the states reached by the learner corresponding to the chosen actions and the dynamics, and $x_{t,h}^{\pi^\star}$ denotes the states that would have been generated if we executed $\pi^\star$ from the beginning of the episode under the same dynamics. The query complexity $N_T$ is the total number of queries to the expert across all $H$ steps in $T$ episodes.

Given the selective sampling results we provided in the earlier section, one may be tempted to apply them to the imitation learning problem. However, there is a caveat. A key to the reduction in Ross et al. [2011] is to apply Performance Difference Lemma (PDL) to reduce the problem of IL to online classification under the sequence of state distributions induced by the policies played by the learning algorithm. Hence, if one blindly applied this reduction, then in the margin term, one would need to account for the states that the learner visits (which could be arbitrary). Thus, for DAgger to have meaningful bounds, we would require a large margin over the entire state space. This is too much to ask for in practical applications. Consider the example of learning autonomous driving from a human driver as the expert. It is reasonable to believe that human drivers can confidently provide the right actions when they are driving themselves or are faced with situations they are more familiar with. However, assuming that the human driver is going to be confident in an unfamiliar situation (e.g., an emergency situation that is not often encountered by the human driver), is a strong assumption. Towards that end, we make a significantly weaker, and much more realistic, margin assumption that the expert has a large margin only on the state distribution induced by $\pi^\star$, and not on the state distribution of the learner or the noisy expert.[8] In particular, we define $T_{\varepsilon,h}$ to denote the total number of episodes where the comparator policy $\pi^\star$ visits a state with low margin at time step $h$, i.e., $T_{\varepsilon,h} = \sum_{t=1}^{T} \mathbf{1}\{\mathtt{Margin}(f_h^\star(x_{t,h}^{\pi^\star})) \leq \varepsilon\}$.

We now proceed to our main results in this section. Learning from a noisy expert is indeed very challenging. In fact, learning from noisy expert feedback may even be statistically intractable in the non-interactive IL setting, where the learner is only limited to accessing offline noisy expert demonstrations for learning, e.g. in offline RL, Behavior Cloning, etc. The following lower bound formalizes this. In fact, the same lower bound also shows that AggreVaTe [Ross and Bagnell, 2014] style algorithms would not succeed under noisy expert feedback, AggreVaTe relies on roll-outs obtained by running the (noisy) expert suggestions.

---

[7]Note that the comparator policy $\pi^\star$ reflects the experts models, and may not be the optimal policy for the underlying MDP.

[8]The precise definition of the $\mathtt{Margin}$ for IL is given in the appendix.

**Proposition 1** (Lower bound for learning from non-interactive noisy demonstrations). *There exists an MDP, for every $h \leq H$, a function class $\mathcal{F}_h$ with $|\mathcal{F}_h| \leq 2^H$, a noisy expert whose optimal policy $\pi^\star(x) = \arg\max_a(f_h^\star(x)[a])$ for some $f_h^\star \in \mathcal{F}_h$ with $T_{\varepsilon,h} = 0$ for any $\varepsilon \leq 1/4$, such than any non-interactive algorithm needs $\Omega(2^H)$ many noisy expert trajectory demonstrations to learn, with probability at least $3/4$, a policy $\widehat{\pi}$ that is $1/8$-suboptimal w.r.t. $\pi^\star$.*

Proposition 1 implies that in order to learn with a reasonable sample complexity (that is polynomial in $H$), a learner must be able to interactively query the expert. In Algorithm 2, we provide an interactive imitation learning algorithm (with selective querying) that can learn from noisy expert feedback. The regret bound and query complexity bounds for Algorithm 2 are:

---

**Algorithm 2** Inte**RA**cti**V**e **I**mitati**O**n **L**earning V**I**a Active Expert Querying (RAVIOLI)

---

**Input:** Params $\delta, \gamma, \lambda, T$, function classes $\{\mathcal{F}_h\}_{h \leq H}$, online regression oracle $\mathsf{Oracle}_h$ w.r.t. $\ell_\phi$ for $h \in [H]$.
1: Set $\Psi_\delta^{\ell_\phi}(\mathcal{F}_h, T) = \frac{4}{\lambda}\mathrm{Reg}^{\ell_\phi}(\mathcal{F}_h; T) + \frac{112}{\lambda^2}\log(4H\log^2(T)/\delta)$.
2: Compute $f_{1,h} = \mathsf{Oracle}_{1,h}(\varnothing)$ for $h \in [H]$.
3: **for** $t = 1$ to $T$ **do**
4:     Nature chooses the state $x_{t,1}$.
5:     **for** $h = 1$ to $H$ **do**
6:         Learner plays $\widehat{y}_{t,h} = \mathtt{SelectAction}(f_{t,h}(x_{t,h}))$
7:         Learner transitions to the next state in this round $x_{t,h+1} \leftarrow \mathbb{T}_{t,h}(x_{t,h}, \widehat{y}_{t,h})$.
8:         Learner computes

$$\Delta_{t,h} := \max_{f \in \mathcal{F}_h} \|f(x_{t,h}) - f_{t,h}(x_{t,h})\| \text{ s.t. } \sum_{s=1}^{t-1} Z_{s,h}\|f(x_{s,h}) - f_{s,h}(x_{s,h})\|^2 \leq \Psi_\delta^{\ell_\phi}(\mathcal{F}_h, T). \quad (6)$$

9:         Learner decides whether to query: $Z_{t,h} = \mathbf{1}\{\mathtt{Margin}(f_{t,h}(x_{t,h})) \leq 2\gamma\Delta_{t,h}\}$.
10:        **if** $Z_{t,h} = 1$ **then**
11:            Learner queries the label $y_{t,h}$ for $x_{t,h}$.
12:            $f_{t+1,h} \leftarrow \mathsf{Oracle}_{t+1,h}(\{x_{t,h}, y_{t,h}\})$
13:        **else**
14:            $f_{t+1,h} \leftarrow f_{t,h}$

---

**Theorem 4.** *Let $\delta \in (0,1)$. Under the modeling assumptions above, with probability at least $1 - \delta$, Algorithm 2 obtains:*

$$\mathrm{Reg}_T = \widetilde{\mathcal{O}}\left(\inf_\varepsilon\left\{H\sum_{h=1}^{H} T_{\varepsilon,h} + \frac{H\gamma^2}{\lambda\varepsilon^2}\sum_{h=1}^{H}\mathrm{Reg}^{\ell_\phi}(\mathcal{F}_h; T)\right\}\right), \qquad and,$$

$$N_T = \widetilde{\mathcal{O}}\left(\inf_\varepsilon\left\{H\sum_{h=1}^{H} T_{\varepsilon,h} + \frac{H\gamma^2}{\lambda\varepsilon^2}\sum_{h=1}^{H}\mathrm{Reg}^{\ell_\phi}(\mathcal{F}_h; T) \cdot \mathfrak{E}(\mathcal{F}_h, \varepsilon/8\gamma; f_h^\star)\right\}\right).$$

Since the above bound holds for any sequence of dynamics $\{\mathbb{T}_{h,t}\}_{h \leq H, t \leq T}$, the result of Theorem 4 also holds for the stochastic IL setting where the transition dynamic is stochastic but fixed during the interaction. In particular, setting $\mathbb{T}_{h,t} \sim \mathscr{T}_h$ sampled i.i.d. from a fixed stochastic dynamics $\{\mathscr{T}_h\}_{h \leq H}$ recovers a similar bound for the stochastic setting.

## 4.1 Learning from Multiple Experts

In Dekel et al. [2012], the problem of selective sampling from multiple experts is considered with the main motivation being that we can consider each expert as being confident (and correct) in certain states or scenarios, and we would like to learn from their joint feedback. The goal there is to perform not only as well as the best of them individually but even as well as the best combination of them. Consider the example of learning to drive from human demonstrations, we might have one human demonstrator who is an expert in highway driving, another human who is an expert in city driving, and the third one in off-road conditions. Each expert is confident in their own terrain, but we would like to learn a policy that can perform well in all terrains.

The formal model is similar to the single-expert case, but we now have $M$ experts. For every time step $h \leq H$, the $m$-th expert has an underlying ground truth model $f_h^{\star,m} \in \mathcal{F}_h^m$ that it uses to produce

its label, i.e. for a given state $x_h$ it draws its label as $y_h^m \sim \phi(f_h^{\star,m}(x_h))$, where $\phi$ is the link function. On rounds in which the learner queries for the experts feedback, it gets back a label from each of the $M$ experts, i.e. $\{y_h^1, \ldots, y_h^M\}$. While on every query the learner gets a different label from each expert, its objective is to perform as well as a comparator policy that is defined w.r.t. some ground truth aggregation function that we define next.

The aggregation function $\mathscr{A} : \Delta([K])^M \mapsto \Delta([K])$, known to the learner, combines the recommendation of the $M$ experts to obtain a ground truth label for the corresponding state. In particular, on a given state $x_h$, the label $y_h$ is samples as:

$$y_h \sim \mathscr{A}\big(\phi(f_h^{\star,1}(x_h)), \ldots, \phi(f_h^{\star,M}(x_h))\big). \tag{7}$$

Given the aggregation function $\mathscr{A}$ and the above label generation process, the policy $\pi^\star$ that we wish to compete with in our regret bound is simply the Bayes optimal predictor given by

$$\pi^\star(x_h) = \texttt{SelectAction}(\mathscr{A}(\phi(f_h^{\star,1}(x_h)), \ldots, \phi(f_h^{\star,M}(x_h)))), \tag{8}$$

where $\texttt{SelectAction} : \Delta(K) \mapsto [K]$ is given by $\texttt{SelectAction}(p) = \operatorname{argmax}_{k \in [K]} p[K]$. Some illustrative examples of aggregation functions are given in Appendix F.5. Our main Theorem 5 below bounds the number of label queries to the experts, and regret with respect to this $\pi^\star$, and is obtained using the imitation learning algorithm given in Algorithm 4 in Appendix F.5.

Our bounds depend on a margin term $T_{\varepsilon,h}$, that captures the number of rounds in which the Bayes optimal predictor $\pi^\star$ can flip its label if our estimates of the $M$ experts are off by at most $\varepsilon$ (in $\ell_\infty$ norm). Similar to the single expert case, we only pay in the margin term for time steps in which the counterfactual trajectory w.r.t. the policy $\pi^\star$ has a small-margin. We note that while the trajectories taken by the learner or the noisy experts may go through states that have a large-margin, the margin term $T_{\varepsilon,h}$ that appears in our bounds only accounts for time steps when the comparator policy $\pi^\star$ (the optimal aggregation of expert recommendations) would go to a small-margin region, which could be much smaller. For the ease of notation, we defer the exact definition of margin, and the term $T_{\varepsilon,h}$ to Appendix F.5, and state the main result below:

**Theorem 5.** *Let $\delta \in (0,1)$. Under the modeling assumptions above for the multiple experts setting, with probability at least $1 - \delta$, the imitation learning Algorithm 4 (given in the appendix) obtains:*

$$\text{Reg}_T = \widetilde{\mathcal{O}}\bigg(\inf_\varepsilon\bigg\{H \sum_{h=1}^{H} T_{\varepsilon,h} + \frac{H}{\lambda\varepsilon^2} \sum_{m=1}^{M} \sum_{h=1}^{H} \text{Reg}^{\ell_\phi}(\mathcal{F}_h^m; T)\bigg\}\bigg), \qquad and,$$

$$N_T = \widetilde{\mathcal{O}}\bigg(\inf_\varepsilon\bigg\{H \sum_{h=1}^{H} T_{\varepsilon,h} + \frac{H}{\lambda\varepsilon^2} \sum_{h=1}^{H} \sum_{m=1}^{M} \text{Reg}^{\ell_\phi}(\mathcal{F}_h^m; T) \cdot \mathfrak{E}(\mathcal{F}_h^m, \varepsilon/8; f_h^{\star,m})\bigg\}\bigg).$$

In Appendix A, we evaluate our IL algorithm on the Cartpole environment, with single and multiple experts. We found that our algorithm can match the performance of passive querying algorithms while making a significantly lesser number of expert queries. Finally, note that setting $H = 1$ in the above result, recovers an algorithm, and a similar result for selective sampling with multiple experts.

## Conclusion

In this paper, or goal is to develop algorithms for online IL with active queries with small regret and query complexity bounds. Towards that end, we started by considering the selective sampling setting (IL with $H = 1$), and provided a selective sampling algorithm that can work with general function classes $\mathcal{F}$ and modeling assumptions, and relies on access to an online regression oracle w.r.t. $\mathcal{F}$ to make its predictions (Section 3). The provided regret and query complexity bounds depend on the margin of the expert model. We then extended our selective sampling algorithm to interactive IL (Section 4). For IL, we showed that the margin term that appears in the regret and the query complexity depends on the margin of the expert on counterfactual trajectories that would have been observed on following the expert policy (that we wish to compare to), instead of the trajectories that the learner observes. Thus, if the expert always chooses actions that leads to states where it is confident (i.e. has less margin), the margin term will be smaller. We also considered extensions to learning with multiple experts.

**Acknowledgements**

AS thanks Sasha Rakhlin and Dylan Foster for helpful discussions. AS acknowledges support from the Simons Foundation and NSF through award DMS-2031883, as well as from the DOE through award DE-SC0022199. WS acknowledges support from NSF grant IIS-2154711. KS acknowledges support from NSF CAREER Award 1750575, and LinkedIn-Cornell grant.

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

# Contents of Appendix

# A  Experiments

We conduct experiments to verify our theory. To this end, we first introduce the simulator, *Cart Pole* [Barto et al., 1983, Brockman et al., 2016], and then explain the implementation of our algorithm and the baselines. Finally, we present the results.

**Cart Pole.**  Cart Pole is a classical control problem, in which a pole is attached by an un-actuated joint to a cart. The goal is to balance the pole by applying force to the cart either towards the left or towards the right (so binary action). The episode is terminated once either the pole is out of balance or the cart deviates too far from the origin. A reward of 1 is obtained in each time step (however, the algorithm does not get any reward signal). The observations are four-dimensional, with the values representing the cart's position, velocity, the pole's angle, and angular velocity. The action is binary, indicating the force is either to the left or to the right.

**Expert policies generation.**  We first generate an optimal policy $\pi^\star$ (that attains the maximum possible reward of 500) by policy gradient. We notice that when running the optimal policy $\pi^\star$, the absolute value of the cart's position only lies in $[0, 2]$. Hence, to generate $M$ experts, we first divide this interval into $M$ sub-intervals $[a_0, a_1], [a_1, a_2], \ldots, [a_{n-1}, a_M]$ ($a_0 = 0$ and $a_M = 2$) by geometric progression. For the $i$-th expert, it plays the same action as $\pi^\star$ when the absolute value of the cart's position is in the interval $[a_{i-1}, a_i]$ and plays uniformly at random outside of this interval. We find that using such generation, each expert individually cannot achieve a good performance (when $M > 1$), while a proper combination of them can still be as strong as $\pi^\star$. We conduct experiment for $M = 1, 2, 3$, and $5$, respectively. Given this design of expert generation, when the cart is in the sub-interval $[a_{i-1}, a_i]$, the only expert with non-zero margin is exactly the $i$-th expert.

**Implementation.**  The algorithm is similar to Algorithm 4 but with some modification for practical purpose. First, we use a neural network (single hidden layer neural network, with 4 neurons in the hidden layer) as our function class $\{\mathcal{F}_h^m\}_{h \leq H, m \leq M}$. Second, we specify SelectAction to pick the action of the most confident expert, i.e.,

$$\texttt{SelectAction}(f_{t,h}^1(x), \ldots, f_{t,h}^M(x)) := \text{sign}(f_{t,h}^{\hat{i}}(x)) \quad \text{where} \quad \hat{i} = \arg\max_{i \in [M]} |f_{t,h}^i(x)|.$$

Since we are considering binary action, we assume $f_{t,h}^i(x) \in [-1, 1]$, and the action space is $\{-1, 1\}$. Third, to compute $\Delta_{t,h}^m$ efficiently, we apply the Lagrange multiplier to (60) to arrive at the following equivalent problem:

$$\Delta_{t,h}^m(x_{t,h}) := \min_{f \in \mathcal{F}_h^m} \max_{\alpha \geq 0} -\|f(x_{t,h}) - f_{t,h}^m(x_{t,h})\|$$

$$+ \alpha \left( \sum_{s=1}^{t-1} Z_{s,h} \|f(x_{s,h}) - f_{s,h}^m(x_{s,h})\|^2 - \Psi_\delta^{\ell_\phi}(\mathcal{F}_h^m, T) \right).$$

Then we treat the Lagrange multiplier $\alpha$ as a constant, which converts the problem into the following:

$$\Delta_{t,h}^m(x_{t,h}) := \min_{f \in \mathcal{F}_h^m} -\|f(x_{t,h}) - f_{t,h}^m(x_{t,h})\| + \alpha \sum_{s=1}^{t-1} Z_{s,h} \|f(x_{s,h}) - f_{s,h}^m(x_{s,h})\|^2. \quad (9)$$

The study of varying $\alpha$ is shown in Figure 1. We found that small values (e.g., $\alpha = 1$) mostly lead to poor performance, while the results are fairly similar for large values. In our key experiments, we choose $\alpha = 50$ when the number of experts is 1, 2 or 3, and choose 200 for 5-expert experiments. We note that since computing (9) for each time step involves repetitively fitting neural networks, which is time-consuming, we do a warm start at each round. In particular, we set the initial weights for the neural network of each round to be the weights of the trained network from the previous round. We also implemented *early stopping* that stops the iteration if the loss does not significantly decrease for multiple consecutive iterations. The online regression oracle Oracle is instantiated as applying gradient descent for certain steps on the mean squared loss over all data collected so far, using warm start for speedup as well.

We first conduct experiments on a single expert setting. In Figure 2 we plot the curves of return and number of queries with respect to iterations for our method, and compare to DAgger (which passively makes queries at every time step; Ross and Bagnell [2014]). We note that while our algorithm does

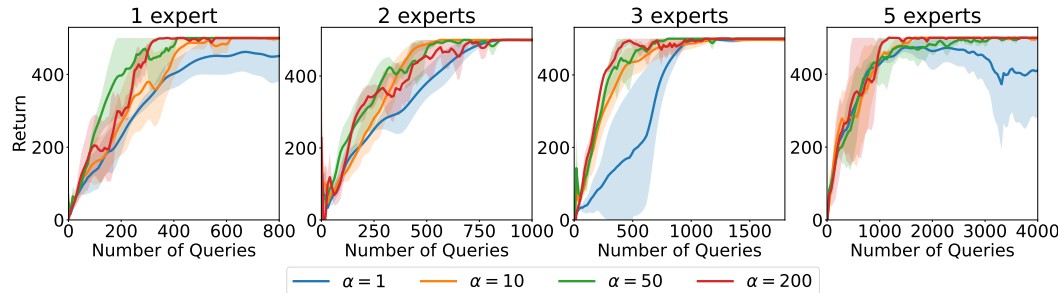

Figure 1: Learning curves of return with respect to the number of queries for different values of $\alpha$ and different numbers of experts.

not converge to the optimal value as fast as DAgger, the number of queries made by our algorithm is significantly fewer, which means that our method is indeed balancing the speed of learning and the number of queries.

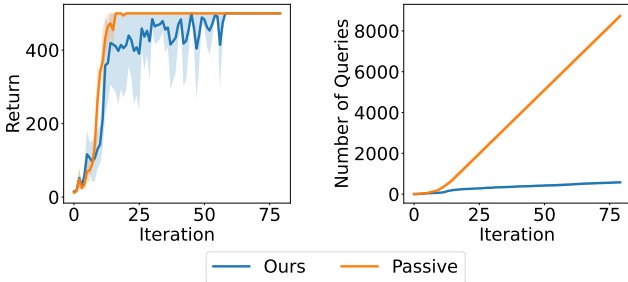

Figure 2: Learning curves of the return and the number of queries for 1 expert.

In additional to DAgger, we also compare to the following baselines:

- **Passive learning.** By passive learning, we mean running our algorithms with $Z_{t,h} = 1$, i.e., making queries whenever possible. Based on different styles of expert feedback, we divide the passive learning baselines into two: *noisy experts* and *noiseless experts*. For the former we get the noisy label $y_{t,h}^m$ for $x_{t,h}$ (generated by $y_{t,h}^m \sim \phi(f_h^{\star,m}(x_{t,h}))$), and for the latter we directly get the action of the optimal policy (i.e. the action $\pi_h^\star(x_{t,h})$). Intuitively, noiseless feedback is more helpful than the noisy one.

- **MAMBA.** We compare our algorithm with (a slight variant of) MAMBA [Cheng et al., 2020]. At each time step, it creates copies of the environment and run each expert policy on these copies, and then it selects the action of the expert policy with the highest return. For simplicity, we refer to this algorithm as MAMBA. Note that MAMBA assumes that one has access to the underlying reward function. Thus this baseline is using significantly more information than our approach.

- **Best expert.** We also compared our algorithm with the best expert policy.

The main results are shown in Figure 3. We first noticed that our algorithm outperforms passive learning with noisy experts in all settings. Moreover, we beat the noiseless version when there is only one expert. Intuitively, getting feedback from noiseless experts is a very strong assumption and it is not surprising to see that the performance is improved with this stronger feedback. Note that our algorithm is only getting noisy labels as feedback. We also note that, despite the fact that MAMBA achieves better results than the best expert policy (in terms of the value function), it is still worse than our algorithm. Indeed, MAMBA does not even learn a policy that can solve the task when $M \geq 2$. This is because by our construction of experts, there is no single expert that is capable of solving the task alone. Note that MAMBA performs well in the one expert case because in that case, the (single) expert can reliably solve the control task.

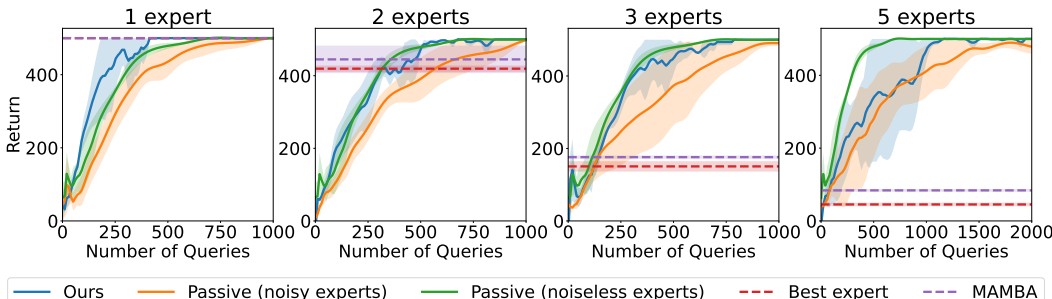

Figure 3: Learning curves of return with respect to the number of queries for different algorithms and numbers of experts.

## B  Further Discussion on Related Works

**Selective Sampling.**   There is a large bank of both theoretical and empirical work for active learning and selective sampling. Perhaps the work closest to ours is the work of Zhu and Nowak [2022]. In this paper, the authors consider binary classification problem and provide bounds on number of queries and bound on excess risk in the active learning framework. Their algorithm also relies on regression oracle. However, there are many key differences: Firstly, their guarantees for regret for selective sampling problem (see for instance Theorem 10 on page 28 of Zhu and Nowak [2022]) has a dependence on disagreement coefficient in the regret bound as well as number of queries. On the contrary, as we show in our work, one only needs to pay for eluder dimension or disagreement coefficient in query complexity and not in regret bound. Furthermore, we supplement our result with lower bound showing that unless one has label complexity that depends on star number (and hence can be also related to worst case disagreement coefficient), one can not get a small enough regret bound. So the separation between regret bound (that is independent of eluder dimension/star number/disagreement coefficient) and query complexity (that depends on those quantities) is real. Secondly, the results in [Zhu and Nowak, 2022] dont automatically adapt to the margin region and in general there is no way to estimate the parameters of Tsybakov's noise condition. Finally, their regret bounds depend on pseudo dimension and are thus generally suboptimal for complex $\mathcal{F}$.

**Imitation Learning.**   IL has enjoyed tremendous research from both theoretical and empirical perspective in the last decade; notable references include Ross et al. [2011], Ross and Bagnell [2014], Sun et al. [2017], Chang et al. [2015], Brantley et al. [2019, 2020], Nguyen and Daumé III [2020]. Ross et al. [2011] initiated research on using online regression oracles to model the expert feedback, and provided regret bounds for IL. The key differences between our work and prior theoretical works on IL are as follows: Firstly, we consider active querying, and provide query complexity bounds for our algorithms. Secondly, and more importantly, we consider interactive IL with noisy expert feedback whereas prior works was restricted to exact expert feedback. Finally, our regret and query complexity bounds scale with the number of times when the comparator policy (induced by the expert) goes to the states where expert has a small margin (instead of the number of times when the learner goes to such states). In many cases, the margin error term corresponding to the comparator policy could be much smaller. On the empirical side, there is a long line of research that provided algorithms and empirical heuristics for making IL sample efficient by modeling the experts in both single expert and multiple expert settings [Beliaev et al., 2022, Cao et al., 2022, Hejna and Sadigh, 2023, Du et al., 2023, Hao et al., 2022]; however most of these algorithm do not come with any rigorous guarantees.

## C  Discussion on Computationally Efficiency

Given the theoretical nature of this paper, our focus is to understand the statistically efficient (query complexity), and to develop algorithms, for selective sampling and imitation learning with general model classes $\mathcal{F}$, given access to an online regression oracle w.r.t. $\mathcal{F}$. For a general function class, our algorithm is computationally inefficient. However, in many cases, our algorithm (or its slight modification) can also be implemented efficiently. We describe some scenarios below:

- **1D-Linear models**: When $f : \mathcal{X} \mapsto \mathbb{R}$ is linear, the optimization objective in (5) in Algorithm 1 (or (6) in Algorithm 2) can be efficiently by instead solving the objectives

$$\Delta_t^{(1)}(x_t) = \max_{f \in \mathcal{F}} f(x_t) - f_t(x_t) \text{ s.t. } \sum_{s=1}^{t-1} Z_s |f(x_s) - f_s(x_s)| \leq \Psi,$$

and

$$\Delta_t^{(2)}(x_t) = \max_{f \in \mathcal{F}} -(f(x_t) - f_t(x_t)) \text{ s.t. } \sum_{s=1}^{t-1} Z_s |f(x_s) - f_s(x_s)| \leq \Psi$$

and then picking the maximum absolute value. Both of these new objectives are linear functions, with convex constraints, and thus can be solved efficiently using a standard solver (e.g. CvXOPT).

- **Differentiable parameterizations**: When class $\mathcal{F}$ could be parameterized in a differentiable way, e.g. using a neural network, we can simply add the constraints as a penalty (with the appropriate multiplicative scale parameter) to convert (5) into an unconstrained optimization problem w.r.t the parameters $\theta$ (of the differentiable parameterization) and then solve it using SGD algorithm. While this is only a heuristic, it works well in practice and is precisely what we do for our experiments in A with 2 layer neural networks.

- **Efficient Implementation of (5) via calls to a Regression Oracles w.r.t. $\mathcal{F}$.** Below we discuss some scenarios and minor modifications of our algorithms under which the computation of $\Delta_t$ (as in (5) in Algorithm 1 or (6) in Algorithm 2) can be performed efficiently via calls to a regression oracle w.r.t. $\mathcal{F}$. Suppose that $\mathcal{F}$ is closed under convexification. We consider two scenarios:

- **Binary Actions Setting**: In this case, we can simply choose $\mathcal{F}$ to be a class of 1D functions, since we can model the expert using a function $f^\star : \mathcal{X} \mapsto \mathbb{R} \in \mathcal{F}$ where for any $x$, $f^\star(x)$ denotes the probability of choosing the first action. The probability of choosing the second action would then be $1 - f^\star(x)$. In this case, (5) in Algorithm 1 simply reduces to

$$\Delta_t'(x_t) = \max_{f \in \mathcal{F}} |f(x_t) - f_t(x_t)| \text{ s.t. } \sum_{s=1}^{t-1} Z_s |f(x_s) - f_s(x_s)| \leq \Psi.$$

The above can be implemented efficiently using the techniques from Foster et al. [2018a]. In particular, let $r_{\max}$ denote the maximum value that $f(x)$ can take. We can solve the above objective using the BINSEARCH procedure in Foster et al. [2018a] where we perform a binary search over a weight parameter $w$, by solving for each $w$ the optimization problems:

$$\arg \min_{f \in \mathcal{F}} \left( w \cdot (f(x_t) - f_t(x_t) - 2r_{\max})^2 + \sum_{s=1}^{t-1} Z_s |f(x_s) - f_s(x_s)| \right)$$

and

$$\arg \min_{f \in \mathcal{F}} \left( w \cdot (f(x_t) - f_t(x_t) + 2r_{\max})^2 + \sum_{s=1}^{t-1} Z_s |f(x_s) - f_s(x_s)| \right)$$

both of which can be efficiently implemented using square loss regression oracles to $\mathcal{F}$. We refer to Foster et al. [2018a] for more details.

- **Multiple Actions Setting:** Suppose that $\mathcal{F}$ is a product class of $\{\mathcal{F}_k\}$ for different actions $k \in [K]$. We can get an oracle-efficient algorithm for a slight modification of (5), at a price of an extra multiplicative $K$ factor in the query complexity bound. Consider the $\Delta_t(x_t)$ given by $\max_{f \in \mathcal{F}} \|f(x_t) - f_t(x_t)\|_\infty$ s.t. $\sum_{s=1}^{t-1} Z_s \|f(x_s) - f_s(x_s)\|_2 \leq \Psi$, which is equal to $\max_{k \in [K]} \max_{f \in \mathcal{F}} |f_k(x_t) - f_{t,k}(x_t)|$ s.t. $\sum_{s=1}^{t-1} Z_s \|f(x_s) - f_s(x_s)\|_2 \leq \Psi$, which can again be implemented via calls to a square loss regression oracle w.r.t. $\mathcal{F}$ by using BINSEARCH procedure in Foster et al. [2018a] (similar to what we did for the Binary actions case above).

# D Useful Tools and Notation

**Additional notation.** Throughout the paper, we assume that the ties are broken arbitrarily but consistently. Vector-valued variables are denoted with small alphabets like $u, v$, etc, and matrix-valued variables are denoted with capital alphabets like $F, G$, etc. For any two distributions $D_1$ and $D_2$, we define $\mathrm{KL}(D_1 \| D_2)$ to denote the KL divergence between $D_1$ and $D_2$. Furthermore, $\mathrm{kl}(b_1 \| b_2)$ denotes the KL divergence between $\mathrm{Bernoulli}(b_1)$ and $\mathrm{Bernoulli}(b_2)$. Finally, we assume that $\|f(x)\| \le B \le 1$ for any $f \in \mathcal{F}$ and $x \in \mathcal{X}$.

The following lemma is used throughout the appendix, and its proof is trivial.

**Lemma 1.** *Let $\mathcal{E}_1$ and $\mathcal{E}_2$ be any two events such that $\mathcal{E}_1 \implies \mathcal{E}_2$ then $\mathbf{1}\{\mathcal{E}_1\} \le \mathbf{1}\{\mathcal{E}_2\}$.*

## D.1 Basic Probabilistic Tools

**Lemma 2** (Theorem 1 in Srebro et al. [2010]). *Let $T > 0$, and let $\mathcal{F} = \{\mathcal{X} \times \mathcal{Y}\}$ be an arbitrary function class, and $\ell$ be an $\gamma$-smooth and non-negative loss such that $|\ell(f(x), y)| \le B$ for all $x \in \mathcal{X}, y \in \mathcal{Y}, f \in \mathcal{F}$. For any $\delta > 0$, we have with probability at least $1 - \delta$ over a random sample of size $T$, for any $f \in \mathcal{F}$,*

$$T \mathbb{E}_{(x,y)\sim\mu}[\ell(f(x), y)] \le 2 \sum_{t=1}^{T} \ell(f(x_t), y_t) + c_1 \big( HT \log^3(T) \mathsf{Rad}_T^2(\mathcal{F}) + B \log(1/\delta) \big)$$

*where $c_1 < 10^5$ is a numeric constant, and $\mathsf{Rad}_T(\mathcal{F})$ denotes the Rademacher complexity of the class $\mathcal{F}$.*

The precise value of the numeric constant $c_1$ in the above can be derived from Srebro et al. [2010] and Mendelson [2002]. Note that for finite function classes, we have $\mathsf{Rad}_T(\mathcal{F}) = \mathcal{O}\left(\sqrt{\log(|\mathcal{F}|)/T}\right)$ and thus the second term above is bounded by $\widetilde{\mathcal{O}}(\log(|\mathcal{F}|))$. In general, we have that $T\mathsf{Rad}_T^2(\mathcal{F}) = \widetilde{\mathcal{O}}(\mathrm{Reg}^{\mathrm{sq}}(\mathcal{F}; T))$ [Rakhlin and Sridharan, 2014], and thus the second term is always dominated by the other terms in our regret and query complexity bounds.

The following inequalities are well-known; we use the version stated in Zhu and Nowak [2022].

**Lemma 3** (Freedman's inequality). *Let $\{X_t\}_{t \le T}$ be a real-valued martingale different sequence adapted to the filtration $\mathfrak{F}_t$, and let $\mathbb{E}_t[\cdot] := \mathbb{E}[\cdot \mid \mathfrak{F}_{t-1}]$. If $|X_t| \le B$ almost surely, then for any $\eta \in (0, 1/B)$, the following holds with probability at least $1 - \delta$:*

$$\sum_{t=1}^{T} X_t \le \eta \sum_{t=1}^{T} \mathbb{E}_t[X_t^2] + \frac{B \log(1/\delta)}{\eta}.$$

**Lemma 4.** *Let $\{X_t\}_{t \le T}$ be a sequence of positive valued random variables adapted to the filtration $\mathfrak{F}_t$, and and let $\mathbb{E}_t[\cdot] := \mathbb{E}[\cdot \mid \mathfrak{F}_{t-1}]$. If $X_t \le B$ almost surely, then with probability at least $1 - \delta$,*

$$\sum_{t=1}^{T} X_t \le \frac{3}{2} \sum_{t=1}^{T} \mathbb{E}_t[X_t] + 4B \log(2/\delta),$$

*and*

$$\sum_{t=1}^{T} \mathbb{E}_t[X_t] \le 2 \sum_{t=1}^{T} X_t + 8B \log(2/\delta).$$

## D.2 Online Learning

**Lemma 5.** *Suppose that the labels are generated according to the (2) where the link function satisfies Assumption 1. Additionally, assume that the regression oracle satisfies the guarantee (3). Then, for any $\delta \le 1/e$ and $T \ge 3$, with probability at least $1 - \delta$, we have for all $t \le T$,*

$$\sum_{s=1}^{t} \|f_s(x_s) - f^\star(x_s)\|^2 \le \Psi_\delta^{\ell_\phi}(\mathcal{F}, T) := \frac{4}{\lambda} \mathrm{Reg}^{\ell_\phi}(\mathcal{F}; T) + \frac{112}{\lambda^2} \log(4 \log^2(T)/\delta),$$

*where $B$ is defined such that $\sup_x f(x) \le B$.*

*Proof.* Using Agarwal [2013, Lemma 2] along with an Union bound implies that for all $t \le T$,

$$\sum_{s=1}^{t} \|f_s(x_s) - f^\star(x_s)\|^2 \le \frac{4}{\lambda} \sum_{s=1}^{t} (\ell_\phi(f_s(x_s), y_s) - \ell_\phi(f^\star(x_s), y_s)) + \frac{112}{\lambda^2} \log(4 \log^2(T)/\delta).$$

Plugging in the regret bound (3) in the above, we get that

$$\sum_{s=1}^{t} \|f_s(x_s) - f^\star(x_s)\|^2 \le \sum_{s=1}^{T} \|f_s(x_s) - f^\star(x_s)\|^2 \le \frac{4}{\lambda} \mathrm{Reg}^{\ell_\phi}(\mathcal{F};T) + \frac{112}{\lambda^2} \log(4 \log^2(T)/\delta).$$

$\square$

### D.3  Eluder Dimension, Disagreement Coefficient, and Star Number

For the sake of completeness, we recall the scalar versions of scale-sensitive eluder dimension, and disagreement coefficient introduced in Russo and Van Roy [2013], Foster et al. [2020], which is defined for a class $\mathcal{F} \subseteq \{\mathcal{X} \mapsto \mathbb{R}\}$ of scalar valued functions.

**Definition 4** (Scale-sensitive eluder dimension (scalar version), Russo and Van Roy [2013], Foster et al. [2020])**.** *Let $\mathcal{F} \subseteq \{\mathcal{X} \mapsto \mathbb{R}\}$. Fix any $f^\star \in \mathcal{F}$, and define $\mathfrak{E}'(\mathcal{F}, \beta; f^\star)$ to be the length of the longest sequence of contexts $x_1, x_2, \ldots x_m$ such that for all $i$, there exists $f_i \in \mathcal{F}$ such that*

$$|f_i(x_i) - f^\star(x_i)| > \beta, \quad \text{and} \quad \sum_{j<i} (f_i(x_j) - f^\star(x_j))^2 \le \beta^2.$$

*We define the scale-sensitive eluder dimension as $\mathfrak{E}(\mathcal{F}, \beta_0; f^\star) := \sup_{\beta_0 \ge \beta} \mathfrak{E}'(\mathcal{F}, \beta; f^\star)$.*

We next provide some examples of function classes with bounded eluder dimension. The examples $(a) - (c)$ first appeared in Russo and Van Roy [2013], and $(d)$ first appeared in Osband and Van Roy [2014].

- $(a)$ For any function class $\mathcal{F} : \{\mathcal{X} \mapsto \mathbb{R}\}$ and $f^\star \in \mathcal{F}$, $\mathfrak{E}(\mathcal{F}, \beta_0; f^\star) \le O(|\mathcal{X}|)$.

- $(b)$ For the class $\mathcal{F}$ of linear functions on a known feature map $\phi$ i.e. , $\mathcal{F} = \{f \mid f(x) = \langle \theta_f, \phi(x) \rangle , \, \theta_f \in \mathbb{R}^d, \|\theta_f\| \le 1\}$, we have $\mathfrak{E}(\mathcal{F}, \beta_0; f^\star) \le O(d \log(1/\varepsilon))$.

- $(c)$ For the class $\mathcal{F}$ of generalized linear functions on a known feature map $\phi$ i.e. , $\mathcal{F} = \{f \mid f(x) = g(\langle \theta_f, \phi(x) \rangle) , \, \theta_f \in \mathbb{R}^d, \|\theta_f\| \le 1\}$ where $g$ is an increasing continuously differentiable function, we have $\mathfrak{E}(\mathcal{F}, \beta_0; f^\star) \le O(dr^2 \log(L/\varepsilon))$ where $r = \sup_{\theta,x} g'(\langle \theta, x \rangle)/\inf_{\theta,x} g'(\langle \theta, x \rangle)$ and $L = \sup_{\theta,x} g'(\langle \theta, \phi(x) \rangle)$.

- $(d)$ For the class $\mathcal{F}$ of quadratic functions on a known feature map $\phi$ i.e. $\mathcal{F} = \{f \mid f(x) = \phi(x)^\top \Sigma_f \phi(x) , \, \Sigma_f \in \mathbb{R}^{d \times d}, \|\Sigma_f\|_F \le 1\}$.

We next recall the definition of scale-sensitive disaggrement coefficient which appears in our bounds for the case of stochastic contexts.

**Definition 5** (Scale-sensitive disagreement coefficient (scalar version), Foster et al. [2020])**.** *Let $\mathcal{F} \subseteq \{\mathcal{X} \mapsto \mathbb{R}\}$. For any $f^\star \in \mathcal{F}$, and $\gamma_0, \varepsilon_0 > 0$ , the value function disagreement coefficient $\theta^{\mathrm{val}}(\mathcal{F}, \varepsilon_0, \gamma_0; f^\star)$ is defined as*

$$\sup_\mu \sup_{\gamma > \gamma_0, \varepsilon > \varepsilon_0} \left\{ \frac{\varepsilon^2}{\gamma^2} \cdot \mathrm{Pr}_{x \sim \mu}(\exists f \in \mathcal{F} \mid |f(x) - f^\star(x)| > \varepsilon, \|f - f^\star\|_\mu \le \gamma) \right\} \vee 1$$

*where $\|f\| = \sqrt{\mathbb{E}_{x \sim \mu}[f^2(x)]}$.*

As we will show in Lemma 6 below, the scale-sensitive disagreement coefficient of $\mathcal{F}$ is always bounded by the eluder dimension of $\mathcal{F}$ upto a constant factor on the dependence on $\varepsilon_0$ and $\gamma_0$. However, the disagreement coefficient can be significantly smaller than the eluder dimension because it can leverage additional distributional structure. We refer the reader to Foster et al. [2020] for bounds on the eluder dimension, and the disagreement coefficient for various function classes. In the following, we extend the above definitions to vector-valued functions to account for the vector-valued function classes that we consider in this work.

**Definition 6** (Scale-sensitive eluder dimension (normed version)). *Let $\mathcal{F} \subseteq \{\mathcal{X} \mapsto \mathbb{R}^K\}$. Fix any $f^\star \in \mathcal{F}$, and define $\widetilde{\mathfrak{E}}(\mathcal{F}, \beta; f^\star)$ to be the length of the longest sequence of contexts $x_1, x_2, \ldots x_m$ such that for all $i$, there exists $f_i \in \mathcal{F}$ such that*

$$\|f_i(x_i) - f^\star(x_i)\| > \beta, \quad and \quad \sum_{j<i} \|f_i(x_j) - f^\star(x_j)\|^2 \le \beta^2.$$

*We define the scale-sensitive eluder dimension as $\mathfrak{E}(\mathcal{F}, \beta'; f^\star) = \sup_{\beta \ge \beta'} \widetilde{\mathfrak{E}}(\mathcal{F}, \beta; f^\star)$.*

We note that the normed eluder dimension can be lower bounded in terms of the eluder dimension of scalar-valued function class obtained by projecting the output of the functions in $\mathcal{F}$ along different coordinates. Let $\mathcal{F}_j = \{P_j f \mid P_j f(x) = f(x)[j], f \in \mathcal{F}\}$, then clearly, for any $f^\star \in \mathcal{F}$, $\mathfrak{E}(\mathcal{F}, \beta; f^\star) \ge \sup_{j \in [d]} \mathfrak{E}(P_j \mathcal{F}, \beta; P_j f^\star)$. Furthermore, we always have that $\mathfrak{E}(\mathcal{F}, \beta; f^\star) \le \kappa \sum_{j=1}^d \mathfrak{E}(P_j \mathcal{F}, \beta; P_j f^\star)$, where $\kappa$ hides $\text{poly}(d)$ factors. We next define the normed version of disagreement coefficient for vector-valued functions.

**Definition 7** (Scale sensitive disagreement coefficient (normed version), Foster et al. [2020]). *Let $\mathcal{F} \subseteq \{\mathcal{X} \mapsto \mathbb{R}^K\}$. For any $f^\star \in \mathcal{F}$, and $\beta_0, \varepsilon_0 > 0$, the value function disagreement coefficient $\theta^{\text{val}}(\mathcal{F}, \varepsilon_0, \beta_0; f^\star)$ is defined as*

$$\sup_{\mu} \sup_{\beta > \beta_0, \varepsilon > \varepsilon_0} \left\{ \frac{\varepsilon^2}{\beta^2} \cdot \Pr_{x \sim \mu}(\exists f \in \mathcal{F} \mid \|f(x) - f^\star(x)\| > \varepsilon, \|f - f^\star\|_\mu \le \beta) \right\} \vee 1$$

*where $\|f\|_\mu = \sqrt{\mathbb{E}_{x \sim \mu}[\|f(x)\|^2]}$.*

We additionally also define the following bivariate version of eluder dimension for vector-valued functions.

**Definition 8** (Scale-sensitive eluder dimension (bivariate version)). *Let $\mathcal{F} \subseteq \{\mathcal{X} \mapsto \mathbb{R}^K\}$. Fix any $f^\star \in \mathcal{F}$, and define $\check{\mathfrak{E}}'(\mathcal{F}, \beta; f^\star)$ to be the length of the longest sequence of contexts and actions $(x_1, y_1), (x_2, y_2) \ldots (x_m, y_m)$ such that for all $i$, there exists $f_i \in \mathcal{F}$ such that*

$$|f_i(x_i)[y_i] - f^\star(x_i)[y_i]| > \beta, \quad and \quad \sum_{j<i} (f_i(x_j)[y_j] - f^\star(x_j)[y_j])^2 \le \beta^2.$$

*We define the scale sensitive eluder dimension (mixed version) as $\check{\mathfrak{E}}(\mathcal{F}, \beta; f^\star) := \sup_{\beta \ge \beta_0} \check{\mathfrak{E}}'(\mathcal{F}, \beta_0; f^\star)$.*

We next define the strong variant of scale-sensitive star number.

**Definition 9** (scale-sensitive star number (strong version), Foster et al. [2020]). *Let $\mathcal{F} \subseteq \{\mathcal{X} \mapsto \mathbb{R}^K\}$. For any $f^\star \in \mathcal{F}$ and $\beta > 0$, let $\underline{\check{\mathfrak{s}}}^{\text{val}}(\mathcal{F}, \beta)$ denote the length of the longest sequence of contexts $\{x_1, \ldots, x_m\}$ such that for all $i$, there exists $f_i \in \mathcal{F}$ such that*

$$\|f_i(x_i) - f^\star(x_i)\| > \beta, \quad and \quad \sum_{j \ne i} \|f_i(x_j) - f^\star(x_j)\|^2 \le \beta^2.$$

*We define the scale-sensitive star number as $\check{\mathfrak{s}}^{\text{val}}(\mathcal{F}, \beta) := \sup_{\beta > \beta_0} \underline{\check{\mathfrak{s}}}^{\text{val}}(\mathcal{F}, \beta_0)$.*

The next result provides a relation between the star number, disagreement coefficient and the eluder dimension.

**Lemma 6** (Foster et al. [2020]). *Suppose $\mathcal{F} \subseteq \{\mathcal{X} \mapsto \mathbb{R}^K\}$ is a uniform Glivenko-Cantelli class. For any $f^\star \in \mathcal{F}$ and $\gamma, \varepsilon > 0$, we have $\mathfrak{s}^{\text{val}}(\mathcal{F}, \beta; f^\star) \le \mathfrak{E}(\mathcal{F}, \beta; f^\star)$, $\theta^{\text{val}}(\mathcal{F}, \varepsilon, \gamma; f^\star) \le 4(\mathfrak{s}^{\text{val}}(\mathcal{F}, \gamma; f^\star))^2$ and $\theta^{\text{val}}(\mathcal{F}, \varepsilon, \gamma; f^\star) \le 4\mathfrak{E}(\mathcal{F}, \gamma; f^\star)$.*

The following two technical lemmas are useful in bounding the total number of queries made by our selective sampling and imitation learning algorithms. We first provide a technical result which bounds the number of times we can find a function $f'$ in a refinement $\mathcal{F}_t$ of $\mathcal{F}$, such that $f'$ is sufficiently far away from $f^\star \in \mathcal{F}$. This result is a variant of Russo and Van Roy [2013, Lemma 3], and first appears in Foster et al. [2020, Lemma E.4].

**Lemma 7.** *Let $\{x_t, y_t, Z_t\}_{t=1}^T$ be sequence of tuples, where $x_t \in \mathcal{X}$ and $Z_t \in \{0, 1\}$. Fix any $f^\star \in \mathcal{F}$, and define the set $\mathcal{F}_t = \{f \in \mathcal{F} \mid \sum_{s=1}^{t-1} Z_s(f(x_s)[y_s] - f^\star(x_s)[y_s])^2 \le \beta^2\}$. Then, for any $\zeta > 0$,*

$$\sum_{t=1}^T Z_t \mathbf{1}\{\exists f' \in \mathcal{F}_t : (f'(x_t)[y_t] - f^\star(x_t)[y_t]) \ge \zeta\} \le \left(\frac{\beta^2}{\zeta^2} + 1\right) \check{\mathfrak{E}}(\mathcal{F}, \zeta; f^\star).$$

*Proof.* We first note that we can always remove a tuple $\{(x_t, y_t), Z_t\}$ whenever $Z_t = 0$ without any effect on the conclusion. Hence, we can assume $Z_t = 1$ for all $t \in [T]$ without loss of generality. Then the rest of the proof essentialy follows from Foster et al. [2020, Lemma E.4]. For completeness, we state the full proof here.

For simplicity of presentation, we say $(x_t, y_t)$ is $\zeta$-independent of $(x_1, y_1), \ldots, (x_{t-1}, y_{t-1})$ if there exists $f \in \mathcal{F}$ such that $|f(x_t)[y_t] - f^\star(x_t)[y_t]| \geq \zeta$ and $\sum_{s=1}^{t-1}(f(x_s)[y_s] - f^\star(x_s)[y_s])^2 \leq \zeta^2$. Otherwise, we say $x$ is $\zeta$-dependent. The proof consists of the following two claims.

First, we claim that for any $t \in [T]$, if there exists $f \in \mathcal{F}_t$ such that $|f(x_t)[y_t] - f^\star(x_t)[y_t]| \geq \zeta$, then $x_t$ is $\zeta$-dependent on at most $\beta^2/\zeta^2$ disjoint sequences of $(x_1, y_1), \ldots, (x_{t-1}, y_{t-1})$. To show this, let's say $x_t$ is $\zeta$-dependent on a particular subsequence $(x_{i_1}, y_{i_1}), \ldots, (x_{i_k}, y_{i_k})$ while $|f(x)[y] - f^\star(x)[y]| \geq \zeta$. Then it must holds that

$$\sum_{j=1}^{k}\left(f(x_{i_j})[y_{i_j}] - f^\star(x_{i_j})[y_{i_j}]\right)^2 \geq \zeta^2.$$

If there are $M$ such disjoint subsequence, then we can add them up and obtain the following:

$$\sum_{s=1}^{t-1}\left(f(x_s)[y_s] - f^\star(x_s)[y_s]\right)^2 \geq M\zeta^2.$$

By the construction of $\mathcal{F}_t$, the left-hand side above is at most $\beta^2$. Hence we conclude that $\beta^2 \geq M\zeta^2$, which implies $M \leq \beta^2/\zeta^2$.

Second, we claim that for any $k$ and any sequence $(x_1, y_1), \ldots, (x_k, y_k)$, there exists $j \leq k$ such that $x_j$ is $\zeta$-dependent on at least $N := \lfloor k/\check{\mathfrak{E}}(\mathcal{F}, \zeta; f^\star)\rfloor$ disjoint subsequences of $(x_1, y_1), \ldots, (x_{j-1}, y_{j-1})$. This can be proved by construction. Let $B_1, \ldots, B_N$ be $N$ subsequences of $(x_1, y_1), \ldots, (x_k, y_k)$ and are initialized with $B_i = \{(x_i, y_i)\}$. Then we repeat the following process for $j = N + 1, N + 2 \ldots, k$.

- We first check if $x_j$ is $\zeta$-dependent on $B_i$ for all $i \in [N]$. If so, we are done.

- Otherwise, pick an arbitrary $i \in [N]$ for which $x_j$ is $\zeta$-independent of $B_i$ and append $(x_j, y_j)$ to $B_i$, i.e., $B_i \leftarrow B_i \cup \{(x_j, y_j)\}$.

If we don't reach any $j$ while running the above process for which the first statement above is satisfied, we should end up with $\sum_{i=1}^{N}|B_i| = k \geq N \cdot \check{\mathfrak{E}}(\mathcal{F}, \zeta; f^\star)$. We note that by construction $|B_i| \leq \check{\mathfrak{E}}(\mathcal{F}, \zeta; f^\star)$ and thus $|B_i| = \check{\mathfrak{E}}(\mathcal{F}, \zeta; f^\star)$ for all $i \in [N]$, which implies $x_k$ must be $\zeta$-dependent on all $B_i$.

Finally, let $x_{i_1}, \ldots, x_{i_k}$ be the subsequence where, for all $s \in [k]$, there exists $f \in \mathcal{F}_{i_s}$ such that $|f(x_{i_s})[y_{i_s}] - f^\star(x_{i_s})[y_{i_s}]| \geq \zeta$. By our first claim we know each element of this subsequence is $\zeta$-dependent on at most $\beta^2/\zeta^2$ disjoint subsequences. By the second claim, we know that there exists an element that is $\zeta$-dependent on at least $\lfloor k/\check{\mathfrak{E}}(\mathcal{F}, \zeta; f^\star)\rfloor$ disjoint subsequences. So we must have $\lfloor k/\check{\mathfrak{E}}(\mathcal{F}, \zeta; f^\star)\rfloor \leq \beta^2/\zeta^2$. Hence, $k \leq (\beta^2/\zeta^2 + 1) \cdot \check{\mathfrak{E}}(\mathcal{F}, \zeta; f^\star)$. □

The following is an extension of Lemma 7 that holds for the normed version of eluder dimension given in Definition 1. The proof is essentially the same so we skip it for conciseness.

**Lemma 8.** *Let $\{x_t, Z_t\}_{t=1}^{T}$ be sequence of tuples, where $x_t \in \mathcal{X}$ and $Z_t \in \{0, 1\}$. Fix any $f^\star \in \mathcal{F}$, and define the set $\mathcal{F}_t = \{f \in \mathcal{F} \mid \sum_{s=1}^{t-1} Z_s\|f(x_s) - f^\star(x_s)\|^2 \leq \beta^2\}$. Then, for any $\zeta > 0$,*

$$\sum_{t=1}^{T} Z_t \mathbf{1}\{\exists f' \in \mathcal{F}_t : \|f'(x_t) - f^\star(x_t)\| \geq \zeta\} \leq \left(\frac{\beta^2}{\zeta^2} + 1\right)\mathfrak{E}(\mathcal{F}, \zeta; f^\star).$$

# E Selective Sampling: Learning from Single Expert

## E.1 Comparison to Related Works

**Selective Sampling.** There is a large bank of both theoretical and empirical work for active learning and selective sampling. Perhaps the work closest to ours is the work of Zhu and Nowak [2022]. In this paper, the authors consider binary classification problem and provide bounds on number of queries and bound on excess risk in the active learning framework. Their algorithm also relies on regression oracle. However, there are many key differences: Firstly, their guarantees for regret for selective sampling problem (see for instance Theorem 10 on page 28 of Zhu and Nowak [2022]) has a dependence on disagreement coefficient in the regret bound as well as number of queries. On the contrary, as we show in our work, one only needs to pay for eluder dimension or disagreement coefficient in query complexity and not in regret bound. Furthermore, we supplement our result with lower bound showing that unless one has label complexity that depends on star number (and hence can be also related to worst case disagreement coefficient), one can not get a small enough regret bound. So the separation between regret bound (that is independent of eluder dimension/star number/disagreement coefficient) and query complexity (that depends on those quantities) is real. Secondly, the results in [Zhu and Nowak, 2022] dont automatically adapt to the margin region and in general there is no way to estimate the parameters of Tsybakov's noise condition. Finally, their regret bounds depend on pseudo dimension and are thus generally suboptimal for complex $\mathcal{F}$.

## E.2 Proof Sketch for Selective Sampling and Binary Labels

Let $\mathcal{A} = \{1, 2\}$, and the link function $\phi(z) = z$ corresponding to square-loss $\ell_\phi = (v - y)^2/2$; here $\lambda = \gamma = 1$.

Let $\bar{\mathcal{F}} \subseteq \{\mathcal{X} \mapsto [-1, 1]\}$ be a function class, and $\bar{f}^\star \in \mathcal{F}$. We assume that for any context $x$, the label $y$ is drawn according to the distribution $\Pr(y_t = 2) = 1+\bar{f}^\star(x)/2$. Using $\bar{\mathcal{F}}$, we can define the score function class $\mathcal{F} = \{f_{\bar{f}} \mid \bar{f} \in \bar{\mathcal{F}}\}$ where $f(x) = \frac{1}{2}(1 - \bar{f}(x), 1 + \bar{f}(x))^\top \in [0, 1]^2$, and additionally define $f^\star = f_{\bar{f}^\star}$. Clearly, the Bayes optimal predictor that chooses the action with the largest score is given by $\texttt{SelectAction}(f^\star(x)) = 1 + \text{sign}(\bar{f}^\star(x))$. Furthermore, $\texttt{Margin}(f^\star(x)) :=$ $|\Pr(y = 2 \mid x) - \Pr(y = 1 \mid x)| = |\bar{f}^\star(x)|$ which implies that $T_\varepsilon = \sum_{t=1}^T \mathbf{1}\{|\bar{f}^\star(x_t)| \le \varepsilon\}$. Finally, the oracle in (3) reduces to a square-loss online regression oracle, which implies that with probability at least $1 - \delta$, for all $t \le T$,

$$\sum_{s=1}^t Z_s(\bar{f}_s(x_s) - \bar{f}^\star(x_s))^2 \lesssim \sum_{s=1}^t Z_s(\bar{f}_s(x_s) - y_s)^2 - \sum_{s=1}^t Z_s(\bar{f}^\star(x_s) - y_s)^2 \lesssim \text{Reg}^{\text{sq}}(\bar{\mathcal{F}}; T) + \log(T/\delta), \tag{10}$$

The above implies that $\bar{f}^\star$ satisfies the constraints in (5) with the right choice of constants, $\lambda$, and $\gamma$, and thus $|\bar{f}_t(x_t) - \bar{f}^\star(x_t)| \le \Delta_t(x_t)$ (see Lemma 10 for proof). However, since the query condition in Algorithm 1 is $Z_t = \mathbf{1}\{|\bar{f}_t(x_t)| \le \Delta_t(x_t)\}$, we have that if $Z_t = 0$, then $|\bar{f}_t(x_t)| > \Delta_t(x_t)$ which implies that $\text{sign}(\bar{f}^\star(x_t)) = \text{sign}(\bar{f}_t(x_t))$. Thus,

$$\sum_{s=1}^t \bar{Z}_s \mathbf{1}\{\text{sign}(\bar{f}^\star(x_t)) \ne \text{sign}(\bar{f}_t(x_t))\} = 0. \tag{11}$$

*Regret bound.* Using the fact that $y_t \sim 1 + \text{Ber}(1+\bar{f}^\star(x_t)/2)$, $\widehat{y}_t = \texttt{SelectAction}(f_t(x_t)) = 1 + \text{sign}(\bar{f}_t(x_t))$, we have

$$\text{Reg}_T = \sum_{t=1}^T \Pr(\widehat{y}_t \ne y_t) - \Pr(\texttt{SelectAction}(f^\star(x_t)) \ne y_t)$$

$$\le \sum_{t=1}^T \mathbf{1}\{\text{sign}(\bar{f}_t(x_t)) \ne \text{sign}(\bar{f}^\star(x_t))\} \cdot |2\Pr(y_t = 1) - 1|$$

$$= \sum_{t=1}^T \mathbf{1}\{\text{sign}(\bar{f}_t(x_t)) \ne \text{sign}(\bar{f}^\star(x_t))\} \cdot |\bar{f}^\star(x_t)|$$

The right hand side above can be split and upper bound via the following three terms:

$$\text{Reg}_T \le \varepsilon \sum_{t=1}^T \mathbf{1}\{|\bar{f}^\star(x_t)| \le \varepsilon\} + \sum_{t=1}^T Z_t \mathbf{1}\{\text{sign}(\bar{f}_t(x_t)) \ne \text{sign}(\bar{f}^\star(x_t)), |\bar{f}^\star(x_t)| > \varepsilon\} \cdot |\bar{f}^\star(x_t)|$$

$$+ \sum_{t=1}^{T} \bar{Z}_t \mathbf{1}\{\text{sign}(\bar{f}_t(x_t)) \neq \text{sign}(\bar{f}^\star(x_t))\} \cdot |\bar{f}^\star(x_t)|.$$

$$= \varepsilon T_\varepsilon + \underbrace{\sum_{t=1}^{T} Z_t \mathbf{1}\{\text{sign}(\bar{f}_t(x_t)) \neq \text{sign}(\bar{f}^\star(x_t)), |\bar{f}^\star(x_t)| > \varepsilon\} \cdot |\bar{f}^\star(x_t)|}_{:=\mathsf{T}_A},$$

where the first term is $T_\varepsilon$, and the last term is zero due to (11). The term $\mathsf{T}_A$ denotes the regret for the rounds in which the learner queries for the label, and the margin for $\bar{f}^\star(x_t)$ is larger than $\varepsilon$. We note that

$$\mathsf{T}_A \leq \sum_{t=1}^{T} Z_t \mathbf{1}\{|\bar{f}^\star(x_t) - \bar{f}_t(x_t)| > \varepsilon\} \cdot |\bar{f}^\star(x_t) - \bar{f}_t(x_t)|$$

where the inequality holds because $|\bar{f}^\star(x_t) - \bar{f}_t(x_t)| \geq |\bar{f}^\star(x_t)|$ since they have opposite signs. Using the fact that $\mathbf{1}\{a \geq b\} \leq a/b$ for all $a, b \geq 0$, and the bound in (10), we get

$$\mathsf{T}_A \leq \frac{1}{\varepsilon} \sum_{t=1}^{T} Z_t (\bar{f}^\star(x_t) - \bar{f}_t(x_t))^2 \lesssim \frac{1}{\varepsilon} \text{Reg}^{\text{sq}}(\mathcal{F}; T) + \frac{1}{\varepsilon} \log(T/\delta),$$

Gathering all the terms, we get

$$\text{Reg}_T = \widetilde{\mathcal{O}}\left(\varepsilon T_\varepsilon + \frac{1}{\varepsilon} \text{Reg}^{\text{sq}}(\mathcal{F}; T) + \frac{1}{\varepsilon} \log(1/\delta)\right).$$

*Query complexity.* Plugging in the query rule, and splitting as in the regret bound, we get

$$N_T = \sum_{t=1}^{T} Z_t = \sum_{t=1}^{T} \mathbf{1}\{|\bar{f}_t(x_t)| \leq \Delta_t(x_t)\}$$

$$\leq \underbrace{\sum_{t=1}^{T} \mathbf{1}\{|\bar{f}^\star(x_t)| \leq \varepsilon\}}_{=T_\varepsilon} + \underbrace{\sum_{t=1}^{T} \mathbf{1}\{|\bar{f}_t(x_t)| \leq \Delta_t(x_t), |\bar{f}^\star(x_t)| > \varepsilon, \Delta_t(x_t) \leq \varepsilon/3\}}_{:=\mathsf{T}_C}$$

$$+ \underbrace{\sum_{t=1}^{T} \mathbf{1}\{|\bar{f}_t(x_t)| \leq \Delta_t(x_t), |\bar{f}^\star(x_t)| > \varepsilon, \Delta_t(x_t) > \varepsilon/3\}}_{:=\mathsf{T}_D}$$

$\mathsf{T}_C$ denotes the rounds in which we make a query, $\Delta_t(x_t) \leq \varepsilon/3$, and the margin for $\bar{f}^\star(x_t)$ is larger than $\varepsilon$. Since $|\bar{f}_t(x_t) - \bar{f}^\star(x_t)| \leq \Delta_t(x_t)$ (as shown above), we have

$$|\bar{f}^\star(x_t)| \leq |\bar{f}_t(x_t) - \bar{f}^\star(x_t)| + |\bar{f}_t(x_t)| \leq \Delta_t(x_t) + |\bar{f}_t(x_t)|.$$

Thus,

$$\mathsf{T}_C \leq \sum_{t=1}^{T} \mathbf{1}\{|\bar{f}^\star(x_t)| \leq 2\Delta_t(x_t), |\bar{f}^\star(x_t)| > \varepsilon, \Delta_t(x_t) \leq \varepsilon/3\} = 0.$$

$\mathsf{T}_D$ is bounded by the number of rounds for which we make a query and $\Delta_t(x_t) \geq \varepsilon/3$. Using the properties of eluder dimension, we get that

$$\mathsf{T}_D \leq \sum_{t=1}^{T} Z_t \mathbf{1}\{\Delta_t(x_t) \geq \varepsilon/3\} \lesssim \frac{1}{\varepsilon^2} \text{Reg}^{\text{sq}}(\mathcal{F}; T) \cdot \mathfrak{E}(\mathcal{F}, \varepsilon/6; \bar{f}^\star) + \log(1/\delta).$$

Gathering all the terms, we conclude

$$N_T = \widetilde{\mathcal{O}}\left(T_\varepsilon + \frac{1}{\varepsilon^2} \text{Reg}^{\text{sq}}(\mathcal{F}; T) \cdot \mathfrak{E}(\mathcal{F}, \varepsilon/6; \bar{f}^\star) + \frac{1}{\varepsilon^2} \log(1/\delta)\right).$$

In Appendix E.3, we provide the complete proof and show how to generalize it for multiple actions, link function $\phi$ and corresponding regression oracles w.r.t. $\ell_\phi$.

### E.3 Proof of Theorem 1

Before delving into the proof, we recall the relevant notation. In Algorithm 1,

- The label $y_t \sim \phi(f^\star(x_t))$, where $\phi$ denotes the link-function given in (2).
- The function $\texttt{SelectAction}(f_t(x_t)) \coloneqq \mathrm{argmax}_k\, \phi(f_t(x_t))[k]$.
- For any vector $v \in \mathbb{R}^K$, the margin is given by the gap between the value at the largest and the second largest coordinate, i.e.

$$\texttt{Margin}(v) = \phi(v)[k^\star] - \max_{k \neq k^\star} \phi(v)[k],$$

  where $k^\star \in \mathrm{argmax}_{k \in [K]} \phi(v)[k]$.

- We also define $T_\varepsilon = \sum_{t=1}^T \mathbf{1}\{\texttt{Margin}(f^\star(x_t)) \leq \varepsilon\}$ to denote the number of samples within $T$ rounds of interaction for which the margin w.r.t. $f^\star$ is smaller than $\varepsilon$.
- We define the function $\mathrm{Gap} : \mathbb{R}^K \times [K] \mapsto \mathbb{R}^+$ as

$$\mathrm{Gap}(v, k) = \max_{k'} \phi(v)[k'] - \phi(v)[k], \tag{12}$$

  to denote the gap between the largest and the $k$-th coordinate of $v$.

#### E.3.1 Supporting Technical Results

**Lemma 9.** *For any $u$, and $k' \neq \mathrm{argmax}_k \phi(u)[k]$,*

$$\texttt{Margin}(u) \leq \mathrm{Gap}(u, k').$$

*Proof.* Let $k^\star = \mathrm{argmax}_k \phi(u)[k]$. By definition,

$$\begin{aligned}
\mathrm{Gap}(u, k') &= \phi(u)[k^\star] - \phi(u)[k'] \\
&\geq \phi(u)[k^\star] - \max_{k' \neq k} \phi(u)[k'] = \texttt{Margin}(u).
\end{aligned}$$

$\square$

The following technical result establishes a certain favorable property for the function $f^\star$, whose proof follows from the regret bound of the online oracle used in Algorithm 1.

**Lemma 10.** *With probability at least $1 - \delta$, the function $f^\star \in \mathcal{F}$ satisfies the following for all $t \leq T$:*

$$\sum_{s=1}^t Z_s \|f^\star(x_s) - f_s(x_s)\|^2 \leq \Psi_\delta^{\ell_\phi}(\mathcal{F}, T),$$

*where $\Psi_\delta^{\ell_\phi}(\mathcal{F}, T) \coloneqq \frac{4}{\lambda} \mathrm{Reg}^{\ell_\phi}(\mathcal{F}; T) + \frac{112}{\lambda^2} \log(4 \log^2(T)/\delta)$.*

*Proof.* The desired result follows from an application of Lemma 5, where we note that we do not query oracle when $Z_s = 0$, and thus do not count the time steps for which $Z_s = 0$. $\square$

Throughout the proof, we condition on the $1 - \delta$ probability event that Lemma 10 holds. The next technical lemma allows us to bound the number of times when we query for the label and $\Delta_t(x_t) \geq \zeta$ in terms of the eluder dimension (normed version) of the function class $\mathcal{F}$. Note that Lemma 11 holds even if the sequence $\{x_t\}_{t \leq T}$ could be adversarially generated.

**Lemma 11.** *Let $f^\star$ satisfy Lemma 10, and let $\Delta_t(x_t)$ be defined in (5) in Algorithm 1. Then, for any $\zeta > 0$, with probability at least $1 - \delta$,*

$$\sum_{t=1}^T Z_t \mathbf{1}\{\Delta_t(x_t) \geq \zeta\} \leq \widetilde{O}\left( \frac{\Psi_\delta^{\ell_\phi}(\mathcal{F}, T)}{\zeta^2} \cdot \mathfrak{E}(\mathcal{F}, \zeta/2; f^\star) \right).$$

*where $\mathfrak{E}$ denotes the eluder dimension is given in Definition 1.*

*Proof.* Let $f_t^\star$ denote the maximizer of (5) at round $t$ on point $x_t$. Thus,

$$\Delta_t(x_t) = \|f_t^\star(x_t) - f_t(x_t)\|, \qquad \text{and} \qquad \sum_{s=1}^{t-1} Z_s \|f_t^\star(x_s) - f_s(x_s)\|^2 \leq \Psi_\delta^{\ell_\phi}(\mathcal{F}, T). \tag{13}$$

However, recall that Lemma 10 implies that, with probability at least $1 - \delta$, the function $f^\star$ satisfies the bound

$$\sum_{s=1}^{t-1} Z_s \|f^\star(x_s) - f_s(x_s)\|^2 \leq \Psi_\delta^{\ell_\phi}(\mathcal{F}, T). \tag{14}$$

Using (13), (14) and Triangle inequality, we get that

$$\sum_{s=1}^{t-1} Z_s \|f_t^\star(x_s) - f^\star(x_s)\|^2 \leq 2 \sum_{s=1}^{t-1} Z_s \|f_t^\star(x_t) - f_s(x_s)\|^2 + 2 \sum_{s=1}^{t-1} Z_s \|f^\star(x_t) - f_s(x_s)\|^2$$

$$\leq 4\Psi_\delta^{\ell_\phi}(\mathcal{F}, T). \tag{15}$$

Next, note that, an application of Triangle inequality implies that $\|f_t^\star(x_t) - f_t(x_t)\| \leq \|f_t^\star(x_t) - f^\star(x_t)\| + \|f^\star(x_t) - f_t(x_t)\|$. Thus,

$$\sum_{t=1}^{T} Z_t \mathbf{1}\{\Delta_t(x_t) \geq \zeta\} = \sum_{t=1}^{T} Z_t \mathbf{1}\{\|f_t^\star(x_t) - f_t(x_t)\| \geq \zeta\}$$

$$\leq \sum_{t=1}^{T} Z_t \mathbf{1}\{\|f_t^\star(x_t) - f^\star(x_t)\| + \|f^\star(x_t) - f_t(x_t)\| \geq \zeta\}$$

$$\leq \sum_{t=1}^{T} Z_t \mathbf{1}\left\{\|f_t^\star(x_t) - f^\star(x_t)\| \geq \frac{\zeta}{2}\right\} + \sum_{t=1}^{T} Z_t \mathbf{1}\left\{\|f_t(x_t) - f^\star(x_t)\| \geq \frac{\zeta}{2}\right\}$$

$$\leq \sum_{t=1}^{T} Z_t \mathbf{1}\left\{\|f_t^\star(x_t) - f^\star(x_t)\| \geq \frac{\zeta}{2}\right\} + \frac{4}{\zeta^2} \sum_{t=1}^{T} Z_t (f_t(x_t) - f^\star(x_t))^2$$

$$\leq \sum_{t=1}^{T} Z_t \mathbf{1}\left\{\|f_t^\star(x_t) - f^\star(x_t)\| \geq \frac{\zeta}{2}\right\} + \frac{4\Psi_\delta^{\ell_\phi}(\mathcal{F}, T)}{\zeta^2}, \tag{16}$$

where in the last line we used Lemma 10 to bound the second term. In the following, we show how to bound the first term. Recall that for any $t \leq T$, the function $f_t^\star$ satisfies (15). Thus, we wish to bound

$$\sum_{t=1}^{T} Z_t \mathbf{1}\left\{\|f_t^\star(x_t) - f^\star(x_t)\| \geq \frac{\zeta}{2}\right\} \quad \text{s.t.} \quad \sum_{s=1}^{t-1} Z_s (f_t^\star(x_s) - f^\star(x_s))^2 \leq 4\Psi_\delta^{\ell_\phi}(\mathcal{F}, T),$$

for all $t \leq T$. An application of Lemma 8 in the above implies that

$$\sum_{t=1}^{T} Z_t \mathbf{1}\left\{\|f_t^\star(x_t) - f^\star(x_t)\| \geq \frac{\zeta}{2}\right\} \leq \frac{17\Psi_\delta^{\ell_\phi}(\mathcal{F}, T)}{\zeta^2} \cdot \mathfrak{E}(\mathcal{F}, \zeta/2; f^\star). \tag{17}$$

where in the last line, we used the fact that $\Psi_\delta^{\ell_\phi}(\mathcal{F}, T)/\zeta^2 \geq 1$, for our parameter setting.

Plugging in the bound (17) in (16), and using the fact that $\mathfrak{E}(\mathcal{F}, \zeta/2; f^\star) \geq 1$, we get that

$$\sum_{t=1}^{T} Z_t \mathbf{1}\{\Delta_t(x_t) \geq \zeta\} \leq \frac{20\Psi_\delta^{\ell_\phi}(\mathcal{F}, T)}{\zeta^2} \cdot \mathfrak{E}(\mathcal{F}, \zeta/2; f^\star).$$

$\square$

The next two technical lemma's relate the margin to the gap between functions, and are useful in the analysis for regret / total number of queries.

**Lemma 12.** *Suppose the functions $\pi_1$ and $\pi_2$ are defined such that $\pi_i(x) = \operatorname{argmax}_{k \in [K]} \phi(f_i(x))[k]$. Then, for any $x$ for which $\pi_1(x) \neq \pi_2(x)$, we have*

$$\texttt{Margin}(f_1(x)) \leq \phi(f_1(x))[\pi_1(x)] - \phi(f_1(x))[\pi_2(x)] \leq 2\gamma \|f_1(x) - f_2(x)\|_2,$$

*where $\gamma$-denotes the Lipschitz parameter of the link function $\phi$.*

*Proof.* First note $\phi(f_2(x))[\pi_2(x)] \geq \phi(f_2(x))[\pi_1(x)]$ by the definition of $\pi_2$. Thus,

$\phi(f_1(x))[\pi_1(x)] - \phi(f_1(x))[\pi_2(x)]$
$$\leq \phi(f_1(x))[\pi_1(x)] - \phi(f_2(x))[\pi_1(x)] + \phi(f_2(x))[\pi_2(x)] - \phi(f_1(x))[\pi_2(x)]$$
$$\leq 2\|\phi(f_1(x)) - \phi(f_2(x))\|_\infty$$
$$\leq 2\|\phi(f_1(x)) - \phi(f_2(x))\|_2.$$

Using the fact that $\phi$ is $\gamma$-Lipschitz, we immediately get that

$$\phi(f_1(x))[\pi_1(x)] - \phi(f_1(x))[\pi_2(x)] \leq 2\gamma\|f_1(x) - f_2(x)\|_2.$$

$\square$

**Lemma 13.** *For any two function $f_1, f_2 \in \mathcal{F}$, and $x \in \mathcal{X}$,*
$$\text{Margin}(f_1(x)) - \text{Margin}(f_2(x)) \leq 2\gamma\|f_1(x) - f_2(x)\|.$$

*Proof.* For the ease of notation, define
$$k_1 = \underset{k \in [k]}{\text{argmax}}\, \phi(f_1(x))[k] \qquad \text{and} \qquad k_1' = \underset{k' \neq k_1}{\text{argmax}}\, \phi(f_1(x))[k'],$$

where ties are broken arbitrarily but consistently. Similarly, we define
$$k_2 = \underset{k \in [k]}{\text{argmax}}\, \phi(f_2(x))[k] \qquad \text{and} \qquad k_2' = \underset{k' \neq k_2}{\text{argmax}}\, \phi(f_2(x))[k']. \tag{18}$$

Thus, we have that
$$\text{Margin}(f_1(x)) = \phi(f_1(x))[k_1] - \phi(f_1(x))[k_1'],$$

and
$$\text{Margin}(f_2(x)) = \phi(f_2(x))[k_2] - \phi(f_2(x))[k_2']. \tag{19}$$

Finally, also note that for any coordinate $k$,
$$\phi(f_1(x))[k] - \phi(f_2(x))[k] \leq \|\phi(f_2(x)) - \phi(f_1(x))\|. \tag{20}$$

We now proceed with the proof. Plugging in the form in (19), we get that

$\text{Margin}(f_1(x)) - \text{Margin}(f_2(x))$
$$= \phi(f_1(x))[k_1] - \phi(f_1(x))[k_1'] - (\phi(f_2(x))[k_2] - \phi(f_2(x))[k_2'])$$
$$= (\phi(f_1(x))[k_1] - \phi(f_2(x))[k_2]) + (\phi(f_2(x))[k_2'] - \phi(f_1(x))[k_1'])$$
$$\leq (\phi(f_1(x))[k_1] - \phi(f_2(x))[k_1]) + (\phi(f_2(x))[k_2'] - \phi(f_1(x))[k_1'])$$
$$\leq \|\phi(f_2(x)) - \phi(f_1(x))\| + (\phi(f_2(x))[k_2'] - \phi(f_1(x))[k_1']),$$

where the first inequality uses the fact that $k_2$ is the maximizer coordinate of $\phi(f_2(x))$ and the last inequality uses (20). In the following, we bound the second term in the right hand side above under the following three cases:

- *Case 1: $k_2' \neq k_1$:* Since $k_2' \neq k_1$, we note that replacing $k_1'$ by $k_2'$ in the second term will only increase the value (see the definition in (18)). Thus,
$$\phi(f_2(x))[k_2'] - \phi(f_1(x))[k_1'] \leq \phi(f_2(x))[k_2'] - \phi(f_1(x))[k_2']$$
$$\leq \|\phi(f_2(x)) - \phi(f_1(x))\|,$$

  where the last line uses (20).

- *Case 2a: $k_2' = k_1, k_2 = k_1'$:* Using definition of $k_2$ in (18), we note that
$$\phi(f_2(x))[k_2'] - \phi(f_1(x))[k_1'] = \phi(f_2(x))[k_2'] - \phi(f_1(x))[k_2]$$
$$\leq \phi(f_2(x))[k_2] - \phi(f_1(x))[k_2]$$
$$\leq \|\phi(f_2(x)) - \phi(f_1(x))\|,$$

  where the last line uses (20).

- *Case 2b:* $k'_2 = k_1, k_2 \neq k'_1$: Using the fact that $k'_2 = k_1$ and that $k_2 \neq k'_2$, we get that $k_2 \neq k_1$. Thus using the definition of $k'_1$ along with the fact that $k_2 \neq k_1$, we get that

$$\phi(f_2(x))[k'_2] - \phi(f_1(x))[k'_1] \leq \phi(f_2(x))[k'_2] - \phi(f_1(x))[k_2]$$
$$\leq \phi(f_2(x))[k_2] - \phi(f_1(x))[k_2]$$
$$\leq \|\phi(f_2(x)) - \phi(f_1(x))\|,$$

where the second last line uses definition of $k_2$ and the last line uses (20).

Combining all the above bounds together implies that

$$\texttt{Margin}(f_1(x)) - \texttt{Margin}(f_2(x)) \leq 2\|\phi(f_2(x)) - \phi(f_1(x))\|.$$

The final statement follows since $\phi$ is $\gamma$-Lipschitz. □

### E.3.2 Regret Bound

For the ease of notation, for the rest of the proof in this section we define the function $\pi^\star$ such that

$$\pi^\star(x) = \underset{k}{\operatorname{argmax}}\, \phi(f^\star(x))[k].$$

Additionally, we recall that for any time $t$, $\widehat{y}_t = \texttt{SelectAction}(f_t(x_t)) = \operatorname{argmax}_k \phi(f_t(x))[k]$. Starting from the definition of the regret, we have

$$\operatorname{Reg}_T = \sum_{t=1}^T \operatorname{Pr}(\widehat{y}_t \neq y_t) - \operatorname{Pr}(\pi^\star(x_t) \neq y_t)$$

$$= \sum_{t=1}^T \mathbf{1}\{\widehat{y}_t \neq \pi^\star(x_t)\} \cdot |\operatorname{Pr}(y_t = \pi^\star(x_t)) - \operatorname{Pr}(y_t = \widehat{y}_t)|$$

$$= \sum_{t=1}^T \mathbf{1}\{\widehat{y}_t \neq \pi^\star(x_t)\} \cdot |\phi(f^\star(x_t))[\pi^\star(x_t)] - \phi(f^\star(x_t))[\widehat{y}_t]|$$

$$\leq \sum_{t=1}^T \mathbf{1}\{\widehat{y}_t \neq \pi^\star(x_t)\} \cdot \operatorname{Gap}(f^\star(x_t), \widehat{y}_t),$$

where the second last line uses the probabilistic model from which labels are generated, and the last inequality plugs in the definition of Gap from (45). Let $\varepsilon > 0$ be a free parameter. We can decompose the above regret bound further as:

$$\operatorname{Reg}_T \leq \sum_{t=1}^T \mathbf{1}\{\widehat{y}_t \neq \pi^\star(x_t), \operatorname{Gap}(f^\star(x_t), \widehat{y}_t) \leq \varepsilon\} \cdot \operatorname{Gap}(f^\star(x_t), \widehat{y}_t)$$

$$+ \sum_{t=1}^T \mathbf{1}\{\widehat{y}_t \neq \pi^\star(x_t), \operatorname{Gap}(f^\star(x_t), \widehat{y}_t) > \varepsilon\} \cdot \operatorname{Gap}(f^\star(x_t), \widehat{y}_t)$$

Using the fact that $y_t(x_t) = \operatorname{argmax}_{k \in [K]} \phi(f_t(x))[k]$ along with the definition of Gap and Lemma 12, we get that

$$\operatorname{Reg}_T \leq \sum_{t=1}^T \mathbf{1}\{\widehat{y}_t \neq \pi^\star(x_t), \operatorname{Gap}(f^\star(x_t), \widehat{y}_t) \leq \varepsilon\} \cdot \varepsilon$$

$$+ 2\gamma \sum_{t=1}^T \mathbf{1}\{\widehat{y}_t \neq \pi^\star(x_t), \operatorname{Gap}(f^\star(x_t), \widehat{y}_t) > \varepsilon\} \cdot \|f^\star(x_t) - f_t(x_t)\|$$

$$\leq \sum_{t=1}^T \mathbf{1}\{\texttt{Margin}(f^\star(x_t)) \leq \varepsilon\} \cdot \varepsilon$$

$$+ 2\gamma \sum_{t=1}^T \mathbf{1}\{\widehat{y}_t \neq \pi^\star(x_t), \operatorname{Gap}(f^\star(x_t), \widehat{y}_t) > \varepsilon\} \cdot \|f^\star(x_t) - f_t(x_t)\|$$

$$\leq \sum_{t=1}^T \mathbf{1}\{\texttt{Margin}(f^\star(x_t)) \leq \varepsilon\} \cdot \varepsilon$$

$$+ 2\gamma \sum_{t=1}^{T} Z_t \mathbf{1}\{\widehat{y}_t \neq \pi^\star(x_t), \operatorname{Gap}(f^\star(x_t), \widehat{y}_t) > \varepsilon\} \cdot \|f^\star(x_t) - f_t(x_t)\|$$

$$+ 2\gamma \sum_{t=1}^{T} \bar{Z}_t \mathbf{1}\{\widehat{y}_t \neq \pi^\star(x_t)\} \cdot \|f^\star(x_t) - f_t(x_t)\| \tag{21}$$

$$= T_\varepsilon \cdot \varepsilon + 2\gamma \cdot \mathtt{T}_A + 2\gamma \cdot \mathtt{T}_B \cdot \|f^\star(x_t) - f_t(x_t)\|,$$

where the second inequality holds because $\operatorname{Gap}(f^\star(x_t), \widehat{y}_t) \leq \varepsilon$ implies that $\mathtt{Margin}(f^\star(x_t)) \leq \varepsilon$ whenever $\widehat{y}_t \neq \pi^\star(x_t)$. In the last line above, we plugged in the definition of $T_\varepsilon$, and defined $\mathtt{T}_A$ and $\mathtt{T}_B$ as the second term and the last term respectively (upto constants). We bound them separately below:

- *Bound on $\mathtt{T}_A$:* We note that

$$\mathtt{T}_A = \sum_{t=1}^{T} Z_t \mathbf{1}\{\widehat{y}_t \neq \pi^\star(x_t), \operatorname{Gap}(f^\star(x_t), \widehat{y}_t) > \varepsilon\} \cdot \|f^\star(x_t) - f_t(x_t)\|$$

$$\leq \sum_{t=1}^{T} Z_t \mathbf{1}\{\|f^\star(x_t) - f_t(x_t)\| > \varepsilon/2\gamma\} \cdot \|f^\star(x_t) - f_t(x_t)\|$$

  where the second line follows from Lemma 12 and because $\widehat{y}_t \neq \pi^\star(x_t)$. Using the fact that $\mathbf{1}\{a \geq b\} \leq a/b$ for all $a, b \geq 0$, we get that

$$\mathtt{T}_A \leq 4\gamma \sum_{t=1}^{T} Z_t \frac{\|f^\star(x_t) - f_t(x_t)\|^2}{\varepsilon}. \tag{22}$$

- *Bound on $\mathtt{T}_B$:* Fix any $t \leq T$, and note that Lemma 10 implies that $\sum_{s=1}^{t} \|f^\star(x_t) - f_t(x_t)\|^2 \leq \Psi_\delta^{\ell_\phi}(\mathcal{F}, T)$. Thus $f^\star$ satisfies the constraint in the definition of $\Delta_t$ in (5) and we must have that

$$\|f^\star(x_t) - f_t(x_t)\| \leq \Delta_t(x_t). \tag{23}$$

  Plugging in the definition of $Z_t$, we note that

$$\mathtt{T}_B = \sum_{t=1}^{T} \mathbf{1}\{\mathtt{Margin}(f_t(x_t)) > 2\gamma\Delta_t(x_t), \widehat{y}_t \neq \pi^\star(x_t)\}$$

$$\leq \sum_{t=1}^{T} \mathbf{1}\{\|f_t(x_t) - f^\star(x_t)\| > \Delta_t(x_t)\},$$

  where the second inequality is due Lemma 12. However, note that the term inside the indicator contradicts (23) (which always holds). Thus,

$$\mathtt{T}_B = 0. \tag{24}$$

Combining the bounds (22) and (24), we get that

$$\operatorname{Reg}_T \leq \varepsilon T_\varepsilon + 8\gamma^2 \sum_{t=1}^{T} Z_t \frac{\|f_t(x_t) - f^\star(x_t)\|^2}{\varepsilon}$$

$$\leq \varepsilon T_\varepsilon + \frac{8\gamma^2}{\varepsilon} \Psi_\delta^{\ell_\phi}(\mathcal{F}, T),$$

where the last inequality is due to Lemma 10.

Since $\varepsilon$ is a free parameter above, the final bound follows by choosing the best parameter $\varepsilon$, and by plugging in the form of $\Psi_\delta^{\ell_\phi}(\mathcal{F}, T)$.

### E.3.3  Total Number of Queries

We use the notation $N_T$ to denote the total number of expert queries made by the learner within $T$ rounds of interactions. Let $\varepsilon > 0$ be a free parameter. Using the definition of $Z_t$, we have that

$$N_T = \sum_{t=1}^{T} Z_t$$

$$= \sum_{t=1}^{T} \mathbf{1}\{\texttt{Margin}(f_t(x_t)) \le 2\gamma\Delta_t(x_t)\}$$

$$= \sum_{t=1}^{T} \mathbf{1}\{\texttt{Margin}(f_t(x_t)) \le 2\gamma\Delta_t(x_t), \texttt{Margin}(f^\star(x_t)) \le \varepsilon\}$$

$$\qquad\qquad + \sum_{t=1}^{T} \mathbf{1}\{\texttt{Margin}(f_t(x_t)) \le 2\gamma\Delta_t(x_t), \texttt{Margin}(f^\star(x_t)) > \varepsilon\}$$

$$\le \sum_{t=1}^{T} \mathbf{1}\{\texttt{Margin}(f^\star(x_t)) \le \varepsilon\}$$

$$\qquad\qquad + \sum_{t=1}^{T} \mathbf{1}\{\texttt{Margin}(f_t(x_t)) \le 2\gamma\Delta_t(x_t), \texttt{Margin}(f^\star(x_t)) > \varepsilon, \Delta_t(x_t) \le \varepsilon/4\gamma\}$$

$$\qquad\qquad + \sum_{t=1}^{T} \mathbf{1}\{\texttt{Margin}(f_t(x_t)) \le 2\gamma\Delta_t(x_t), \texttt{Margin}(f^\star(x_t)) > \varepsilon, \Delta_t(x_t) > \varepsilon/4\gamma\}$$

$$\tag{25}$$

$$= T_\varepsilon + \texttt{T}_D + \texttt{T}_E,$$

where in the last line we use the definition of $T_\varepsilon$, and defined $\texttt{T}_D$ and $\texttt{T}_E$ respectively. We bound them separately below:

- *Bound on $\texttt{T}_D$.* Recall (23) which implies that $f^\star$ satisfies the bound $\|f_t(x_t) - f^\star(x_t)\| \le \Delta_t(x_t)$. Thus, using Lemma 13, we get that

$$\texttt{Margin}(f^\star(x_t)) \le 2\gamma\|f_t(x_t) - f^\star(x_t)\| + t\texttt{Margin}(f_t(x_t)) \le 2\gamma\Delta_t(x_t) + \texttt{Margin}(f_t(x_t)).$$

  The above implies that

$$\texttt{T}_D = \sum_{t=1}^{T} \mathbf{1}\{\texttt{Margin}(f_t(x_t)) \le 2\gamma\Delta_t(x_t), \texttt{Margin}(f^\star(x_t)) > \varepsilon, \Delta_t(x_t) \le \varepsilon/4\gamma\}$$

$$\le \sum_{t=1}^{T} \mathbf{1}\{\texttt{Margin}(f^\star(x_t)) \le 4\gamma\Delta_t(x_t), \texttt{Margin}(f^\star(x_t)) > \varepsilon, \Delta_t(x_t) \le \varepsilon/4\gamma\}$$

$$\le 0,$$

  where the last line follows from the fact that all the conditions inside the indictor can not hold simultaneously.

- *Bound on $\texttt{T}_E$.* We note that

$$\texttt{T}_E = \sum_{t=1}^{T} \mathbf{1}\{\texttt{Margin}(f_t(x_t)) \le 2\gamma\Delta_t(x_t), \texttt{Margin}(f^\star(x_t)) > \varepsilon, \Delta_t(x_t) > \varepsilon/4\gamma\}$$

$$\le \sum_{t=1}^{T} Z_t \mathbf{1}\{\Delta_t(x_t) \ge \varepsilon/4\gamma\}$$

$$\le \frac{320\Psi_\delta^{\ell_\phi}(\mathcal{F}, T)}{\varepsilon^2} \cdot \mathfrak{E}(\mathcal{F}, \varepsilon/4\gamma; f^\star).$$

  where the last line follows from setting $\zeta = \varepsilon/4\gamma$ in Lemma 11.

Gathering the bounds above, we get that

$$N_T \le T_\varepsilon + \frac{640\gamma^2\Psi_\delta^{\ell_\phi}(\mathcal{F}, T)}{\varepsilon^2} \cdot \mathfrak{E}(\mathcal{F}, \varepsilon/4\gamma; f^\star).$$

Since $\varepsilon$ is a free parameter above, the final bound follows by choosing the best parameter $\varepsilon$, and by plugging in the form of $\Psi_\delta^{\ell_\phi}(\mathcal{F}, T)$.

### E.4 Proof of Theorem 2

Before delving into the proof, we recall the relevant notation. In Algorithm 3,

**Algorithm 3** Selective Sampling with Expert Feedback for Stochastic Contexts

**Input:** Parameters $\delta, \gamma, \lambda, T$, function class $\mathcal{F}$, and online regression oracle Oracle w.r.t $\ell_\phi$.

1: Set $\Psi_\delta^{\ell_\phi}(\mathcal{F},T) = \frac{4}{\lambda}\mathrm{Reg}^{\ell_\phi}(\mathcal{F};T) + \frac{112}{\lambda^2}\log(4\log^2(T)/\delta)$, Compute $f_1 \leftarrow \mathrm{Oracle}_1(\varnothing)$.
2: Set $E = \lceil\log(T)\rceil$ and $\tau_e = 2^{e-1}$ for $e \le E$.
3: **for** $e = 1,\ldots,E-1$ **do**
4:     Learner constructs the feasible set of optimal functions $\mathcal{F}_e$ as

$$\mathcal{F}_e = \left\{ f \in \mathcal{F} \mid \sum_{s=1}^{\tau_e-1} Z_s\|f(x_s) - f_s(x_s)\|^2 \le \Psi_\delta^{\ell_\phi}(\mathcal{F},T) \right\}. \tag{26}$$

5:     **for** $t \leftarrow \tau_e$ to $\tau_{e+1}-1$ **do**
6:         Nature samples $x_t$ from an (unknown) distribution $\mu$.
7:         Learner computes

$$g_t \in \underset{g \in \mathcal{F}_e}{\arg\min}\ \mathrm{Margin}(g(x_t)), \quad \text{and,} \quad \Delta_e(x_t) \coloneqq \max_{f,f' \in \mathcal{F}_e}\|f(x_t) - f'(x_t)\|. \tag{27}$$

8:         Learner decides whether to query: $Z_t = \mathbf{1}\{\mathrm{Margin}(g_t(x_t)) \le 2\gamma\Delta_e(x_t)\}$.
9:         **if** $Z_t = 1$ **then**
10:             Learner plays the action $\widehat{y}_t = \mathrm{SelectAction}(f_t(x_t))$.
11:             Learner queries the label $y_t$ on $x_t$.
12:             $f_{t+1} \leftarrow \mathrm{Oracle}_t(\{x_t, y_t\})$.
13:         **else**
14:             Learner plays the action $\widehat{y}_t = \mathrm{SelectAction}(g_t(x_t))$.
15:             $f_{t+1} \leftarrow f_t$.

---

- The label $y_t \sim \phi(f^\star(x_t))$, where $\phi$ denotes the link-function given in (2).

- The function $\mathrm{SelectAction}(f_t(x_t)) \coloneqq \arg\max_k \phi(f_t(x_t))[k]$.

- For any vector $v \in \mathbb{R}^K$, the margin is given by the gap between the value at the largest and the second largest coordinate, i.e.

$$\mathrm{Margin}(v) = \phi(v)[k^\star] - \max_{k \ne k^\star}\phi(v)[k],$$

where $k^\star \in \arg\max_{k \in [K]}\phi(v)[k]$.

- We also define $T_\varepsilon = \sum_{t=1}^T \mathbf{1}\{\mathrm{Margin}(f^\star(x_t)) \le \varepsilon\}$ to denote the number of samples within $T$ rounds of interaction for which the margin w.r.t. $f^\star$ is smaller than $\varepsilon$.

- We define the function $\mathrm{Gap} : \mathbb{R}^K \times [K] \mapsto \mathbb{R}^+$ as

$$\mathrm{Gap}(v,k) = \max_{k'}\phi(v)[k'] - \phi(v)[k], \tag{28}$$

to denote the gap between the largest and the $k$-th coordinate of $v$. Recall that Lemma 9 holds.

- Additionally, we define the function $Z^e$ to denote the query condition

$$Z^e(x) = \mathbf{1}\{\inf_{g \in \mathcal{F}_e}\mathrm{Margin}(g(x)) \le 2\gamma \sup_{f,f' \in \mathcal{F}_e}\|f(x) - f'(x)\|\}. \tag{29}$$

The definition in (29) suggests that for all $t \in [\tau_e, \tau_{e+1})$, $Z_t = Z^e(x_t)$, for all $e \le E-1$.

**Intuition for epoching.** We next provide intuition on why epoching in needed in Algorithm 3 to get the improved query complexity bound. From the proof sketch in Section E.2 , the term $\sum_{t=1}^T Z_t\mathbf{1}\{\Delta_t(x_t) \ge \varepsilon\}$ appearing in the query complexity bound is handled using the eluder dimension of $\mathcal{F}$. When $x_t$ is sampled i.i.d. we wish to bound this using disagreement-coefficient instead. However, note that in Algorithm 1 the query condition $Z_t$ depends on the samples $\{x_s\}_{s<t}$ drawn in all previous time steps and the corresponding query conditions $\{Z_s\}_{s<t}$. This introduces a bias, and thus the terms $Z_t\mathbf{1}\{\Delta_t(x_t) \ge \varepsilon\}$ are no longer independent to each other. Thus, we can not directly used distributional properties like the disagreement coefficient to bound the query complexity. Algorithm 3 fixes this issue by defining epochs of doubling length such that the query condition in

epoch $e$ only depends on the samples presented to the learner at time steps before this epoch (i.e. in time steps $1 \leq t \leq \tau_e - 1$. Thus, the terms $Z_t \mathbf{1}\{\Delta_t(x_t) \geq \varepsilon\}$ for $\tau_e \leq t < \tau_{e+1}$ are i.i.d. allowing us to get bounds in terms of distributional properties like the disagreement coefficient of $\mathcal{F}$.

However, note that whenever we query in Algorithm 3, we still choose the labels according to the estimate from the online regression oracle and thus the regret bound remains unchanged.

### E.4.1 Supporting Technical Results

The following lemma establishes useful technical properties of the function $f^\star$ and the sets $\mathcal{F}_e$.

**Lemma 14.** *Suppose Algorithm 3 is run on the sequence $\{x_t\}_{t \leq T}$ drawn i.i.d. from the unknown distribution $\mu$. Then, with probability at least $1 - \delta$, each of the following holds:*

$(a)$ *For all $t \leq T$, the function $f^\star \in \mathcal{F}$ satisfies*

$$\sum_{s=1}^{t} Z_s \|f^\star(x_s) - f_s(x_s)\|^2 \leq \Psi_\delta^{\ell_\phi}(\mathcal{F}, T),$$

*where $\Psi_\delta^{\ell_\phi}(\mathcal{F}, T) = \frac{4}{\lambda} \mathrm{Reg}^{\ell_\phi}(\mathcal{F}; T) + \frac{112}{\lambda^2} \log(4 \log^2(T)/\delta)$.*

*Thus, $f^\star \in \mathcal{F}_e$ for all $e \leq E - 1$, and $\|f^\star(x_t) - g_t(x_t)\| \leq \Delta_e(x_t)$ for all $\tau_e \leq t \leq \tau_{e+1} - 1$.*

$(b)$ *For any function $f \in \mathcal{F}_e$, we have*

$$\mathbb{E}\left[ \sum_{s=1}^{\tau_e - 1} Z_s \|f(x_s) - f^\star(x_s)\|^2 \right] \leq \widehat{\Psi}_\delta^{\ell_\phi}(\mathcal{F}; T),$$

*where $\widehat{\Psi}_\delta^{\ell_\phi}(\mathcal{F}; T) := 2\Psi_\delta^{\ell_\phi}(\mathcal{F}, T) + 4c_2\big(\log^4(T) \sup_{\tau \leq T} \big(\tau \mathrm{Rad}_\tau^2(\mathcal{F})\big) + 2\log(T)\log(E/\delta)\big)$.*

$(c)$ *For any $e \leq E$, and any function $f \in \mathcal{F}_e$, we have*

$$\|f(x_s) - f^\star(x_s)\|_{\bar{\nu}_e} \leq \sqrt{\frac{\widehat{\Psi}_\delta^{\ell_\phi}(\mathcal{F}; T)}{\tau_e - 1}}$$

*where the sub-distributions $\bar{\mu}_{\bar{e}}(x) := Z^{\bar{e}}(x)\mu(x)$ and $\bar{\nu}_e := \frac{1}{\tau_e - \tau_1} \sum_{\bar{e}=1}^{e-1} (\tau_{\bar{e}+1} - \tau_{\bar{e}})\bar{\mu}_{\bar{e}}$.*

$(d)$ *For any $\bar{e} < e$, the corresponding sets $\mathcal{F}_e$ and $\mathcal{F}_{\bar{e}}$ satisfy the relation $\mathcal{F}_e \subseteq \mathcal{F}_{\bar{e}}$.*

$(e)$ *For any $\bar{e} \leq e$, we have $\bar{\mu}_e \lesssim \bar{\mu}_{\bar{e}}$.*

*Proof.* We prove each part separately below:

$(a)$ An application of Lemma 5, where we note that we do not query oracle when $Z_s = 0$, and thus do not count the time steps for which $Z_s = 0$, implies that

$$\sum_{s=1}^{t} Z_s \|f^\star(x_s) - f_s(x_s)\|^2 \leq \frac{4}{\lambda} \mathrm{Reg}^{\ell_\phi}(\mathcal{F}; T) + \frac{112}{\lambda^2} \log(4\log^2(T)/\delta) =: \Psi_\delta^{\ell_\phi}(\mathcal{F}, T)$$

for all $t \leq T$ with probability at least $1 - \delta$. Using the above for $t = \tau_{e+1} - 1$ implies that $f^\star \in \mathcal{F}_e$ for all $e \leq E - 1$. Since, we also have that $g_t \in \mathcal{F}_e$ (by construction) for all $\tau_e \leq t \leq \tau_{e+1} - 1$, plugging in the definition of $\Delta_e(x_t)$, we immediately get that $\|f^\star(x_t) - g_t(x_t)\| \leq \Delta_e(x_t)$.

$(b)$ Fix any epoch number $\bar{e} \leq E - 1$, and consider the time steps $\tau_{\bar{e}} \leq t < \tau_{\bar{e}+1}$. Define the loss function

$$\ell_{\bar{e}}(f(x), f^\star(x)) = Z^{\bar{e}}(x)\|f(x) - f^\star(x)\|^2$$

where $Z^{\bar{e}}$ denotes the query conditions at epoch $\bar{e}$ (defined in (29)), and recall that $Z^{\bar{e}}$ does not depend on any samples that are drawn at epoch $\bar{e}$ (by definition). Furthermore, note that

$\ell_{\bar{e}}$ is 2-smooth w.r.t. $f$ and satisfies $\ell_{\bar{e}}(f(x), f^\star(x)) \le 4$ for all $f, f^\star \in \mathcal{F}$ and $x$. Thus using Lemma 2, we get that for any $f \in \mathcal{F}_e$, with probability at least $1 - \delta/E$,

$$\mathbb{E}\left[\sum_{s=\tau_{\bar{e}}}^{\tau_{\bar{e}+1}-1} Z^{\bar{e}}(x_s)\|f(x_s) - f^\star(x_s)\|^2\right]$$

$$\le 2\sum_{s=\tau_{\bar{e}}}^{\tau_{\bar{e}+1}-1} Z^{\bar{e}}(x_s)\|f(x_s) - f^\star(x_s)\|^2 + c_2\big(2\tau_e \log^3(\tau_e)\mathsf{Rad}^2_{\tau_e}(\mathcal{F}_e) + 4\log(E/\delta)\big)$$

$$\le 2\sum_{s=\tau_{\bar{e}}}^{\tau_{\bar{e}+1}-1} Z^{\bar{e}}(x_s)\|f(x_s) - f^\star(x_s)\|^2 + 2c_2\bigg(\log^3(T)\sup_{\tau \le T}\big(\tau\mathsf{Rad}^2_\tau(\mathcal{F})\big) + 2\log(E/\delta)\bigg),$$

where in the last line we used the fact that $\mathcal{F}_e \subseteq \mathcal{F}$. Summing the above for all $\bar{e} \le e - 1$, we get that for any $f \in \mathcal{F}_e$,

$$\mathbb{E}\left[\sum_{t=1}^{\tau_e-1} Z^e(x_s)\|f(x_s) - f^\star(x_s)\|^2\right]$$

$$\le 2\sum_{t=1}^{\tau_e-1} Z^e(x_s)\|f(x_s) - f^\star(x_s)\|^2 + 4c_2 E\bigg(\log^3(T)\sup_{\tau \le T}\big(\tau\mathsf{Rad}^2_\tau(\mathcal{F})\big) + 2\log(E/\delta)\bigg)$$

$$\le 2\Psi^{\ell_\phi}_\delta(\mathcal{F}, T) + 4c_2\bigg(\log^4(T)\sup_{\tau \le T}\big(\tau\mathsf{Rad}^2_\tau(\mathcal{F})\big) + 2\log(T)\log(E/\delta)\bigg),$$

where the last line follows by using the definition of $\mathcal{F}_e$, and that $E \le \lceil\log(T)\rceil$, and that $\mathcal{F}_e \subseteq \mathcal{F}$.

(c) Starting from part-(b), we first note that

$$\mathbb{E}\left[\sum_{s=1}^{\tau_e-1} Z_s\|f(x_s) - f^\star(x_s)\|^2\right] \le \widehat{\Psi}^{\ell_\phi}_\delta(\mathcal{F}; T). \tag{30}$$

Additionally, also note that

$$\mathbb{E}\left[\sum_{s=1}^{\tau_e-1} Z_s\|f(x_s) - f^\star(x_s)\|^2\right] = \mathbb{E}\left[\sum_{\bar{e}=1}^{e-1}\sum_{s=\tau_{\bar{e}}}^{\tau_{\bar{e}+1}-1} Z_s\|f(x_s) - f^\star(x_s)\|^2\right]$$

$$= \mathbb{E}\left[\sum_{\bar{e}=1}^{e-1}\sum_{s=\tau_{\bar{e}}}^{\tau_{\bar{e}+1}-1} Z^{\bar{e}}(x_s)\|f(x_s) - f^\star(x_s)\|^2\right]$$

$$= \sum_{\bar{e}=1}^{e-1}\sum_{s=\tau_{\bar{e}}}^{\tau_{\bar{e}+1}-1} \mathbb{E}_{x_s\sim\mu}\Big[Z^{\bar{e}}(x_s)\|f(x_s) - f^\star(x_s)\|^2\Big]$$

$$= \sum_{\bar{e}=1}^{e-1}(\tau_{\bar{e}+1} - \tau_{\bar{e}})\,\mathbb{E}_{x\sim\mu}\Big[Z^{\bar{e}}(x)\|f(x) - f^\star(x)\|^2\Big],$$

where the last two lines use the fact that the query condition $Z^{\bar{e}}$ does not depend on the samples from rounds $t = \tau_{\bar{e}}$ to $\tau_{\bar{e}+1} - 1$. Plugging in the definition of $\bar{\mu}_{\bar{e}}$ in the above, we get that

$$\mathbb{E}\left[\sum_{s=1}^{\tau_e-1} Z_s\|f(x_s) - f^\star(x_s)\|^2\right] = \sum_{\bar{e}=1}^{e-1}(\tau_{\bar{e}+1} - \tau_{\bar{e}}) \cdot \mathbb{E}_{x\sim\bar{\mu}_{\bar{e}}}\Big[\|f(x_s) - f^\star(x_s)\|^2\Big]$$

$$= (\tau_e - \tau_1) \cdot \mathbb{E}_{x\sim\bar{\nu}_e}\Big[\|f(x_s) - f^\star(x_s)\|^2\Big]$$

$$= (\tau_e - \tau_1) \cdot \|f(x_s) - f^\star(x_s)\|^2_{\bar{\nu}_e}, \tag{31}$$

where in the second line, we used the fact that the sub-distribution $\bar{\nu}_e := \frac{1}{\tau_e-\tau_1}\sum_{\bar{e}=1}^{e-1}(\tau_{\bar{e}+1} - \tau_{\bar{e}})\bar{\mu}_{\bar{e}}$.

Combining (30) and (31), we get that

$$\|f(x_s) - f^\star(x_s)\|_{\bar{\nu}_e} \le \sqrt{\frac{\widehat{\Psi}^{\ell_\phi}_\delta(\mathcal{F}; T)}{\tau_e - 1}}$$

($d$) The argument follows from the definition of the set $\mathcal{F}_e$ as any function $f \in \mathcal{F}_e$ that satisfies

$$\sum_{s=1}^{\tau_e-1} Z_s \|f(x_s) - f_s(x_s)\|^2 \le \Psi_\delta^{\ell_\phi}(\mathcal{F}, T),$$

also satisfies the constraint

$$\sum_{s=1}^{\tau_{\bar{e}}-1} Z_s \|f(x_s) - f_s(x_s)\|^2 \le \Psi_\delta^{\ell_\phi}(\mathcal{F}, T),$$

for any $\bar{e} \le e$, since the left hand side consists of lesser number of terms and all terms are non-negative. Thus, $\mathcal{F}_e \subseteq \mathcal{F}_{\bar{e}}$.

($e$) Recall that for any $e \le E - 1$, the sub-probability measure $\bar{\mu}_e(x) := Z^e(x)\mu(x)$ where $Z^e(x) = \mathbf{1}\{\min_{g \in \mathcal{F}_e} \|g(x)\| \le \Delta_e(x)\}$, and $\Delta_e(x) = \max_{f',f \in \mathcal{F}_e} \|f(x) - f'(x)\|$. First note that for any $\bar{e} \le e$,

$$\Delta_e(x) = \max_{f',f \in \mathcal{F}_e} \|f(x) - f'(x)\| \le \max_{f',f \in \mathcal{F}_{\bar{e}}} \|f(x) - f'(x)\| = \Delta_{\bar{e}}(x),$$

where the inequality above holds because $\mathcal{F}_e \subseteq \mathcal{F}_{\bar{e}}$ due to part-(d) above. Furthermore,

$$\min_{g \in \mathcal{F}_e} \|g(x)\| \ge \min_{g \in \mathcal{F}_{\bar{e}}} \|g(x)\|,$$

again because $\mathcal{F}_e \subseteq \mathcal{F}_{\bar{e}}$. Thus,

$$Z^e(x) = \mathbf{1}\{\min_{g \in \mathcal{F}_e} \|g(x)\| \le \Delta_e(x)\} \le \mathbf{1}\{\min_{g \in \mathcal{F}_{\bar{e}}} \|g(x)\| \le \Delta_{\bar{e}}(x)\} \le Z^{\bar{e}}(x).$$

The above implies that $\bar{\mu}_e \le \bar{\mu}_{\bar{e}}$.

$\square$

**Lemma 15.** *Let $\varepsilon_0, \gamma_0 \ge 0$, and $f^\star \in \mathcal{F}$. Then, for any sub distribution $\bar{\mu}$ such that $\mathbb{E}_{x \sim \bar{\mu}}[\mathbf{1}\{x \in \mathcal{X}\}] > 0$, $\varepsilon \ge \varepsilon_0$ and $\gamma \ge \gamma_0$,*

$$\frac{\varepsilon^2}{\gamma^2} \Pr_{x \sim \bar{\mu}}\big(\exists f \in \mathcal{F} \,:\, \|f(x) - f^\star(x)\| > \varepsilon, \|f(x_s) - f^\star(x_s)\|_{\bar{\mu}} \le \gamma\big) \le \theta^{\mathrm{val}}(\mathcal{F}, \varepsilon_0, \gamma_0; f^\star).$$

*Proof.* The key idea in the proof is to go from sub distributions to distributions, and then invoking the definition of $\theta$ from Definition 2. Define $\kappa = \mathbb{E}_{x \sim \bar{\mu}}[\mathbf{1}\{x \in \mathcal{X}\}]$. Since $0 < \kappa \le 1$, we can define a probability measure $\mu$ such that $\mu(x) = \bar{\mu}(x)/\kappa$. Thus, for any $\varepsilon \ge \varepsilon_0$ and $\gamma \ge \gamma_0$,

$$\frac{\varepsilon^2}{\gamma^2} \Pr_{x \sim \bar{\mu}}\big(\exists f \in \mathcal{F} \,:\, \|f(x) - f^\star(x)\| > \varepsilon, \|f(x_s) - f^\star(x_s)\|_{\bar{\mu}} \le \gamma\big)$$

$$= \frac{\varepsilon^2}{\gamma^2/k} \Pr_{x \sim \mu}\big(\exists f \in \mathcal{F} \,:\, \|f(x) - f^\star(x)\| > \varepsilon, \|f(x_s) - f^\star(x_s)\|_{\bar{\mu}} \le \gamma\big)$$

$$\le \frac{\varepsilon^2}{\gamma^2/k} \Pr_{x \sim \mu}\big(\exists f \in \mathcal{F} \,:\, \|f(x) - f^\star(x)\| > \varepsilon, \|f(x_s) - f^\star(x_s)\|_{\mu} \le \gamma/\sqrt{k}\big)$$

$$= \frac{\varepsilon^2}{\bar{\gamma}^2} \Pr_{x \sim \mu}\big(\exists f \in \mathcal{F} \,:\, \|f(x) - f^\star(x)\| > \varepsilon, \|f(x_s) - f^\star(x_s)\|_{\mu} \le \bar{\gamma}\big)$$

$$\le \sup_{\varepsilon \ge \varepsilon_0, \bar{\gamma} \ge \gamma_0} \frac{\varepsilon^2}{\bar{\gamma}^2} \Pr_{x \sim \mu}\big(\exists f \in \mathcal{F} \,:\, \|f(x) - f^\star(x)\| > \varepsilon, \|f(x_s) - f^\star(x_s)\|_{\mu} \le \bar{\gamma}\big),$$

where in the second last line, we defined $\bar{\gamma} = \gamma/\sqrt{k}$, and the last line used the fact that both $\varepsilon \ge \varepsilon_0$ and $\bar{\gamma} \ge \gamma_0$. The final statement follows by noting the fact that $\mu$ is a distribution and the definition of the disagreement coefficient $\theta^{\mathrm{val}}(;)$ from Definition 2. $\square$

The following technical result will be useful in bounding the query complexity for Algorithm 3.

**Lemma 16.** *For any $t \leq T$, let $e(t)$ denotes the epoch number such that $\tau_{e(t)} \leq t < \tau_{e(t)+1}$. Let $f^\star \in \mathcal{F}$ satisfy Lemma 14, and let $\Delta_{e(t)}(x_t)$ be defined in Algorithm 3. Then, for any $\zeta > 0$, with probability at least $1 - \delta$,*

$$\sum_{t=1}^{T} Z_t \mathbf{1}\{\Delta_{e(t)}(x_t) \geq \zeta\} \leq 12 \log(T) \cdot \frac{\widehat{\Psi}_{\delta}^{\ell_\phi}(\mathcal{F};T)}{\zeta^2} \cdot \theta^{\mathrm{val}}\left(\mathcal{F}, \frac{\zeta}{2}, \frac{\widehat{\Psi}_{\delta}^{\ell_\phi}(\mathcal{F};T)}{T}; f^\star\right) + 4\log(2/\delta).$$

*Proof.* Recall the definition of the query rule $Z^e$ given in (29), and note that the function $Z^e$ is independent of the samples $\{x_t\}_{t=\tau_e}^{\tau_{e+1}-1}$ chosen by the nature for time steps at epoch $e$. Additionally, also recall that at every time step, $x_t$ is sampled independently from the distribution $\mu$. Thus, using the query condition $Z^e$, we can define the sub-probability measure

$$\bar{\mu}_e := \mu(x) Z^e(x), \tag{32}$$

such that $\bar{\mu}_e(x) = \mu(x)$ whenever $Z_e(x) = 1$ and is $0$ otherwise. Furthermore, for any $e \in E - 1$, we define the sub-probability measure $\bar{\nu}_e$ as

$$\bar{\nu}_e = \frac{1}{\tau_e - \tau_1} \sum_{\bar{e}=1}^{e-1} (\tau_{\bar{e}+1} - \tau_{\bar{e}}) \bar{\mu}_{\bar{e}}. \tag{33}$$

We now move to the main proof. First fix any epoch $e \leq E-1$, and consider any round $t \in [\tau_e, \tau_{e+1}-1]$. Using the definition of $\Delta_e(x_t)$ and definition of $Z^e$ from (29) in the above, we get that

$$\mathbb{E}_{x_t \sim \mu}[Z_t \mathbf{1}\{\Delta_e(x_t) > \zeta\}] = \mathbb{E}_{x \sim \mu}\left[Z^e(x_t) \mathbf{1}\left\{\sup_{f,f' \in \mathcal{F}_e} \|f(x_t) - f'(x_t)\| > \zeta\right\}\right]$$

$$\leq \mathbb{E}_{x \sim \mu}\left[Z^e(x_t) \mathbf{1}\left\{\sup_{f \in \mathcal{F}_e} \|f(x_t) - f^\star(x_t)\| > \frac{\zeta}{2}\right\}\right]$$

where the second line follows because $f^\star \in \mathcal{F}_e$, and because $\sup_{f,f' \in \mathcal{F}_e} \|f(x_t) - f'(x_t)\| \leq 2\sup_{f \in \mathcal{F}_e} \|f(x_t) - f^\star(x_t)\|$ due to Triangle inequality. Plugging in the definition of $\bar{\mu}_e$ from (32) in the above we get that

$$\mathbb{E}_{x_t \sim \mu}[Z_t \mathbf{1}\{\Delta_e(x_t) > \zeta\}] \leq \mathbb{E}_{x \sim \bar{\mu}_e}\left[\mathbf{1}\left\{\sup_{f \in \mathcal{F}_e} \|f(x_t) - f^\star(x_t)\| > \frac{\zeta}{2}\right\}\right]$$

$$\leq \mathbb{E}_{x \sim \bar{\mu}_{\bar{e}}}\left[\mathbf{1}\left\{\sup_{f \in \mathcal{F}_e} \|f(x_t) - f^\star(x_t)\| > \frac{\zeta}{2}\right\}\right]. \tag{34}$$

for all $\bar{e} \leq e$, where the last inequality follows from Lemma 14-$(e)$. Since the above holds for all $\bar{e} \leq e$, we immediately get that

$$\mathbb{E}_{x_t \sim \mu}[Z_t \mathbf{1}\{\Delta_e(x_t) > \zeta\}] \leq \mathbb{E}_{x \sim \bar{\nu}_{\bar{e}}}\left[\mathbf{1}\left\{\sup_{f \in \mathcal{F}_e} \|f(x_t) - f^\star(x_t)\| > \frac{\zeta}{2}\right\}\right]$$

$$= \mathbb{E}_{x \sim \bar{\nu}_{\bar{e}}}\left[\mathbf{1}\left\{\exists f \in \mathcal{F}_e : \|f(x) - f^\star(x)\| > \frac{\zeta}{2}\right\}\right], \tag{35}$$

where the sub-probability measure $\bar{\nu}_{\bar{e}}$ is defined in (33). Additionally, recall that Lemma 14-$(b)$ implies that with probability at least $1 - \delta$ any $f \in \mathcal{F}_e$ satisfies

$$\|f(x_s) - f^\star(x_s)\|_{\bar{\nu}_e} \leq \sqrt{\frac{\widehat{\Psi}_{\delta}^{\ell_\phi}(\mathcal{F};T)}{\tau_e - 1}}. \tag{36}$$

Conditioning on the above event, and plugging it in (35), we get that

$$\mathbb{E}_{x_t \sim \mu}[Z_t \mathbf{1}\{\Delta_e(x_t) > \zeta\}]$$

$$\leq \mathbb{E}_{x \sim \bar{\nu}_{\bar{e}}}\left[\mathbf{1}\left\{\exists f \in \mathcal{F}_e : \|f(x) - f^\star(x)\| > \frac{\zeta}{2}, \|f(x_s) - f^\star(x_s)\|_{\bar{\nu}_e} \leq \sqrt{\frac{\widehat{\Psi}_{\delta}^{\ell_\phi}(\mathcal{F};T)}{\tau_e - 1}}\right\}\right]$$

$$\leq 4 \cdot \frac{\widehat{\Psi}_\delta^{\ell_\phi}(\mathcal{F};T)}{(\tau_e-1)\zeta^2} \cdot \theta^{\mathrm{val}}\left(\mathcal{F}, \frac{\zeta}{2}, \frac{\widehat{\Psi}_\delta^{\ell_\phi}(\mathcal{F};T)}{\tau_e-1}; f^\star\right), \tag{37}$$

where the last inequality uses Lemma 15.

Summing up the bound in (37) for each term $t = 1$ to $T$, we get that

$$\sum_{t=1}^{T} \mathbb{E}_{x_t}[Z_t \mathbf{1}\{\Delta_e(x_t) > \zeta\}] = \sum_{e=1}^{E-1} \sum_{t=\tau_e}^{\tau_{e+1}-1} \mathbb{E}_{x_t}[Z_t \mathbf{1}\{\Delta_e(x_t) > \zeta\}]$$

$$\leq 4 \sum_{e=1}^{E-1} (\tau_{e+1} - \tau_e) \frac{\widehat{\Psi}_\delta^{\ell_\phi}(\mathcal{F};T)}{(\tau_e-1)\zeta^2} \cdot \theta^{\mathrm{val}}\left(\mathcal{F}, \frac{\zeta}{2}, \frac{\widehat{\Psi}_\delta^{\ell_\phi}(\mathcal{F};T)}{\tau_e-1}; f^\star\right)$$

$$\leq 8 \sum_{e=1}^{E-1} \frac{\widehat{\Psi}_\delta^{\ell_\phi}(\mathcal{F};T)}{\zeta^2} \cdot \theta^{\mathrm{val}}\left(\mathcal{F}, \frac{\zeta}{2}, \frac{\widehat{\Psi}_\delta^{\ell_\phi}(\mathcal{F};T)}{\tau_e-1}; f^\star\right)$$

$$\leq 8 \sum_{e=1}^{E-1} \frac{\widehat{\Psi}_\delta^{\ell_\phi}(\mathcal{F};T)}{\zeta^2} \cdot \theta^{\mathrm{val}}\left(\mathcal{F}, \frac{\zeta}{2}, \frac{\widehat{\Psi}_\delta^{\ell_\phi}(\mathcal{F};T)}{T}; f^\star\right)$$

$$\leq 8 \log(T) \cdot \frac{\widehat{\Psi}_\delta^{\ell_\phi}(\mathcal{F};T)}{\zeta^2} \cdot \theta^{\mathrm{val}}\left(\mathcal{F}, \frac{\zeta}{2}, \frac{\widehat{\Psi}_\delta^{\ell_\phi}(\mathcal{F};T)}{T}; f^\star\right)$$

where the second inequality uses the fact that $\tau_{e+1} = 2\tau_e$ and that $\tau_1 = 1$, the third inequality holds due to monotonicity of $\theta^{\mathrm{val}}\left(\mathcal{F}, \frac{\zeta}{2}, \cdot; f^\star\right)$ and the last line simply plugs in the value of $E = \log(T)$.

Using Lemma 4 with the above bound for the sequence of random variable $X_t = Z_t \mathbf{1}\{\Delta_e(x_t) > \zeta\}$, we get that with probability at least $1 - \delta$,

$$\sum_{t=1}^{T} Z_t \mathbf{1}\{\Delta_e(x_t) > \zeta\} \leq \frac{3}{2} \sum_{t=1}^{T} \mathbb{E}_{x_t}[Z_t \mathbf{1}\{\Delta_e(x_t) > \zeta\}] + 4\log(2/\delta)$$

$$\leq 12 \log(T) \cdot \frac{\widehat{\Psi}_\delta^{\ell_\phi}(\mathcal{F};T)}{\zeta^2} \cdot \theta^{\mathrm{val}}\left(\mathcal{F}, \frac{\zeta}{2}, \frac{\widehat{\Psi}_\delta^{\ell_\phi}(\mathcal{F};T)}{T}; f^\star\right) + 4\log(2/\delta).$$

The final result follows by taking a union bound of the above and the event in (36). $\qquad \square$

### E.4.2 Regret Bound

For the ease of notation, through the proofs in this section we define the operators $\pi^\star$ as

$$\pi^\star(x) = \operatorname*{argmax}_k \phi(f^\star(x))[k].$$

Furthermore, recall that $\widehat{y}_t$ denotes the action chosen by the learner at round $t$ of interaction. Starting from the definition of the regret, we get that

$$\mathrm{Reg}_T = \sum_{t=1}^{T} \Pr(\widehat{y}_t \neq y_t) - \Pr(\pi^\star(x_t) \neq y_t)$$

$$= \sum_{t=1}^{T} \mathbf{1}\{\widehat{y}_t \neq \pi^\star(x_t)\} \cdot |\Pr(y_t = \pi^\star(x_t)) - \Pr(y_t = \widehat{y}_t)|$$

$$= \sum_{t=1}^{T} \mathbf{1}\{\widehat{y}_t \neq \pi^\star(x_t)\} \cdot |\phi(f^\star(x_t))[\pi^\star(x_t)] - \phi(f^\star(x_t))[\widehat{y}_t]|$$

$$\leq \sum_{t=1}^{T} \mathbf{1}\{\widehat{y}_t \neq \pi^\star(x_t)\} \cdot \mathrm{Gap}(f^\star(x_t), \widehat{y}_t),$$

where the last inequality plugs in the definition of $\mathrm{Gap}$ from (45). Let $\varepsilon > 0$ be a free parameter. We can decompose the above regret bound further as:

$$\mathrm{Reg}_T \leq \sum_{t=1}^{T} \mathbf{1}\{\widehat{y}_t \neq \pi^\star(x_t), \mathrm{Gap}(f^\star(x_t), \widehat{y}_t) \leq \varepsilon\} \cdot \mathrm{Gap}(f^\star(x_t), \widehat{y}_t)$$

$$+ \sum_{t=1}^{T} \mathbf{1}\{\widehat{y}_t \ne \pi^\star(x_t), \mathrm{Gap}(f^\star(x_t), \widehat{y}_t) > \varepsilon\} \cdot \mathrm{Gap}(f^\star(x_t), \widehat{y}_t)$$

Using the fact that $\mathsf{y}_t(x_t) = \mathrm{argmax}_{k \in [K]} \phi(f_t(x))[k]$ in the above along with the definition of Gap and Lemma 12, we get that

$$\mathrm{Reg}_T \le \sum_{t=1}^{T} \mathbf{1}\{\widehat{y}_t \ne \pi^\star(x_t), \mathrm{Gap}(f^\star(x_t), \widehat{y}_t) \le \varepsilon\} \cdot \varepsilon$$

$$+ 2\gamma \sum_{t=1}^{T} \mathbf{1}\{\widehat{y}_t \ne \pi^\star(x_t), \mathrm{Gap}(f^\star(x_t), \widehat{y}_t) > \varepsilon\} \cdot \|f^\star(x_t) - f_t(x_t)\|$$

$$\le \sum_{t=1}^{T} \mathbf{1}\{\mathtt{Margin}(f^\star(x_t)) \le \varepsilon\} \cdot \varepsilon$$

$$+ 2\gamma \sum_{t=1}^{T} \mathbf{1}\{\widehat{y}_t \ne \pi^\star(x_t), \mathrm{Gap}(f^\star(x_t), \widehat{y}_t) > \varepsilon\} \cdot \|f^\star(x_t) - f_t(x_t)\|$$

$$\le \sum_{t=1}^{T} \mathbf{1}\{\mathtt{Margin}(f^\star(x_t)) \le \varepsilon\} \cdot \varepsilon$$

$$+ 2\gamma \sum_{t=1}^{T} Z_t \mathbf{1}\{\widehat{y}_t \ne \pi^\star(x_t), \mathrm{Gap}(f^\star(x_t), \widehat{y}_t) > \varepsilon\} \cdot \|f^\star(x_t) - f_t(x_t)\|$$

$$+ 2\gamma \sum_{t=1}^{T} \bar{Z}_t \mathbf{1}\{\widehat{y}_t \ne \pi^\star(x_t)\} \cdot \|f^\star(x_t) - f_t(x_t)\|$$

$$= T_\varepsilon \cdot \varepsilon + 2\gamma \cdot T_A + 2\gamma \cdot T_B \cdot \|f^\star(x_t) - f_t(x_t)\|,$$

where the second inequality holds because $\mathrm{Gap}(f^\star(x_t), \widehat{y}_t) \le \varepsilon$ implies that $\mathtt{Margin}(f^\star(x_t)) \le \varepsilon$ whenever $\widehat{y}_t \ne \pi^\star(x_t)$. In the last line we plugged in the definition of $T_\varepsilon$, and defined $T_A$ and $T_B$ as the second term and the last term respectively. We bound term separately below:

- *Bound on $T_A$:* Note that whenever $Z_t = 1$, we choose $\widehat{y}_t = \mathrm{argmax}_k \phi(f_t(x_t))[k]$. Thus,

$$T_A = \sum_{t=1}^{T} Z_t \mathbf{1}\{\widehat{y}_t \ne \pi^\star(x_t), \mathrm{Gap}(f^\star(x_t), \widehat{y}_t) > \varepsilon\} \cdot \|f^\star(x_t) - f_t(x_t)\|$$

$$\le \sum_{t=1}^{T} Z_t \mathbf{1}\{\|f^\star(x_t) - f_t(x_t)\| > \varepsilon/2\gamma\} \cdot \|f^\star(x_t) - f_t(x_t)\|$$

where the second line follows from Lemma 12 and because $\widehat{y}_t \ne \pi^\star(x_t)$. Using the fact that $\mathbf{1}\{a \ge b\} \le a/b$ for all $a, b \ge 0$, we get that

$$T_A \le 4\gamma \sum_{t=1}^{T} Z_t \frac{\|f^\star(x_t) - f_t(x_t)\|^2}{\varepsilon}. \tag{38}$$

- *Bound on $T_B$:* Fix any $t \le T$, and let $e$ be such that $\tau_e \le t < \tau_{e+1}$. Next, note that from Lemma 14, we have

$$\|f^\star(x_t) - g_t(x_t)\| \le \Delta_e(x_t). \tag{39}$$

Plugging in the definition of $Z_t$, we note that

$$T_B = \sum_{t=1}^{T} \mathbf{1}\{\mathtt{Margin}(g_t(x_t)) > 2\gamma\Delta_e(x_t), \widehat{y}_t \ne \pi^\star(x_t)\}$$

$$\le \sum_{t=1}^{T} \mathbf{1}\{\|g_t(x_t) - f^\star(x_t)\| > \Delta_e(x_t), \mathtt{Margin}(f^\star(x_t)) > \varepsilon\},$$

where the second inequality is due Lemma 12 and by noting that $\widehat{y}_t \ne \pi^\star(x_t)$ and that when $Z_t = 0$, we choose $\widehat{y}_t = \mathrm{argmax}_k \phi(g_t(x_t))[k]$. However, note that the term inside the indicator contradicts (39) (which always holds). Thus,

$$T_B = 0. \tag{40}$$

Combining the bounds (38) and (40), we get that:

$$\text{Reg}_T \le \varepsilon T_\varepsilon + 8\gamma^2 \sum_{t=1}^{T} Z_t \frac{\|f_t(x_t) - f^\star(x_t)\|^2}{\varepsilon}$$

$$\le \varepsilon T_\varepsilon + \frac{8\gamma^2}{\varepsilon} \Psi_\delta^{\ell_\phi}(\mathcal{F}, T),$$

where the last inequality is due to Lemma 14.

Since $\varepsilon$ is a free parameter above, the final bound follows by choosing the best parameter $\varepsilon$, and by plugging in the form of $\Psi_\delta^{\ell_\phi}(\mathcal{F}, T)$.

### E.4.3 Total Number of Queries

Let $N_T$ denote the total number of expert queries made by the learner within $T$ rounds of interactions. For the ease of notation, define $\Delta_t(x_t) = \Delta_{e(t)}(x_t)$ where $e(t)$ denotes the epoch number for which $\tau_{e(t)} \le t < \tau_{e(t)+1}$. Additionally, let $\varepsilon > 0$ be a free parameter. Thus,

$$N_T = \sum_{t=1}^{T} Z_t$$

$$= \sum_{t=1}^{T} \mathbf{1}\{\texttt{Margin}(g_t(x_t)) \le 2\gamma\Delta_t(x_t)\}$$

$$= \sum_{t=1}^{T} \mathbf{1}\{\texttt{Margin}(g_t(x_t)) \le 2\gamma\Delta_t(x_t), \texttt{Margin}(f^\star(x_t)) \le \varepsilon\}$$

$$+ \sum_{t=1}^{T} \mathbf{1}\{\texttt{Margin}(g_t(x_t)) \le 2\gamma\Delta_t(x_t), \texttt{Margin}(f^\star(x_t)) > \varepsilon\}$$

$$\le \sum_{t=1}^{T} \mathbf{1}\{\texttt{Margin}(f^\star(x_t)) \le \varepsilon\}$$

$$+ \sum_{t=1}^{T} \mathbf{1}\{\texttt{Margin}(g_t(x_t)) \le 2\gamma\Delta_t(x_t), \texttt{Margin}(f^\star(x_t)) > \varepsilon, \Delta_t(x_t) \le \varepsilon/4\gamma\}$$

$$+ \sum_{t=1}^{T} \mathbf{1}\{\texttt{Margin}(g_t(x_t)) \le 2\gamma\Delta_t(x_t), \texttt{Margin}(f^\star(x_t)) > \varepsilon, \Delta_t(x_t) > \varepsilon/4\gamma\}$$

$$= T_\varepsilon + \texttt{T}_D + \texttt{T}_E,$$

where in the last line we used the definition of $T_\varepsilon$ and defined $\texttt{T}_D$ and $\texttt{T}_E$ respectively, which we bound separately below.

- *Bound on* $\texttt{T}_D$. From Lemma 14 recall that $\|f^\star(x_t) - g_t(x_t)\| \le \Delta_e(x_t)$. Thus, for any $x_t$ for which $\|g_t(x_t)\| \le \Delta_e(x_t)$, Lemma 13 implies that

  $$\texttt{Margin}(f^\star(x_t)) \le 2\gamma\|f_t(x_t) - f^\star(x_t)\| + \texttt{Margin}(g_t(x_t)) \le 2\gamma\Delta_t(x_t) + \texttt{Margin}(f_t(x_t)).$$

  The above implies that

  $$\texttt{T}_D = \sum_{t=1}^{T} \mathbf{1}\{\texttt{Margin}(g_t(x_t)) \le 2\gamma\Delta_t(x_t), \texttt{Margin}(f^\star(x_t)) > \varepsilon, \Delta_t(x_t) \le \varepsilon/4\gamma\}$$

  $$\le \sum_{t=1}^{T} \mathbf{1}\{\texttt{Margin}(f^\star(x_t)) \le 4\gamma\Delta_t(x_t), \texttt{Margin}(f^\star(x_t)) > \varepsilon, \Delta_t(x_t) \le \varepsilon/4\gamma\}$$

  $$\le 0,$$

  where the last line follows from the fact that all the conditions inside the indictor can not hold simultaneously for any $\varepsilon > 0$.

- *Bound on* $\texttt{T}_E$. We note that

  $$\texttt{T}_E = \sum_{t=1}^{T} \mathbf{1}\{\texttt{Margin}(g_t(x_t)) \le 2\gamma\Delta_t(x_t), \texttt{Margin}(f^\star(x_t)) > \varepsilon, \Delta_t(x_t) > \varepsilon/4\gamma\}$$

$$\leq \sum_{t=1}^{T} Z_t \mathbf{1}\{\Delta_t(x_t) \geq \varepsilon/4\gamma\}$$

$$\leq \sum_{t=1}^{T} Z_t \mathbf{1}\{\Delta_{e(t)}(x_t) \geq \varepsilon/4\gamma\}.$$

Using Lemma 16 with $\zeta = \varepsilon/4\gamma$ to bound the term on the right hand side above, we get that with probability at least $1 - 2\delta$,

$$\mathsf{T}_E \leq O\left(\log(T)\gamma^2 \cdot \frac{\widehat{\Psi}_{\delta}^{\ell_\phi}(\mathcal{F};T)}{\varepsilon^2} \cdot \theta^{\mathrm{val}}\left(\mathcal{F}, \frac{\varepsilon}{8\gamma}, \frac{\widehat{\Psi}_{\delta}^{\ell_\phi}(\mathcal{F};T)}{T}; f^\star\right) + \log(2/\delta)\right).$$

Gathering the bounds above, we get that

$$N_T \leq T_\varepsilon + O\left(\log(T)\gamma^2 \cdot \frac{\widehat{\Psi}_{\delta}^{\ell_\phi}(\mathcal{F};T)}{\varepsilon^2} \cdot \theta^{\mathrm{val}}\left(\mathcal{F}, \frac{\varepsilon}{8\gamma}, \frac{\widehat{\Psi}_{\delta}^{\ell_\phi}(\mathcal{F};T)}{T}; f^\star\right) + \log(2/\delta)\right).$$

Since $\varepsilon$ is a free parameter above, the final bound follows by choosing the best parameter $\varepsilon$, and by plugging in the form of $\widehat{\Psi}_{\delta}^{\ell_\phi}(\mathcal{F};T)$.

### E.5 Proof of Corollary 1

Note that the Tsybakov noise condition implies that there exists constants $c, \rho \geq 0$ such that for all $\varepsilon \in (0,1)$:

$$\Pr_{x\sim\mu}(\mathtt{Margin}(f^\star(x_t)) \leq \varepsilon) \leq c\varepsilon^\rho.$$

Thus, using Lemma 4, we get that

$$\begin{aligned} T_\varepsilon &= \sum_{t=1}^{T} \mathbf{1}\{\mathtt{Margin}(f^\star(x_t)) \leq \varepsilon\} \\ &\leq \frac{3T}{2} \Pr_{x\sim\mu}(\mathtt{Margin}(f^\star(x)) \leq \varepsilon) + 4\log(2/\delta) \\ &\leq 2cT\varepsilon^\rho + 4\log(2/\delta). \end{aligned}$$

Using the above in the bound for Theorem 2, we get that for any $\varepsilon > 0$,

$$\mathrm{Reg}_T \lesssim cT\varepsilon^{\rho+1} + \frac{\gamma^2}{\lambda\varepsilon}\mathrm{Reg}^{\ell_\phi}(\mathcal{F};T) + \log(1/\delta),$$

Setting $\varepsilon = \left(\frac{\gamma^2}{\lambda CT}\mathrm{Reg}^{\ell_\phi}(\mathcal{F};T)\right)^{\frac{1}{\rho+2}}$ in the above implies that

$$\mathrm{Reg}_T \lesssim \left(\frac{\gamma^2}{\lambda}c^{\frac{1}{\rho+1}}\right)^{\frac{\rho+1}{\rho+2}}\left(\mathrm{Reg}^{\ell_\phi}(\mathcal{F};T)\right)^{\frac{\rho+1}{\rho+2}} \cdot (T)^{\frac{1}{\rho+2}} + \log(1/\delta).$$

Similarly, we can bound the query complexity bound for any $\varepsilon > 0$ as:

$$N_T \lesssim T\varepsilon^\rho + \frac{\gamma^2}{\lambda\varepsilon^2} \cdot \mathrm{Reg}^{\ell_\phi}(\mathcal{F};T) \cdot \theta^{\mathrm{val}}\left(\mathcal{F}, \varepsilon/8\gamma, \mathrm{Reg}^{\ell_\phi}(\mathcal{F};T)/T; f^\star\right) + \log(1/\delta).$$

Setting $\varepsilon = \left(\frac{\gamma^2}{\lambda T} \cdot \mathrm{Reg}^{\ell_\phi}(\mathcal{F};T) \cdot \theta^{\mathrm{val}}\left(\mathcal{F}, \varepsilon/8\gamma, \mathrm{Reg}^{\ell_\phi}(\mathcal{F};T)/T; f^\star\right)\right)^{\frac{1}{\rho+2}}$ in the above implies that

$$N_T \leq \left(\frac{\gamma^2}{\lambda} \cdot \mathrm{Reg}^{\ell_\phi}(\mathcal{F};T) \cdot \theta^{\mathrm{val}}\left(\mathcal{F}, \varepsilon/8\gamma, \mathrm{Reg}^{\ell_\phi}(\mathcal{F};T)/T; f^\star\right)\right)^{\frac{\rho}{\rho+2}} \cdot T^{\frac{2}{\rho+2}}.$$

### E.6 Proofs for Lower Bounds in Section 3.2

The proof of Theorem 3 below follows along the lines of the proof of the lower bound in Theorem 28 of Foster et al. [2020] with minor changes.

*Proof of Theorem 3.* Let $\beta \leq \zeta/2$ be the largest number such that $\beta^2 \leq \min\{\zeta^2/\mathfrak{s}^{\mathrm{val}}(\mathcal{F}, \zeta, \beta), \zeta^2/16\}$. Given the function class $\mathcal{F}$, assume that $m = \mathfrak{s}^{\mathrm{val}}(\mathcal{F}, \zeta, \beta) > 0$ (the lower bound is obvious when the star number is 0). Let the target function $f^\star$, the data sequence $x^1, \ldots, x^m$, and the function $f_1, \ldots, f_m \in \mathcal{F}$ be the witnesses for the fact that the star number is $m$ (see Definition 3). First, for any $i \in [m]$, we have that $|f^\star(x^i)| > \zeta$ by definition of the star number. Next note that, for each $f_i$, we have that, for any $j \neq i$,

$$|f_i(x^j)| \geq |f^\star(x^j)| - |f_i(x^j) - f^\star(x^j)| \geq \zeta - \beta \geq \zeta/2$$

On the other hand, from our definition of star number we have that,

$$|f_i(x^i)| \geq \zeta/2.$$

Hence we are guaranteed that each $f_i$ has a margin of at least $\zeta/2$ on $x^1, \ldots, x^m$. Now consider the distribution $\mu$ over the context to be the uniform distribution over $\{x^1, \ldots, x^m\}$. Also let $P^i(y = 1 \mid x) = 1 + f_i(x)/2$ be the conditional probability of label given context $x$. Let $D_i$ denote the joint distribution over $\mathcal{X} \times \{\pm 1\}$ given by drawing $x$'s from $\mu$ and labels from $P^i$. Additionally, let $D_0$ be given by drawing $x$'s from $\mu$ and $y$ conditioned on $x$ as $P(y = 1 \mid x) = 1 + f^\star(x)/2$. Finally, let $\nu = 0$ with probability $1/2$ and $\nu \sim \mathrm{Uniform}([m])$ with probability $1/2$. Note that by our premise,

$$\frac{1}{2}\mathbb{E}_{D_0}\left[\mathrm{Reg}_T\right] + \frac{1}{2m}\sum_{i=1}^{m}\mathbb{E}_{D_i}\left[\mathrm{Reg}_T\right] = \mathbb{E}_\nu\left[\mathbb{E}_{D_\nu}\left[\mathrm{Reg}_T\right]\right] \leq c\frac{\zeta T}{m}, \tag{41}$$

where the value of the constant $c$ will be set later. Furthermore, let $p_t : \mathcal{X} \mapsto \Delta(\mathcal{Y})$ denote the distribution over the label chosen by the given algorithm at round $t$ given the history $(x_1, y_1, \ldots, x_{t-1}, y_{t-1})$. We note that for any $i \in [m]$,

$$\mathbb{E}_{D_i}\left[\mathrm{Reg}_T\right] \geq \mathbb{E}_{D_i}\left[\sum_{t=1}^{T}\frac{\zeta}{2}\mathbb{E}_{x\sim\mu}\mathbb{E}_{\hat{y}\sim p_t(x)}\left[\mathbf{1}\{f_i(x)\hat{y} < 0\}\right]\right]$$

$$= \frac{T\zeta}{2}\mathbb{E}_{D_i}\left[\frac{1}{T}\sum_{t=1}^{T}\mathbb{E}_{x\sim\mu}\mathbb{E}_{\hat{y}\sim p_t(x)}\mathbf{1}\{\mathrm{sign}(f_i(x)) \neq \hat{y}\}\right]$$

$$\geq \frac{T\zeta}{4}\mathbb{E}_{D_i}\left\|\frac{1}{T}\sum_{t=1}^{T}p_t - \mathrm{sign}(f_i)\right\|_{L_1(\mu)}$$

where in the first inequality we used that $f_i$ has a margin of at least $\zeta/2$ for any $x \in \{x^1, \ldots, x^m\}$ as shown above, and the last inequality simply follows from the fact that

$$\left|\mathbb{E}_{\hat{y}\sim p_t(x)}\mathbf{1}\{\mathrm{sign}(f_i(x)) \neq \hat{y}\}\right| = \left|\mathbb{E}_{\hat{y}\sim p_t(x)}(\mathbf{1}\{\mathrm{sign}(f_i(x)) = 1\} - \mathbf{1}\{\hat{y} = 0\})\right| = \frac{1}{2}|p_t(x) - f_i(x)|,$$

and via an application of the Jensen's inequality. Using the property that for any distribution with margin at least $\zeta/2$, the algorithm satisfies $\mathbb{E}_{D_i}[\mathrm{Reg}_T] \leq c\zeta T/m$, the above implies that

$$\mathbb{E}_{D_i}\left\|\frac{1}{T}\sum_{t=1}^{T}p_t - \mathrm{sign}(f_i)\right\|_{L_1(\mu)} \leq \frac{8c}{m},$$

which implies that

$$\frac{1}{m}\sum_{i=1}^{m}\mathbb{E}_{D_i}\left\|\frac{1}{T}\sum_{t=1}^{T}p_t - \mathrm{sign}(f_i)\right\|_{L_1(\mu)} \leq \frac{8c}{m}. \tag{42}$$

Hence, setting $c = 64$ and using Markov's inequality, we have that

$$\frac{1}{m}\sum_{i=1}^{m}\mathrm{Pr}_{D_i}\left(\left\|\frac{1}{T}\sum_{t=1}^{T}p_t - \mathrm{sign}(f_i)\right\|_{L_1(\mu)} > \frac{2}{m}\right) \leq \frac{1}{16}.$$

Further note that $\|\text{sign}(f_j) - \text{sign}(f_i)\|_{L_1(\mu)} > \frac{4}{m}$ and thus, condition on the fact that $\nu$ is not equal to 0, we can can identify $\nu$ correctly with probability at least $1 - 1/16$. Thus, considering the reference measure $D_0$ and using Fano's inequality, we must have that

$$\log(2) \le \frac{1}{m} \sum_{i=1}^{m} \text{KL}(D_0 \| D_i)$$

Now we are left with bounding $\text{KL}(D_0 \| D_i)$. To this end, we first make a simple observation that the distribution on the $x_t$'s is the same under $D_0$ and $D_i$. Hence on rounds $t$ where $Z_t = 0$ we do not query for the labels in these rounds, and thus we do not glean any new information to distinguish $D_i$ from $D_0$. In other words, we only need to consider rounds when $Z_t = 1$. Hence we have:

$$\text{KL}(D_0 \| D_i) = \mathbb{E}_{D_0} \left[ \sum_{t=1}^{T} Z_t \cdot \text{kl}(P_0(y_t = 1|x_t) \| P_i(y_t = 1|x_t)) \right]$$

Assuming $\zeta > 1/4$ and using the bound on KL between Bernoulli variables we get,

$$\text{KL}(D_0 \| D_i) \le \mathbb{E}_{D_0} \left[ 32 N_i \zeta^2 + 8 \left( \max_{j \neq i} N_j \right) \beta^2 \right]$$

where $N_i = |\{t : Z_t = 1, x_t = x^i\}$, and we used item-(3) in the definition of star number for the $\zeta^2$ term and item-(1) for the $\beta^2$ term. Hence we have that,

$$\begin{aligned}
\log(2) &\le \frac{1}{m} \sum_{i=1}^{m} \mathbb{E}_{D_0} \left[ 32 N_i \zeta^2 + 8 \left( \max_{j \neq i} N_j \right) \beta^2 \right] \\
&\le \frac{32}{m} \mathbb{E}_{D_0} \left[ \sum_{i=1}^{m} N_i \right] \zeta^2 + 8 \mathbb{E}_{D_0} \left[ \max_j N_j \right] \beta^2 \\
&\le \frac{32}{m} \mathbb{E}_{D_0} [N_T] \zeta^2 + 8 \mathbb{E}_{D_0} [N_T] \beta^2
\end{aligned}$$

Since $\beta^2 \le \zeta^2/m$, we have that

$$\log(2) \le \frac{40}{m} \mathbb{E}_{D_0} [N_T] \zeta^2$$

Hence we conclude that,

$$\mathbb{E}_{D_0} [N_T] \ge \frac{\log(2)m}{40\zeta^2}$$

which yields the desired lower bound. $\qquad \square$

*Proof of Corollary 2.* Consider the function class $\mathcal{F} = \{f_0, f_1, \ldots, f_{\sqrt{T}}\}$ where $f_0(x_i) = 1/2 + \zeta$ for every $x_1, \ldots, x_{\sqrt{T}}$, and where $f_i(x_i) = 1/2 - \zeta$ and $f_i(x_j) = 1/2 + \zeta$ for any $j \neq i$. Note that selecting $f^\star = f_0$ and $f_1, \ldots, f_m$ on $x_1, \ldots, x_{\sqrt{T}}$, we can show that the star number $\mathfrak{s}^{\text{val}}(\mathcal{F}, 1/8, 1/2) = O(\sqrt{T})$ (the disagreement coefficient of $\mathcal{F}$ is also $O(\sqrt{T})$). Thus, using the converse of Theorem 3 we get that if the number of queries is smaller than $\sqrt{T}$, then there must exist some data distribution under which the regret bound of the algorithm is larger that $\sqrt{T}$. $\qquad \square$

# F Imitation Learning: Learning from Single Expert

## F.1 Imitation Learning Tools

We first recall useful additional notation. Recall that a policy $\pi$ is a mapping from the states $\mathcal{X}$ to actions $\mathcal{A}$. For any $h \leq H$, and random variable $Z(x_h, a_h)$, we use the notation $\mathbb{E}_\pi[Z(x_h, a_h)]$ to denote the expectation w.r.t. trajectories $\{x_1, a_1 \ldots, x_H, a_H\}$ sampled using the policy $\pi$.

The following lemma is the standard performance difference lemma, well known in the imitation learning and reinforcement learning literature.

**Lemma 17** (Performance Difference Lemma; Kakade and Langford [2002], Ross and Bagnell [2014])**.** *For any MDP $M$, and any two arbitrary stationary policies $\pi$ and $\widetilde{\pi}$, we have*

$$V^\pi - V^{\widetilde{\pi}} = \sum_{h=1}^H \mathbb{E}_{x_h, a_h \sim d_h^{\widetilde{\pi}}}[-A_h^\pi(x_h, a_h)],$$

*where $A^\pi$ is the advantage function of the policy $\pi$ in MDP $M$, i.e., $A_h^\pi(x, a) = Q_h^\pi(x, a) - V_h^\pi(x)$.*

## F.2 Proof of Proposition 1

**MDP construction.** The underlying MDP is a binary tree of depth $H$. In particular, we construct the deterministic MDP $M = (\mathcal{X}, \mathcal{A}, P, r, x_1)$ where state space $\mathcal{X} = \cup_{h=1}^H \mathcal{X}_h$ with $\mathcal{X}_h = \{x_{h,i}\}_{i=1}^{2^{h-1}}$ (we assume that $x_1 = x_{1,1}$), action space $\mathcal{A} = \{0, 1\}$, reward $r$ is such that $r(x, a) = \text{Bern}\left(\frac{1}{2} + \frac{1}{4}\mathbf{1}\{x = x^\star\}\right)$ for some special state $x^\star \in \mathcal{X}_H$. The transition dynamics $P$ is deterministic and defines a binary tree over $\mathcal{X}$, i.e. for any $h$ and $x_{h,i}$, $P(x' \mid x_{h,i}, a) = 1$ if $x' = x_{h+1,2i-1}$ and $a = 0$, or $x' = x_{h+1,2i+1}$ and $a = 1$, else $P(x' \mid x, a) = 0$.

We next define the expert policy $\pi^\star$, expert model $f^\star$ and the class $\mathcal{F}$. First, for any path $\tau = (x_1, a_1, \ldots, x_H, a_H)$ from the root state $x_1$ to a terminal state $x_H$ at the layer $H$, define the policy $\pi_\tau$ as

$$\pi_\tau(x_h) = \begin{cases} a_h & \text{if} \quad (x_h, a_h) \in \tau \\ \bar{a}_h \leftarrow \text{Uniform}(\{0, 1\}) & \text{otherwise} \end{cases}.$$

In particular, $\pi_\tau$ is defined such that for any state on the path $\tau$, we choose the corresponding action in $\tau$, and for any state outside of $\tau$, we choose an arbitrary (deterministic) action. Let $\mathcal{T}$ denote the set of all $2^H$ many paths from the root note $x_1$ to a leaf node $x_H \in \mathcal{X}_H$. We define the class $\Pi = \{\pi_\tau \mid \tau \in \mathcal{T}\}$, and $\mathcal{F} = \{f_\tau \mid \tau \in \mathcal{T}\}$, where for any $\tau$, we define $f_\tau : \mathcal{X} \mapsto \mathbb{R}^2$ as

$$f_\tau(x) = \begin{cases} (3/4, 1/4) & \text{if} \quad \pi_\tau(x) = 0 \\ (1/4, 3/4) & \text{if} \quad \pi_\tau(x) = 1 \end{cases}.$$

Next, let $\tau^\star = (x_1, a_1', x_2', \ldots, x_{H-1}', a_{H-1}', x^\star)$ be the path from the root $x_1$ to the special state $x^\star \in \mathcal{X}_H$ on the underlying binary tree. We finally define $\pi^\star = \pi_{\tau^\star}$ and $f^\star = f_{\tau^\star}$.

**Lower bound.** Given the MDP construction, the class $\mathcal{F}$, $f^\star$ and $\pi^\star$ above, we now proceed to the desired lower bound for non-interactive imitation learning. First, note that $\pi^\star(x) = \text{argmax}_a(f^\star(x)[a])$ for any $x \in \mathcal{X}$. Furthermore, $\text{Margin}(f^\star((x)) = \frac{1}{2}|f(x)[0] - f(x)[1]| = \frac{1}{4}$ for all $x \in \mathcal{X}$. Thus, for any $\varepsilon \leq \frac{1}{4}$, $T_{\varepsilon, h} = 0$.

Next, for any policy $\pi$, note that $V^\pi = \frac{1}{2} + \frac{1}{4}\mathbf{1}\{\pi = \pi^\star\}$. Thus, $\pi^\star$ is the unique $1/8$-suboptimal policy. Additionally, consider a noisy expert that draws its label according to (2) with the link function $\phi(z) = z$, i.e. on the state $x$, the expert draws its label from $a \sim f^\star(x)$. Now, suppose that the learner is given a dataset $\mathcal{D}$ of $m$ many trajectories drawn this noisy expert. There are two scenarios: either $\mathcal{D}$ does not contain $\tau^*$, or $\mathcal{D}$ contains the trajectory $\tau^*$.

- In the first case, the learner is restricted to finding $\pi^\star$ by eliminating all other $\pi \neq \pi^\star$ using the observations $\mathcal{D}$. Since, $|\Pi| = 2^H$ and each policy in the class is associated with a different path on the tree, we must have that $m = O(2^H)$.

- In the second case, we need $\tau^\star \in \mathcal{D}$. However, note that probability of observing the trajectory $\tau^\star$ when following the actions proposed by the noisy expert is $\Pr(\tau^\star \mid a_h \sim f^\star(x_h)) = (3/4)^H$. Thus, in order to observe $\tau^\star$ with probability at least $3/4$ in the dataset $\mathcal{D}$, we need $m = O((4/3)^H)$.

In both the scenarios above, we need to collect exponentially many samples.

### F.3 Proof of Theorem 4

Before delving into the proof, we recall the relevant notation. In Algorithm 2, for any $h \leq H$,

- The label $y_{t,h} \sim \phi(f^\star(x_{t,h}))$, , where $\phi$ denotes the link-function given in (2).
- The function $\texttt{SelectAction}(f_{t,h}(x_{t,h})) = \operatorname{argmax}_k \phi(f_{t,h}(x_{t,h}))[k]$.
- For any vector $v \in \mathbb{R}^K$, the margin is given by the gap between the value at the largest and the second largest coordinate (under the link function $\phi$), i.e.

$$\texttt{Margin}(v) = \phi(v)[k^\star] - \max_{k \neq k^\star} \phi(v)[k],$$

where $k^\star \in \operatorname{argmax}_{k \in [K]} \phi(v)[k]$.

- We define $T_\varepsilon = \sum_{t=1}^T \sum_{h=1}^H \mathbf{1}\{\texttt{Margin}(f_h^\star(x_{t,h})) \leq \varepsilon\}$ to denote the number of samples within $T$ rounds of interaction for which the margin w.r.t. $f_h^\star$ is smaller than $\varepsilon$.
- The trajectory at round $t$ is generated using the dynamics $\{\mathbb{T}_{t,h}\}_{h \leq H}$ to determine the states that the learner observes, starting from the state $x_1$.
- At round $t$, the learner collects data using the policy $\pi_t$ such that at time $h$, and state $x$, the action $\pi_t(x) = \texttt{SelectAction}(f_{t,h}(x_{t,h}))$.
- For any policy $\pi$, let $\tau_t^\pi$ denote the (counterfactual) trajectory that one would obtain by running $\pi$ on the deterministic dynamics $\{\mathbb{T}_{t,h}\}_{h \leq H}$ with the start state $x_{t,1}$, i.e.

$$\tau_t^\pi = \left\{ x_{t,1}^\pi, \pi(x_{t,1}^\pi), \ldots, x_{t,H}^\pi, \pi(x_{t,H}^\pi) \right\} \tag{43}$$

where $x_{t,1}^\pi = x_{t,1}$ and $x_{t,h+1}^\pi = \mathbb{T}_{t,h}(x_{t,h}^\pi, \pi(x_{t,h}^\pi))$.

- For a trajectory $\tau = \{x_1, a_1, \ldots, x_H, a_H\}$, we define the total return

$$R(\tau) = \sum_{h=1}^H r(x_h, a_h). \tag{44}$$

- Additionally, for any policy $\pi$ and dynamics $\{\mathbb{T}_h\}_{h \leq H}$, we define the trajectory obtained by running the policy $\pi$ as

$$\tau^\pi = \{x_1^\pi, \pi_1(x_1^\pi), x_2^\pi, \ldots\}.$$

- We define the function $\operatorname{Gap} : \mathbb{R}^K \times [K] \mapsto \mathbb{R}^+$ as

$$\operatorname{Gap}(v, k) = \max_{k'} \phi(v)[k'] - \phi(v)[k], \tag{45}$$

to denote the gap between the largest and the $k$-th coordinate of $v$, and note that $\texttt{Margin}(v) \leq \operatorname{Gap}(v, k)$ for all $k \neq k^\star$ (due to Lemma 9).

#### F.3.1 Supporting Technical Results

We first define a useful technical lemma which allows us to bound the gap between the total returns for policies $\pi_1$ and $\pi_2$, under the dynamics $\{\mathbb{T}_h\}_{h \leq H}$. Recall that for a policy $\pi$, we define the trajectory $\tau^\pi$ under $\{\mathbb{T}_h\}_{h \leq H}$ and the start state $x_1$ as the trajectory $\{x_1^\pi, \pi(x_1^\pi), \ldots, x_H^\pi, \pi(x_H^\pi)\}$ where $x_1^\pi = x_1$, and $x_{h+1}^\pi \leftarrow \mathbb{T}_h(x_h^\pi, \pi(x_h^\pi))$.

**Lemma 18.** *Let $\{\mathbb{T}_h\}_{h \leq H}$ be a deterministic dynamics, and let $x_1$ be the start state. Let $\pi_1$ and $\pi_2$ be any two deterministic policies, and let $\tau^{\pi_1} = \{x_1^{\pi_1}, \pi_1(x_1^{\pi_1}), x_2^{\pi_1}, \ldots\}$ and $\tau^{\pi_2} = \{x_1^{\pi_2}, \pi_1(x_1^{\pi_2}), x_2^{\pi_2}, \ldots\}$ be two trajectories drawn using $\pi_1$ and $\pi_2$ on $\{\mathbb{T}_h\}_{h \leq H}$ with start state $x_1$. Then, for any set $\mathtt{X} \subseteq \mathcal{X}$, the total trajectory rewards satisfy*

$$R(\tau^{\pi_1}) - R(\tau^{\pi_2}) \leq 2H \sum_{h=1}^H \mathbf{1}\{x_h^{\pi_1} \in \mathtt{X}\} + 2H \sum_{h=1}^H \mathbf{1}\{\pi_2(x_h^{\pi_2}) \neq \pi_1(x_h^{\pi_2}), x_h^{\pi_2} \notin \mathtt{X}\}.$$

*Proof.* Let $\mathfrak{h} \le H$ denote the first timestep at which the policies $\pi_1$ and $\pi_2$ choose different actions under $\{\mathbb{T}_h\}_{h \le H}$. Since the trajectories $\tau^{\pi_1} = \{x_1^{\pi_1}, \pi_1(x_1^{\pi_1}), x_2^{\pi_1}, \ldots\}$ and $\tau^{\pi_2} = \{x_1^{\pi_2}, \pi_1(x_1^{\pi_2}), x_2^{\pi_2}, \ldots\}$ are obtained by evolving through (the deterministic dynamics) $\{\mathbb{T}_h\}_{h \le H}$ using policies $\pi_1$ and $\pi_2$ respectively, and with the same state state $x_1$, we have that

$$x_h^{\pi_1} = x_h^{\pi_2} \qquad \text{for all } h \le \mathfrak{h},$$

and

$$\pi_1(x_h^{\pi_1}) = \pi_2(x_h^{\pi_2}) \qquad \text{for all } h \le \mathfrak{h} - 1. \tag{46}$$

Starting from the definition of the cumulative reward $R(\cdot)$, we have that

$$
\begin{aligned}
R(\tau^{\pi_1}) - R(\tau^{\pi_2}) &= \sum_{h=1}^{H} \big( r(x_h^{\pi_1}, \pi_1(x_h^{\pi_1})) - r(x_h^{\pi_2}, \pi_2(x_h^{\pi_2})) \big) \\
&= \sum_{h=1}^{\mathfrak{h}-1} \big( r(x_h^{\pi_1}, \pi_1(x_h^{\pi_1})) - r(x_h^{\pi_2}, \pi_2(x_h^{\pi_2})) \big) + \sum_{h=\mathfrak{h}}^{H} \big( r(x_h^{\pi_1}, \pi_1(x_h^{\pi_1})) - r(x_h^{\pi_2}, \pi_2(x_h^{\pi_2})) \big) \\
&= \sum_{h=\mathfrak{h}}^{H} \big( r(x_h^{\pi_1}, \pi_1(x_h^{\pi_1})) - r(x_h^{\pi_2}, \pi_2(x_h^{\pi_2})) \big),
\end{aligned}
$$

where the last line uses the fact that the trajectories (and thus the rewards) $\tau^{\pi_1}$ and $\tau^{\pi_2}$ are identical for the first $\mathfrak{h} - 1$ states and actions (see (46)). Since (46) also implies that $x_{\mathfrak{h}}^{\pi_1} = x_{\mathfrak{h}}^{\pi_2}$, for the ease of notation we define $x_{\mathfrak{h}} = x_{\mathfrak{h}}^{\pi_1} = x_{\mathfrak{h}}^{\pi_2}$. Using the fact that $|r(x,a)| \le 1$ and that $\pi_1(x_{\mathfrak{h}}) \ne \pi_2(x_{\mathfrak{h}})$ (by definition of $\mathfrak{h}$), we can bound the above as

$$
\begin{aligned}
R(\tau^{\pi_1}) - R(\tau^{\pi_2}) &\le 2(H - \mathfrak{h} + 1)\mathbf{1}\{\pi_1(x_{\mathfrak{h}}) \ne \pi_2(x_{\mathfrak{h}})\} \\
&\le 2H\mathbf{1}\{\pi_1(x_{\mathfrak{h}}) \ne \pi_2(x_{\mathfrak{h}})\} \\
&= 2H\mathbf{1}\{\pi_1(x_{\mathfrak{h}}) \ne \pi_2(x_{\mathfrak{h}}), x_{\mathfrak{h}} \in \mathtt{X}\} + 2H\mathbf{1}\{\pi_1(x_{\mathfrak{h}}) \ne \pi_2(x_{\mathfrak{h}}), x_{\mathfrak{h}} \notin \mathtt{X}\} \\
&\le 2H\mathbf{1}\{x_{\mathfrak{h}} \in \mathtt{X}\} + 2H\mathbf{1}\{\pi_1(x_{\mathfrak{h}}) \ne \pi_2(x_{\mathfrak{h}}), x_{\mathfrak{h}} \notin \mathtt{X}\} \\
&= 2H\mathbf{1}\{x_{\mathfrak{h}}^{\pi_1} \in \mathtt{X}\} + 2H\mathbf{1}\{\pi_2(x_{\mathfrak{h}}^{\pi_2}) \ne \pi_1(x_{\mathfrak{h}}^{\pi_2}), x_{\mathfrak{h}}^{\pi_2} \notin \mathtt{X}\} \\
&\le 2H\sum_{h=1}^{H}\mathbf{1}\{x_h^{\pi_1} \in \mathtt{X}\} + 2H\sum_{h=1}^{H}\mathbf{1}\{\pi_2(x_h^{\pi_2}) \ne \pi_1(x_h^{\pi_2}), x_h^{\pi_2} \notin \mathtt{X}\},
\end{aligned}
$$

where the equality in second last line plugs in the fact that $x_{\mathfrak{h}} = x_{\mathfrak{h}}^{\pi_1} = x_{\mathfrak{h}}^{\pi_2}$, and the last inequality is a straightforward upper bound. $\qquad\square$

We will also be using Lemma 9 and Lemma 13 from Appendix E.3 for bounding the `Margin` in the regret bound proofs. Finally, we note the following properties of the function $f_h^\star$.

**Lemma 19.** *With probability at least $1 - \delta$, the function $f_h^\star$ satisfies for any $h \le H$ and $t \le T$,*

    *(a)* $\sum_{s=1}^{t-1} Z_{s,h} \|f_h^\star(x_{s,h}) - f_{s,h}(x_{s,h})\|^2 \le \Psi_\delta^{\ell_\phi}(\mathcal{F}_h, T),$

    *(b)* $\|f_h^\star(x_{t,h}) - f_{t,h}(x_{t,h})\| \le \Delta_{t,h}(x_{t,h}),$

*where $\Psi_\delta^{\ell_\phi}(\mathcal{F}_h, T) = \frac{4}{\lambda}\mathrm{Reg}^{\ell_\phi}(\mathcal{F}_h; T) + \frac{112}{\lambda^2}\log(4H\log^2(T)/\delta)$.*

*Proof.*     *(a)* We first note that we do not query oracle when $Z_{s,h} = 0$, and thus we can ignore the time steps for which $Z_{s,h} = 0$. Hence, for each $h \in [H]$, applying Lemma 5 along with the fact that $\sup_{x,f \in \mathcal{F}_h}|f(x)| \le 1$ yields

$$\sum_{s=1}^{t-1} Z_{s,h}\|f_h^\star(x_{s,h}) - f_{s,h}(x_{s,h})\|^2 \le \frac{4}{\lambda}\mathrm{Reg}^{\ell_\phi}(\mathcal{F}_h; T) + \frac{112}{\lambda^2}\log(4\log^2(T)/\delta)$$

for all $t \le T$. Then, we take the union bound for all $h \in [H]$, which completes the proof.

    *(b)* The second part follows from using the observation in part-(a) that $f_h^\star$ satisfies the constraint in the definition of $\Delta_{t,h}$ given in (6), and thus $\|f_h^\star(x_{t,h}) - f_{t,h}(x_{t,h})\| \le \Delta_{t,h}(x_{t,h})$.

$\qquad\square$

The next technical lemma bounds the number of times when $\Delta_{t,h}(x_{t,h}) \geq \zeta$ and we query the expert. Note that Lemma 20 holds even if the sequence $\{x_{t,h}\}_{t \leq T}$ was adversarially generated.

**Lemma 20.** *Let $f^\star$ satisfy Lemma 19, and let $\Delta_{t,h}(x_t)$ be defined in (6). Suppose we run Algorithm 2 on data sequence $\{\{x_{t,h}\}_{h \leq H}\}_{t \leq T}$, and let $Z_{t,h}$ be as defined in line 9. Then, for any $\zeta > 0$, with probability at least $1 - H\delta$, for any $h \leq H$,*

$$\sum_{t=1}^{T} Z_{t,h}\mathbf{1}\{\Delta_{t,h}(x_{t,h}) \geq \zeta\} \leq \frac{20\Psi_\delta^{\ell_\phi}(\mathcal{F}_h, T)}{\zeta^2} \cdot \mathfrak{E}(\mathcal{F}_h, \frac{\zeta}{2}; f_h^\star),$$

*where $\mathfrak{E}$ denotes the eluder dimension is given in Definition 1.*

*Proof.* The proof is identical to the proof of Lemma 11 by replacing all $|\cdot|$ with $\|\cdot\|$, and substitute the corresponding bounds for $f_h^\star$ via Lemma 19 (instead of using Lemma 10). We skip the proof for conciseness. $\qquad\square$

### F.3.2   Regret Bound

Recall that the trajectory at round $t$ is generated using the dynamics $\{\mathbb{T}_{t,h}\}_{h \leq H}$. Define the policy $\pi_t$ and $\pi^\star$ such that for any $h \leq H$ and $x \in \mathcal{X}_h$,

$$\pi_t(x_h) = \texttt{SelectAction}(f_{t,h}(x_h)), \quad \text{and}, \quad \pi^\star(x_h) = \texttt{SelectAction}(f_h^\star(x_h)). \tag{47}$$

Furthermore, for any policy $\pi$, let $\tau_t^\pi$ denote the trajectory that one would obtain by running $\pi$ on the deterministic dynamics $\{\mathbb{T}_{t,h}\}_{h \leq H}$ with the start state $x_{t,1}$, i.e.

$$\tau_t^\pi = \left\{x_{t,1}^\pi, \pi(x_{t,1}^\pi), \ldots, x_{t,H}^\pi, \pi(x_{t,H}^\pi)\right\} \tag{48}$$

where $x_{t,1}^\pi = x_{t,1}$ and $x_{t,h+1}^\pi = \mathbb{T}_{t,h}(x_{t,h}^\pi, \pi(x_{t,h}^\pi))$. Note that Algorithm 2 collects trajectories using the policy $\pi_t$ at round $t$. Thus, we have that

$$x_{t,h}^{\pi_t} = x_{t,h}, \tag{49}$$

where $x_{t,h}$ denotes the state at time step $h$ in round $t$ of Algorithm 2. Finally, let $\varepsilon > 0$ be a free parameter. We now have all the notation to proceed to the proof on our regret bound.

**Step 1: Bounding the difference in return at round $t$.** Fix any $t \leq T$, and let $\tau_t^{\pi_t}$ and $\tau_t^{\pi^\star}$ denote the trajectories that would have been sampled using the policies $\pi_t$ and the policy $\pi^\star$ at round $t$. Furthermore, define the set $\mathtt{X}_\varepsilon$ as

$$\mathtt{X}_\varepsilon := \bigcup_{h=1}^{H}\left\{x \in \mathcal{X}_h \mid \texttt{Margin}(f_h^\star(x)) \leq \varepsilon\right\} \tag{50}$$

Using Lemma 18 for the policies $\pi_t$ and $\pi^\star$, and the set $\mathtt{X}_\varepsilon$ defined above, we get that[9]

$$R(\tau_t^{\pi^\star}) - R(\tau_t^{\pi_t}) \leq 2H\sum_{h=1}^{H}\mathbf{1}\{x_{t,h}^{\pi^\star} \in \mathtt{X}_\varepsilon\} + 2H\sum_{h=1}^{H}\mathbf{1}\{\pi_t(x_{t,h}^{\pi_t}) \neq \pi^\star(x_{t,h}^{\pi_t}), x_{t,h}^{\pi_t} \notin \mathtt{X}_\varepsilon\} \tag{51}$$

$$= 2H\sum_{h=1}^{H}\mathbf{1}\{x_{t,h}^{\pi^\star} \in \mathtt{X}_\varepsilon\} + 2H\sum_{h=1}^{H}\mathbf{1}\{\pi_t(x_{t,h}) \neq \pi^\star(x_{t,h}), x_{t,h} \notin \mathtt{X}_\varepsilon\}$$

$$= 2H\sum_{h=1}^{H}\mathbf{1}\{x_{t,h}^{\pi^\star} \in \mathtt{X}_\varepsilon\} + 2H\sum_{h=1}^{H}Z_{t,h}\mathbf{1}\{\pi_t(x_{t,h}) \neq \pi^\star(x_{t,h}), x_{t,h} \notin \mathtt{X}_\varepsilon\}$$

$$\qquad\qquad\qquad + 2H\sum_{h=1}^{H}\bar{Z}_{t,h}\mathbf{1}\{\pi_t(x_{t,h}) \neq \pi^\star(x_{t,h}), x_{t,h} \notin \mathtt{X}_\varepsilon\}$$

$$\leq 2H\sum_{h=1}^{H}\mathbf{1}\{x_{t,h}^{\pi^\star} \in \mathtt{X}_\varepsilon\} + 2H\sum_{h=1}^{H}Z_{t,h}\mathbf{1}\{\pi_t(x_{t,h}) \neq \pi^\star(x_{t,h}), x_{t,h} \notin \mathtt{X}_\varepsilon\}$$

---

[9]The key advantage of using Lemma 18 is that the first term $\sum_{h=1}^{H}\mathbf{1}\{x_{t,h}^{\pi^\star} \in \mathtt{X}_\varepsilon\}$ accounts for the number steps at which a counterfactual trajectory sampled using $\pi^\star$ goes to the state space with margin less than $\varepsilon$. Thus, we only pay for the number of times when the comparator policy $\pi^\star$ would go to states with $\varepsilon$-margin (instead of when $\pi_t$ does to such states).

$$+ 2H \sum_{h=1}^{H} \bar{Z}_{t,h} \mathbf{1}\{\pi_t(x_{t,h}) \ne \pi^\star(x_{t,h})\}$$

$$= 2H \sum_{h=1}^{H} \mathbf{1}\{x_{t,h}^{\pi^\star} \in \mathsf{X}_\varepsilon\} + 2H\mathtt{T}_A + 2H\mathtt{T}_B, \tag{52}$$

where the second line is obtained by plugging in (49) and the last line simply defines $\mathtt{T}_A$ and $\mathtt{T}_B$ to be the second and the third terms in the previous line without the $2H$ multiplicative factor.

We bound $\mathtt{T}_A$ and $\mathtt{T}_B$ separately below.

- *Bound on term* $\mathtt{T}_A$. Using the definition of $\mathsf{X}_\varepsilon$ from (50), we note that

$$\mathtt{T}_A = \sum_{h=1}^{H} Z_{t,h} \mathbf{1}\{\pi_t(x_{t,h}) \ne \pi^\star(x_{t,h}), x_{t,h} \notin \mathsf{X}_\varepsilon\}$$

$$= \sum_{h=1}^{H} Z_{t,h} \mathbf{1}\{\pi_t(x_{t,h}) \ne \pi^\star(x_{t,h}), \mathtt{Margin}(f_h^\star(x_{t,h})) > \varepsilon\}$$

$$\le \sum_{h=1}^{H} Z_{t,h} \mathbf{1}\{\pi^\star(x_{t,h}) \ne \pi_t(x_{t,h}), \phi(f_h^\star(x_{t,h}))[\pi^\star(x_{t,h})] - \phi(f_h^\star(x_{t,h}))[\pi_t(x_{t,h})] \ge \varepsilon\}$$

$$\le \sum_{h=1}^{H} Z_{t,h} \mathbf{1}\{\phi(f_h^\star(x_{t,h}))[\pi^\star(x_{t,h})] - \phi(f_h^\star(x_{t,h}))[\pi_t(x_{t,h})] \ge \varepsilon\},$$

where in the second last line we used the definition of $\mathtt{Margin}(f_h^\star(x_{t,h}))$ along with the fact that $\pi^\star(x_{t,h}) \ne \pi_t(x_{t,h})$. Using the relation in Lemma 12 for the term inside the indicator, we can further bound the above as

$$\mathtt{T}_A \le \sum_{h=1}^{H} Z_{t,h} \mathbf{1}\{2\gamma\|f_h^\star(x_{t,h}) - f_{t,h}(x_{t,h})\| \ge \varepsilon\}$$

$$\le \frac{4\gamma^2}{\varepsilon^2} \sum_{h=1}^{H} Z_{t,h} \|f_h^\star(x_{t,h}) - f_{t,h}(x_{t,h})\|^2,$$

where in the second inequality we used: $\mathbf{1}\{a \ge b\} \le a^2/b^2$ for any $a, b \ge 0$.

- *Bound on term* $\mathtt{T}_B$. Before delving into the proof, first note that Lemma 19-($b$) implies that

$$\|f_h^\star(x_{t,h}) - f_{t,h}(x_{t,h})\| \le \Delta_{t,h}(x_{t,h}). \tag{53}$$

Next, note that

$$\mathtt{T}_B = \sum_{h=1}^{H} \bar{Z}_{t,h} \mathbf{1}\{\pi_t(x_{t,h}) \ne \pi^\star(x_{t,h})\}$$

$$= \sum_{h=1}^{H} \mathbf{1}\{\mathtt{Margin}(f_{t,h}(x_{t,h})) > 2\gamma\Delta_{t,h}(x_{t,h}), \pi_t(x_{t,h}) \ne \pi^\star(x_{t,h})\},$$

where in the last line we just plugged in the query condition under which $Z_{t,h} = 0$. However note that the above two conditions inside the indicator imply that

$$2\gamma\Delta_{t,h}(x_{t,h}) < \mathtt{Margin}(f_{t,h}(x_{t,h}))$$

$$\le \phi(f_{t,h}(x_{t,h}))[\pi_t(x_{t,h})] - \phi(f_{t,h}(x_{t,h}))[\pi^\star(x_{t,h})]$$

$$\le 2\gamma\|f_{t,h}(x_{t,h}) - f_h^\star(x_{t,h})\|,$$

where the second line uses the definition of $\mathtt{Margin}(\cdot)$ and the fact that $\pi_t(x_{t,h}) \ne \pi^\star(x_{t,h})$, the last line is due to Lemma 12 . Thus,

$$\mathtt{T}_B \le \sum_{h=1}^{H} \mathbf{1}\{\|f_{t,h}(x_{t,h}) - f_h^\star(x_{t,h})\| > \Delta_{t,h}(x_{t,h})\},$$

but the conditions inside the indicator in the above contradicts (53) (which holds with probability $1 - \delta$). Thus, with probability at least $1 - \delta$,

$$\mathtt{T}_B = 0. \tag{54}$$

Plugging in the bounds on $\mathtt{T}_A$ and $\mathtt{T}_B$ in (52), we get that with probability at least $1 - \delta$,

$$R(\tau_t^{\pi^\star}) - R(\tau_t^{\pi_t}) \le 2H \sum_{h=1}^{H} \mathbf{1}\{x_{t,h}^{\pi^\star} \in \mathtt{X}_\varepsilon\} + \frac{8H\gamma^2}{\varepsilon^2} \sum_{h=1}^{H} Z_{t,h} \|f_h^\star(x_{t,h}) - f_{t,h}(x_{t,h})\|^2. \qquad (55)$$

**Step 2: Bound on total regret.** Using the bound in (55) for each round $t$, we get that

$$\begin{aligned}
\mathrm{Reg}_T &= \sum_{t=1}^{T} \Big( R(\tau_t^{\pi^\star}) - R(\tau_t^{\pi_t}) \Big) \\
&\le 2H \sum_{h=1}^{H} \sum_{t=1}^{T} \mathbf{1}\{x_{t,h}^{\pi^\star} \in \mathtt{X}_\varepsilon\} + \frac{8H\gamma^2}{\varepsilon^2} \sum_{h=1}^{H} \sum_{t=1}^{T} Z_{t,h} \|f_h^\star(x_{t,h}) - f_{t,h}(x_{t,h})\|^2 \\
&\le 2H \sum_{h=1}^{H} T_{\varepsilon,h} + \frac{8H\gamma^2}{\varepsilon^2} \sum_{h=1}^{H} \Psi_\delta^{\ell_\phi}(\mathcal{F}_h, T), \qquad (56)
\end{aligned}$$

where in the last line we use the definition of $T_{\varepsilon,h}$, and plug in the bound in Lemma 19. Recall that $T_{\varepsilon,h}$ denotes the number of times when the comparator policy $\pi^\star$ enters the region $\mathtt{X}_\varepsilon$ of states of small expert margin. Using the form of $\Psi_\delta^{\ell_\phi}(\mathcal{F}_h, T)$ and ignoring $\log$ factors and constants, we get

$$\mathrm{Reg}_T = \widetilde{\mathcal{O}}\bigg( H \sum_{h=1}^{H} T_{\varepsilon,h} + \frac{H\gamma^2}{\lambda\varepsilon^2} \sum_{h=1}^{H} \mathrm{Reg}^{\ell_\phi}(\mathcal{F}_h; T) + \frac{H\gamma^2}{\lambda^2\varepsilon^2} \log(1/\delta) \bigg).$$

Since $\varepsilon$ is a free parameter above, the final bound follows by taking $\inf$ over all feasible $\varepsilon$.

### F.3.3 Total Number of Queries

Let $N_T$ denote the total number of expert queries made by the learner within $T$ rounds of interaction (with $H$ steps per round). For $t \le T$, let $h_t$ denote the first timestep at which $Z_{t,h_t} = 1$ at round $t$. Additionally, let $\varepsilon > 0$ be a free parameter. Thus, we have that

$$\begin{aligned}
N_T &= \sum_{t=1}^{T} \sum_{h=1}^{H} Z_{t,h} \qquad (57) \\
&\le H \sum_{t=1}^{T} Z_{t,h_t} \\
&= H \sum_{t=1}^{T} Z_{t,h_t} \mathbf{1}\{x_{t,h_t} \in \mathtt{X}_\varepsilon\} + H \sum_{t=1}^{T} Z_{t,h_t} \mathbf{1}\{x_{t,h_t} \notin \mathtt{X}_\varepsilon\} \\
&\le H \sum_{t=1}^{T} Z_{t,h_t} \mathbf{1}\{x_{t,h_t} \in \mathtt{X}_\varepsilon\} + H \sum_{t=1}^{T} \sum_{h=1}^{H} Z_{t,h} \mathbf{1}\{x_{t,h} \notin \mathtt{X}_\varepsilon\} \\
&= H \sum_{t=1}^{T} Z_{t,h_t} \mathbf{1}\{x_{t,h_t} \in \mathtt{X}_\varepsilon\} + H \sum_{t=1}^{T} \sum_{h=1}^{H} Z_{t,h} \mathbf{1}\{x_{t,h} \notin \mathtt{X}_\varepsilon, \Delta_{t,h}(x_{t,h}) \le \frac{\varepsilon}{4\gamma}\} \\
&\qquad\qquad\qquad\qquad\qquad + H \sum_{t=1}^{T} \sum_{h=1}^{H} Z_{t,h} \mathbf{1}\{x_{t,h} \notin \mathtt{X}_\varepsilon, \Delta_{t,h}(x_{t,h}) > \frac{\varepsilon}{4\gamma}\} \\
&= \mathtt{T}_C + H\mathtt{T}_D + H\mathtt{T}_E,
\end{aligned}$$

where $\mathtt{T}_C$, $\mathtt{T}_D$ and $\mathtt{T}_E$ are the first, second and the third term respectively in the previous line. We bound them separately below:

- *Bound on $\mathtt{T}_C$.* Fix any $t \le T$, and note that

$$\begin{aligned}
(\mathtt{T}_C)_t &= HZ_{t,h_t} \mathbf{1}\{x_{t,h_t} \in \mathtt{X}_\varepsilon\} \\
&= HZ_{t,h_t} \mathbf{1}\{x_{t,h_t} \in \mathtt{X}_\varepsilon\} \mathbf{1}\{\forall h < h_t : \pi^\star(x_{t,h}) = \pi_t(x_{t,h})\} \\
&\qquad\qquad + HZ_{t,h_t} \mathbf{1}\{x_{t,h_t} \in \mathtt{X}_\varepsilon\} \mathbf{1}\{\exists h < h_t : \pi^\star(x_{t,h}) \ne \pi_t(x_{t,h})\}. \qquad (58)
\end{aligned}$$

For the second term, note that

$$Z_{t,h_t} \mathbf{1}\{x_{t,h_t} \in \mathtt{X}_\varepsilon\} \mathbf{1}\{\exists h < h_t : \pi^\star(x_{t,h}) \ne \pi_t(x_{t,h})\} \le \sum_{h=1}^{h_t} Z_{t,h_t} \mathbf{1}\{\pi^\star(x_{t,h}) \ne \pi_t(x_{t,h})\}$$

$$\leq \sum_{h=1}^{h_t} Z_{t,h_t} \bar{Z}_{t,h} \mathbf{1}\{\pi^\star(x_{t,h}) \neq \pi_t(x_{t,h})\}$$

$$\leq \sum_{h=1}^{h_t} \bar{Z}_{t,h} \mathbf{1}\{\pi^\star(x_{t,h}) \neq \pi_t(x_{t,h})\}$$

where in second inequality above, we used the fact that $Z_{t,h} = 0$ (and thus $\bar{Z}_{t,h} = 1$) for all $h \leq h_t$, by the definition of $h_t$. However note that the right hand side in the last inequality is equivalent to the term $\mathtt{T}_B$ defined above (where sum is now till $h_t$ instead of $H$). Thus, using the bound in (54) in the above, we immediately get that

$$Z_{t,h_t} \mathbf{1}\{x_{t,h_t} \in \mathtt{X}_\varepsilon\} \mathbf{1}\{\exists h < h_t : \pi^\star(x_{t,h}) \neq \pi_t(x_{t,h})\} = 0.$$

For the first term in (58), using the condition that $\pi^\star(x_{t,h}) = \pi_t(x_{t,h})$ for all $h \leq h_t$, we get that $x_{t,h} = x_{t,h}^{\pi^\star}$ and thus

$$HZ_{t,h_t} \mathbf{1}\{x_{t,h_t} \in \mathtt{X}_\varepsilon\} \mathbf{1}\{\forall h \leq h_t : \pi^\star(x_{t,h}) = \pi_t(x_{t,h})\} \leq HZ_{t,h_t} \mathbf{1}\{x_{t,h_t}^{\pi^\star} \in \mathtt{X}_\varepsilon\}$$

$$\leq H\mathbf{1}\{x_{t,h_t}^{\pi^\star} \in \mathtt{X}_\varepsilon\}$$

$$\leq H \sum_{h=1}^{H} \mathbf{1}\{x_{t,h}^{\pi^\star} \in \mathtt{X}_\varepsilon\}.$$

Gathering the two terms above, and plugging in the definition of $T_{\varepsilon,h}$, we get that

$$\mathtt{T}_C \leq H \sum_{h=1}^{H} \sum_{t=1}^{T} \mathbf{1}\{x_{t,h}^{\pi^\star} \in \mathtt{X}_\varepsilon\} = H \sum_{h=1}^{H} T_{\varepsilon,h}.$$

- *Bound on $\mathtt{T}_D$.* Using the definition of the set $\mathtt{X}_\varepsilon$ and $Z_{t,h}$, we note that

$$(\mathtt{T}_D)_t = \sum_{h=1}^{H} Z_{t,h} \mathbf{1}\{x_{t,h} \notin \mathtt{X}_\varepsilon, \Delta_{t,h}(x_{t,h}) \leq \frac{\varepsilon}{4\gamma}\}$$

$$= \sum_{h=1}^{H} \mathbf{1}\{\mathtt{Margin}(f_{t,h}(x_{t,h})) \leq 2\gamma\Delta_{t,h}(x_{t,h}), \mathtt{Margin}(f_h^\star(x_{t,h})) > \varepsilon, \Delta_{t,h}(x_{t,h}) \leq \frac{\varepsilon}{4\gamma}\}$$

(59)

Recall that Lemma 19 implies that with probability at least $1 - \delta$,

$$\|f_h^\star(x_{t,h}) - f_{t,h}(x_{t,h})\| \leq \Delta_{t,h}(x_{t,h}),$$

using which with Lemma 13 implies that

$$\mathtt{Margin}(f_h^\star(x_{t,h})) \leq \mathtt{Margin}(f_{t,h}(x_{t,h})) + 2\gamma\|f_h^\star(x_{t,h}) - f_{t,h}(x_{t,h})\|$$

$$\leq \mathtt{Margin}(f_{t,h}(x_{t,h})) + 2\gamma\Delta_{t,h}(x_{t,h}).$$

Using the above bound with the conditions in (59) implies that

$$(\mathtt{T}_D)_t = \sum_{h=1}^{H} \mathbf{1}\{\mathtt{Margin}(f_h^\star(x_{t,h})) \leq 4\gamma\Delta_{t,h}(x_{t,h}), \mathtt{Margin}(f_h^\star(x_{t,h})) > \varepsilon, \Delta_{t,h}(x_{t,h}) \leq \frac{\varepsilon}{4\gamma}\}$$

$$= \sum_{h=1}^{H} \mathbf{1}\{\mathtt{Margin}(f_h^\star(x_{t,h})) \leq \varepsilon, \mathtt{Margin}(f_h^\star(x_{t,h})) > \varepsilon\}$$

$$= 0,$$

where the last equality holds because the two conditions in the indicator in the previous line can never occur simultaneously.

- *Bound on $\mathtt{T}_E$.* Note that

$$\mathtt{T}_E = \sum_{t=1}^{T} \sum_{h=1}^{H} Z_{t,h} \mathbf{1}\{x_{t,h} \notin \mathtt{X}_\varepsilon, \Delta_{t,h}(x_{t,h}) > \frac{\varepsilon}{4\gamma}\}$$

$$\leq \sum_{h=1}^{H} \sum_{t=1}^{T} Z_{t,h} \mathbf{1}\{\Delta_{t,h}(x_{t,h}) > \frac{\varepsilon}{4\gamma}\}.$$

An application of Lemma 20 in the above for each $h \leq H$ implies that

$$\mathtt{T}_E \leq \sum_{h=1}^{H} \frac{320\gamma^2 \Psi_\delta^{\ell_\phi}(\mathcal{F}_h, T)}{\varepsilon^2} \cdot \mathfrak{E}(\mathcal{F}_h, \frac{\varepsilon}{8\gamma}; f_h^\star).$$

Gathering the bound above, we get that

$$N_T \leq H \sum_{h=1}^{H} T_{\varepsilon,h} + \frac{320H\gamma^2}{\varepsilon^2} \sum_{h=1}^{H} \Psi_\delta^{\ell_\phi}(\mathcal{F}_h, T) \cdot \mathfrak{E}(\mathcal{F}_h, \frac{\varepsilon}{8\gamma}; f_h^\star).$$

Plugging in the form of $\Psi_\delta^{\ell_\phi}(\mathcal{F}_h, T)$ and ignoring $\log$ factors and constants, we get that

$$N_T \leq \widetilde{\mathcal{O}}\bigg( H \sum_{h=1}^{H} T_{\varepsilon,h} + \frac{H\gamma^2}{\lambda\varepsilon^2} \sum_{h=1}^{H} \mathrm{Reg}^{\ell_\phi}(\mathcal{F}_h; T) \cdot \mathfrak{E}(\mathcal{F}_h, \varepsilon/8\gamma; f_h^\star) + \frac{H\gamma^2}{\lambda^2\varepsilon^2} \log(1/\delta) \bigg).$$

Notice that $\varepsilon$ is a free parameter above so the final bound follows by taking $\inf$ over all feasible $\varepsilon$. s

## F.4 Proof for the Stochastic Setting

Algorithm 2 considers arbitrary deterministic dynamics $\{\{\mathbb{T}_{t,h}\}_{h \leq H}\}_{t \leq T}$. When the underlying dynamics $\mathscr{T}$ is stochastic, we can simply simulate Algorithm 2, where we set $\{\mathbb{T}_{t,h}\}_{h \leq H} = \mathscr{T}(\cdot; \iota_t)$ where $\iota_t$ is drawn i.i.d. for every $t \leq T$. In the following, we provide regret and query complexity bounds for stochastic dynamics.

**Regret bound.** Note that Theorem 4 bounds the difference of cumulative rewards of trajectories drawn using the policies $\pi_t$ and $\pi^\star$ on the adversarially chosen deterministic dynamics $\{\mathbb{T}_{t,h}\}_{h=1}^{H}$ respectively. In particular, we bound

$$\mathrm{Reg}_T = \sum_{t=1}^{T} \Big( R(\tau_t^{\pi^\star}) - R(\tau_t^{\pi_t}) \Big).$$

On the other hand, when the dynamics is stochastic, we aim to bound the gap between the expected values $V^{\pi^\star} - V^{\pi_t}$ obtained under the stochastic dynamics $\mathscr{T}$. We obtain this bound by pushing all the stochasticity into the choice of random seed $\iota$. Fix any $t \leq T$, and consider the deterministic dynamics $(\mathbb{T}_{t,1}, \ldots, \mathbb{T}_{t,H})$ obtained by setting the random seed to be $\iota_t$ in the stochastic dynamics $\mathscr{T}$, i.e. $(\mathbb{T}_{t,1}, \ldots, \mathbb{T}_{t,H}) \coloneqq \mathscr{T}(\,; \iota_t)$. Thus, for any policy $\pi$

$$V^\pi = \mathbb{E}_{\iota_t}[R(\tau_t^\pi) \mid (\mathbb{T}_{t,1}, \ldots, \mathbb{T}_{t,H}) = \mathscr{T}(\,; \iota_t)].$$

In the following, we will bound the difference in the value function $V^\pi - V^{\pi_t}$, by appealing to the regret bound in the proof of Theorem 4 using appropriate concentration inequalities. First, recall that in Algorithm 2, the dynamics $\{\mathbb{T}_{t,h}\}_{h \leq H}$ is chosen before the round $t$, and that the policy $\pi_t$ only depends on the interaction till round $t-1$. Thus,

$$\sum_{t=1}^{T} V^\pi - V^{\pi_t} = \sum_{t=1}^{T} \mathbb{E}_{\iota_t}\Big[ R(\tau_t^\pi) - R(\tau_t^{\pi^\star}) \Big]$$

$$\leq \sum_{t=1}^{T} \mathbb{E}_{\iota_t}\bigg[ 2H \sum_{h=1}^{H} \mathbf{1}\{x_{t,h}^{\pi^\star} \in \mathsf{X}_\varepsilon\} + 2H \sum_{h=1}^{H} \mathbf{1}\{\pi_t(x_{t,h}^{\pi_t}) \neq \pi^\star(x_{t,h}^{\pi_t}), x_{t,h}^{\pi_t} \notin \mathsf{X}_\varepsilon\} \bigg],$$

where the last holds due to Lemma 18 and the set $\mathsf{X}_\varepsilon$ is defined in (50). An application of Lemma 4 in the above implies that with probability at least $1 - \delta$,

$$\sum_{t=1}^{T} V^\pi - V^{\pi_t} \leq 4H \sum_{t=1}^{T} \sum_{h=1}^{H} \mathbf{1}\{x_{t,h}^{\pi^\star} \in \mathsf{X}_\varepsilon\} + 4H \sum_{t=1}^{T} \sum_{h=1}^{H} \mathbf{1}\{\pi_t(x_{t,h}^{\pi_t}) \neq \pi^\star(x_{t,h}^{\pi_t}), x_{t,h}^{\pi_t} \notin \mathsf{X}_\varepsilon\} + 32H^2 \log(2/\delta).$$

The rest of the proof is identical to the proof of Theorem 4 from (51) onwards. They query complexity can be similarly computed.

We next provide the proofs for learning from multiple experts.

## F.5  Proof of Theorem 5

---

**Algorithm 4** InteRActiVe ImitatiOn Learning VIa Active Queries to $M$ Experts (RAVIOLI–M)

---

**Input:** Parameters $\delta, \gamma, \lambda, T$, function classes $\{\mathcal{F}_h^m\}_{h \leq H, m \leq M}$, online oracles $\{\mathsf{Oracle}_h^m\}_{h \leq H, m \leq M}$
    w.r.t. $\ell_\phi$.

1: Set $\Psi_\delta^{\ell_\phi}(\mathcal{F}_h^m, T) = \frac{4}{\lambda}\mathrm{Reg}^{\ell_\phi}(\mathcal{F}_h^m; T) + \frac{112}{\lambda^2}\log(4MH\log^2(T)/\delta)$.

2: Compute $f_{1,h}^m = \mathsf{Oracle}_{1,h}(\varnothing)$ for each $h \in [H]$ and $m \in [M]$.

3: **for** $t = 1$ to $T$ **do**

4:    Nature chooses the state $x_{t,1}$.

5:    **for** $h = 1$ to $H$ **do**

6:        Define $F_{t,h}^m(x) := [f_{t,h}^1(x), \ldots, f_{t,h}^M(x)]$.

7:        Learner plays $\widehat{y}_{t,h} = \mathsf{SelectAction}(F_{t,h}(x_{t,h}))$.

8:        Learner transitions to the next state in this round $x_{t,h+1} \leftarrow \mathbb{T}_{t,h}(x_{t,h}, \widehat{y}_{t,h})$.

9:        For each $m \in [M]$, learner computes

$$\Delta_{t,h}^m(x_{t,h}) := \max_{f \in \mathcal{F}_h^m} \|f(x_{t,h}) - f_{t,h}^m(x_{t,h})\|$$

$$\text{s.t.} \quad \sum_{s=1}^{t-1} Z_{s,h}\|f(x_{s,h}) - f_{s,h}^m(x_{s,h})\|^2 \leq \Psi_\delta^{\ell_\phi}(\mathcal{F}_h^m, T). \qquad (60)$$

        and defines $\vec{\Delta}_{t,h}(x_{t,h}) = [\Delta_{t,h}^1(x_{t,h}), \ldots, \Delta_{t,h}^M(x_{t,h})]$.

10:      Learner decides whether to query: $Z_{t,h} = \mathsf{Query}(F_{t,h}(x_{t,h}), \vec{\Delta}_{t,h}(x_{t,h}))$

11:      **if** $Z_{t,h} = 1$ **then**

12:          **for** $m = 1$ to $M$ **do**

13:              Learner queries expert $m$ for its label $y_{t,h}^m$ for $x_{t,h}$.

14:              $f_{t+1,h}^m \leftarrow \mathsf{Oracle}_{t+1,h}^m(\{x_{t,h}, y_{t,h}\})$

15:      **else**

16:          $f_{t+1,h}^m \leftarrow f_{t,h}^m$ for each $m \in [M]$.

---

We first discuss the setup and the relevant notation. We are in the imitation learning setup introduced in Appendix F.3. In particular, the learner interacts with $M$ experts in $T$ episodes/rounds, each consisting of $H$ timesteps. For any $h \leq H$, each of the $M$ experts have a ground truth model $f_h^{\star,m}$ respectively. Given the context $x_{t,h}$ for time step $h$ in round $t$, the learner plays the actions $\widehat{y}_{t,h} \in [K]$, and can additionally choose to query the experts (by setting $Q_{t,h} = 1$) to receive noisy feedback $y_{t,h}^m \sim \phi(f_h^{\star,m}(x_t))$ from each expert $m \in [M]$. After playing the chosen action $\widehat{y}_{t,h}$, the learner then transitions to state $x_{t,h+1} \leftarrow \mathbb{T}_{t,h}(x_{t,h}, \widehat{y}_{t,h})$ where $\{\mathbb{T}_{t,h}\}_{h \in [H]}$ is a sequence of deterministic dynamics (unknown to the learner).

In Algorithm 4, for any round $t \leq T$ and $h \leq H$:

- The aggregation function $\mathscr{A} : \mathbb{R}^{K \times M} \mapsto \mathbb{R}^K$, known to the learner, maps the predictions of the estimated experts to distributions over actions. Some illustrative examples are given below to illustrate the generality of our setup:

  (*a*) *Random aggregation:* Given a state $x_h$, the aggregation rule chooses an expert uniformly at random and returns the label $y_h$ sampled from its model. In particular,

  $$y_h \sim \phi(f_h^{\star,\widetilde{m}}(x_h)), \qquad \text{where} \qquad \widetilde{m} \sim \mathrm{Uniform}([M]).$$

  Here, the distribution $\mathscr{A}\big(\phi(f_h^{\star,1}(x_h)), \ldots, \phi(f_h^{\star,M}(x_h))\big) = \frac{1}{M}\sum_{m=1}^M \phi(f^{\star,m}(x_h))$.

  (*b*) *Majority label*: $\mathscr{A}$ is deterministic. Given a state $x_t$, the aggregation rule chooses the label $y_h \in [K]$ which is the top preference for the majority of the experts. In particular,

  $$y_h = \mathscr{A}\big(\phi(f_h^{\star,1}(x_h)), \ldots, \phi(f_h^{\star,M}(x_h))\big) = \underset{k \in [K]}{\mathrm{argmax}} \sum_{m=1}^M \mathbf{1}\{k = \underset{\widetilde{k} \in [K]}{\mathrm{argmax}}\, \phi(f_h^{\star,m}(x_h)[\widetilde{k}])\}.$$

($c$) *Majority-of-confident-experts:* This aggregation rule is also deterministic, and was first introduced in Dekel et al. [2012]. Given a state $x_t$, the aggregation rule chooses the label $y_h \in [K]$ which is the top preference for the majority of the $\rho$-*confident* experts on $x_h$ i.e. the experts whose margin on $x_h$ is larger than $\rho$. In particular,

$$y_h = \mathscr{A}\big(\phi(f_h^{\star,1}(x_h)), \dots, \phi(f_h^{\star,M}(x_h))\big)$$

$$= \operatorname*{argmax}_{k \in [K]} \sum_{m=1}^{M} \mathbf{1}\{k = \operatorname*{argmax}_{\widetilde{k} \in [K]} \phi(f_h^{\star,m}(x_h)[\widetilde{k}]) \text{ and } \mathtt{Margin}(\phi(f_h^{\star,m}(x_h)) > \rho)\},$$

where $\mathtt{Margin}(f_h^{\star,m}(x_h) > \rho) = \max_{k_1}\big(\phi(f_h^{\star,m}(x_h))[k_1] - (\max_{k_2 \neq k_1} \phi(f_h^{\star,m}(x_h))[k_2])\big)$. This aggregation rule is useful when there may be many experts that give equal weights to the top and the second-to-top coordinates w.r.t. their respective models, and hence can not be confidently accounted for in the majority rule. Furthermore, instead of choosing the majority label, similar to Dekel et al. [2012], one can also return the label sampled according to a uniform distribution over $\rho$−confident experts.

- The function $\mathtt{SelectAction}: \mathbb{R}^{K \times M} \mapsto [K]$ chooses the action to play at round $t$, and is defined as:

$$\mathtt{SelectAction}(F_{t,h}(x_{t,h})) = \operatorname*{argmax}_{k} \mathscr{A}(\phi(F_{t,h}(x_{t,h})))[k], \tag{61}$$

where $F_{t,h}(x_{t,h})) = [f_{t,h}^1(x_{t,h})), \dots, f_{t,h}^M(x_{t,h}))]$, and $\phi$ denotes the link-function given in (2).

- Our goal in Algorithm 4 is to compete with the policy $\pi^\star$ defined such that for any $x \in \mathcal{X}_h$,

$$\pi^\star(x) = \mathtt{SelectAction}(F_h^\star(x)) \tag{62}$$

where $F_h^\star(x) = [f_h^{\star,1}(x), \dots, f_h^{\star,1}(x)] \in \mathbb{R}^{K \times M}$.

- Given the context $x_{t,h}$ and the function $F_{t,h}: \mathcal{X} \mapsto \mathbb{R}^{K \times M}$, the learner decided whether to query via $Z_{t,h} = \mathtt{Query}(F_{t,h}(x_{t,h}), \vec{\Delta}_{t,h}(x_{t,h}))$ where we define the function $\mathtt{Query}: \mathbb{R}^{K \times M} \times \mathbb{R}^M$ as

$$\mathtt{Query}(U; \vec{\varepsilon}) := \sup_{V \in \mathbb{R}^{K \times M}} \mathbf{1}\{\mathtt{SelectAction}(U) \neq \mathtt{SelectAction}(V)\}$$

$$\text{s.t.} \quad \|U[:,m] - V[:,m]\|_2 \leq \vec{\varepsilon}[m] \qquad \forall m \leq M. \tag{63}$$

At round $t$, the learner interactions with transition dynamics $\{\mathbb{T}_{t,h}\}_{h \leq H}$ and collects data. Without loss of generality, we assume that the learner always starts from the state $x_{t,1}$. We next recall the interaction at round $t$:

- The learner collects data using the policy $\pi_t$, defined such that

$$\pi_t(x) = \mathtt{SelectAction}(f_{t,h}(x_{t,h})).$$

for any $h \leq H$, and state $x \in \mathcal{X}_h$.

- For any policy $\pi$, we use the notation $\tau_t^\pi$ to denote the (counterfactual) trajectory that would have been generated by running $\pi$ on the deterministic dynamics $\{\mathbb{T}_{t,h}\}_{h \leq H}$ with the start state $x_{t,1}$, i.e.

$$\tau_t^\pi = \big\{x_{t,1}^\pi, \pi(x_{t,1}^\pi), \dots, x_{t,H}^\pi, \pi(x_{t,H}^\pi)\big\}, \tag{64}$$

where $x_{t,1}^\pi = x_{t,1}$ and $x_{t,h+1}^\pi = \mathbb{T}_{t,h}(x_{t,h}^\pi, \pi(x_{t,h}^\pi))$.

- For any trajectory $\tau = \{x_1, a_1, \dots, x_H, a_H\}$, we define the total return

$$R(\tau) = \sum_{h=1}^{H} r(x_h, a_h). \tag{65}$$

The goal of the learner is to minimize its regret which is given by

$$\text{Reg}_T = \sum_{t=1}^{T} R(\tau_t^{\pi^\star}) - \sum_{t=1}^{T} R(\tau_t^{\pi_t}). \tag{66}$$

Finally, our bounds depend on the notation of margin defined w.r.t. the function $\texttt{Query}$ defined above. In particular, for a sequence of contexts $\{\{T_{t,h}\}_{h \leq H}\}_{t \leq T}$, we define $T_{\varepsilon,h}$ as

$$T_{\varepsilon,h} = \sum_{t=1}^{T} \mathbf{1}\{\texttt{Query}(F_h^\star(x_{t,h}^{\pi^\star}), \varepsilon \vec{\mathbb{1}}) = 1\}. \tag{67}$$

In the above we count for the number of time steps when the counterfactual states $\{x_{t,h}^{\pi^\star}\}_{t \leq T}$, reached under the given dynamics if we had executed $\pi^\star$, are within the $\varepsilon$-margin region. Note that even though the observed states $\{\{x_{t,h}\}_{h \leq H}\}_{t \leq T}$ are such that $\sum_{t=1}^{T} \mathbf{1}\{\texttt{Query}(F_h^\star(x_{t,h}), \varepsilon \vec{\mathbb{1}}) = 1\}$ is large, we would not pay for this in our margin term $T_{\varepsilon,h}$.

### F.5.1 Supporting Technical Results

**Lemma 21.** *With probability at least $1 - \delta$, for any $m \leq M$, and $t \leq T$ and $h \leq H$, the function $f^{\star,m}$ satisfies*

$(a)$ $\sum_{s=1}^{t-1} Z_{s,h} \|f_h^{\star,m}(x_{s,h}) - f_{s,h}^m(x_{s,h})\|^2 \leq \Psi_\delta^{\ell_\phi}(\mathcal{F}_h^m, T)$,

$(b)$ $\|f_h^{\star,m}(x_{t,h}) - f_{t,h}^m(x_{t,h})\| \leq \Delta_{t,h}^m(x_{t,h})$,

*where $\Psi_\delta^{\ell_\phi}(\mathcal{F}_h^m, T) = \frac{4}{\lambda}\text{Reg}^{\ell_\phi}(\mathcal{F}_h^m; T) + \frac{112}{\lambda^2}\log(4MH\log^2(T)/\delta)$.*

*Proof.*

$(a)$ We first note that we do not query oracle when $Z_{s,h} = 0$, and thus we can ignore the time steps for which $Z_{s,h} = 0$. Hence, for each $h \in [H]$ and $m \in [M]$, applying Lemma 5 yields

$$\sum_{s=1}^{t-1} Z_{s,h} \|f_h^{\star,m}(x_{s,h}) - f_{s,h}(x_{s,h})\|^2 \leq \frac{4}{\lambda}\text{Reg}^{\ell_\phi}(\mathcal{F}_h^m; T) + \frac{112}{\lambda^2}\log(4\log^2(T)/\delta)$$

for all $t \leq T$. Then, we take the union bound for all $h \in [H]$ and $m \in [M]$, which completes the proof.

$(b)$ The second part follows from using part-(a) along with the definition in (60).

$\square$

The next lemma bound the number of times when $\Delta_{t,h}^m(x_{t,h}) \geq \zeta$, and we query. Note that Lemma 22 holds even if the sequence $\{x_{t,h}\}_{t \leq T}$ was adversarially generated.

**Lemma 22.** *Let $f^{\star,m}$ satisfy Lemma 21, and let $\Delta_{t,h}^m(x_{t,h})$ be defined in Algorithm 4. Suppose Algorithm 4 is run on the data sequence $\{x_{t,h}\}_{t \leq 1}$, and let $Z_{t,h}$ be defined in line 10. Then, for any $\zeta > 0$, with probability at least $1 - M\delta$, for any $m \in [M]$, and $h \leq H$,*

$$\sum_{t=1}^{T} Z_{t,h}\mathbf{1}\{\Delta_{t,h}^m(x_t) \geq \zeta\} \leq \frac{20\Psi_\delta^{\ell_\phi}(\mathcal{F}_h^m, T)}{\zeta^2} \cdot \mathfrak{E}(\mathcal{F}_h^m, \zeta/2; f_h^{\star,m}),$$

*where $\mathfrak{E}$ denotes the eluder dimension given in Definition 1.*

*Proof.* The proof is identical to the proof of Lemma 11 where we handle each $m \in [M]$ and $h \in [H]$ separately, and substitute the corresponding bounds for $f_h^{\star,m}$ via Lemma 21 (instead of using Lemma 10). We skip the proof for conciseness. $\square$

### F.5.2 Regret Bound

Suppose the trajectories at round $t$ are generated using the deterministic dynamics $\{\mathbb{T}_{t,1}, \dots, \mathbb{T}_{t,H}\} = \mathscr{T}(\cdot\,;\, \iota_t)$ where $\iota_t$ denotes the random seed that captures all of the stochasticity at round $t$ [10].

Recall that the policies $\pi^\star$ and $\pi_t$ such that for any $h \leq H$ and $x \in \mathcal{X}_h$, $\pi^\star(x) = \texttt{SelectAction}(F_h^\star(x))$, and, $\pi_t(x) = \texttt{SelectAction}(F_{t,h}(x))$. Note that Algorithm 4 collects trajectories using the policy $\pi_t$ at round $t$. Thus, we have

$$x_{t,h}^{\pi_t} = x_{t,h}, \tag{68}$$

where $x_{t,h}$ denotes the state at time step $h$ in round $t$ of Algorithm 4. Finally, let $\varepsilon > 0$ be a free parameter. We start with the bound on the regret at time $t$.

**Step 1: Bounding the difference in cumulative return at round $t$.** Fix any $t \leq T$, and let $\tau_t^{\pi_t}$ and $\tau_t^{\pi^\star}$ denote the trajectories that would have been sampled using the policies $\pi_t$ and the policy $\pi^\star$ at round $t$. Furthermore, define the set $\mathtt{X}_\varepsilon$ as

$$\mathtt{X}_\varepsilon := \bigcup_{h=1}^{H} \{x \in \mathcal{X}_h \mid \texttt{Query}(F_h^\star(x), \varepsilon \vec{\mathbb{1}}) = 1\} \tag{69}$$

Using Lemma 18 for the policies $\pi_t$ and $\pi^\star$, and the set $\mathtt{X}_\varepsilon$ defined above, we get that

$$
\begin{aligned}
R(\tau_t^{\pi^\star}) - R(\tau_t^{\pi_t}) &\leq 2H \sum_{h=1}^{H} \mathbf{1}\{x_{t,h}^{\pi^\star} \in \mathtt{X}_\varepsilon\} + 2H \sum_{h=1}^{H} \mathbf{1}\{\pi_t(x_{t,h}^{\pi_t}) \neq \pi^\star(x_{t,h}^{\pi_t}), x_{t,h}^{\pi_t} \notin \mathtt{X}_\varepsilon\} \\
&= 2H \sum_{h=1}^{H} \mathbf{1}\{x_{t,h}^{\pi^\star} \in \mathtt{X}_\varepsilon\} + 2H \sum_{h=1}^{H} \mathbf{1}\{\pi_t(x_{t,h}) \neq \pi^\star(x_{t,h}), x_{t,h} \notin \mathtt{X}_\varepsilon\} \\
&= 2H \sum_{h=1}^{H} \mathbf{1}\{x_{t,h}^{\pi^\star} \in \mathtt{X}_\varepsilon\} + 2H \sum_{h=1}^{H} Z_{t,h} \mathbf{1}\{\pi_t(x_{t,h}) \neq \pi^\star(x_{t,h}), x_{t,h} \notin \mathtt{X}_\varepsilon\} \\
&\qquad\qquad\qquad\qquad + 2H \sum_{h=1}^{H} \bar{Z}_{t,h} \mathbf{1}\{\pi_t(x_{t,h}) \neq \pi^\star(x_{t,h}), x_{t,h} \notin \mathtt{X}_\varepsilon\} \\
&= 2H \sum_{h=1}^{H} \mathbf{1}\{x_{t,h}^{\pi^\star} \in \mathtt{X}_\varepsilon\} + 2H \mathtt{T}_A + 2H \mathtt{T}_B, \tag{70}
\end{aligned}
$$

where the second line is obtained by using the relation (68) in the second line. The last line simply defines $\mathtt{T}_A$ and $\mathtt{T}_B$ to be the second and the third term in the previous line, respectively, without the $2H$ multiplicative factor. We bound these two terms separately below:

- *Bound on $\mathtt{T}_A$.* Using the definition of $\mathtt{X}_\varepsilon$ from (69), we note that

$$
\begin{aligned}
\mathtt{T}_A &= \sum_{h=1}^{H} Z_{t,h} \mathbf{1}\{\pi_t(x_{t,h}) \neq \pi^\star(x_{t,h}), x_{t,h} \notin \mathtt{X}_\varepsilon\} \\
&= \sum_{h=1}^{H} Z_{t,h} \mathbf{1}\{\pi_t(x_{t,h}) \neq \pi^\star(x_{t,h}), \texttt{Query}(F_h^\star(x_{t,h}), \varepsilon \vec{\mathbb{1}}) = 0\} \\
&= \sum_{h=1}^{H} Z_{t,h} \mathbf{1}\{\exists m \in [m] \,:\, \|f_{t,h}^m(x_{t,h}) - f_h^{\star,m}(x_{t,h})\| > \varepsilon\},
\end{aligned}
$$

where the last line follows from the fact that the definition of $\texttt{Query}$ and the fact that $\pi_t(x_{t,h}) \neq \pi^\star(x_{t,h})$ implies that there exists some $m \in [M]$ for which $\|f_{t,h}^m(x_{t,h}) - f_h^{\star,m}(x_{t,h})\| > \varepsilon$. The above implies that

$$\mathtt{T}_A \leq \sum_{m=1}^{M} \sum_{h=1}^{H} Z_{t,h} \mathbf{1}\{\|f_{t,h}^m(x_{t,h}) - f_h^{\star,m}(x_{t,h})\| > \varepsilon\}.$$

---

[10] We use random seed $\iota_t$ to capture all the stochasticity in the choice of $\{\mathbb{T}_{t,h}\}_{h \leq H, t \leq T}$. However, all our proofs extend to IL learning with an arbitrary, and possibly adversarial, choice of $\{\mathbb{T}_{t,h}\}_{h \leq H, t \leq T}$

- *Bound on* $\mathtt{T}_B$. First note that Lemma 21 implies that with probability at least $1 - \delta$, for all $m \leq M$ and $h \leq H$,

$$\|f_h^{\star,m}(x_{t,h}) - f_{t,h}^m(x_{t,h})\| \leq \Delta_{t,h}^m(x_{t,h}). \tag{71}$$

Next, note that

$$\mathtt{T}_B \leq \sum_{h=1}^{H} \bar{Z}_{t,h} \mathbf{1}\{\pi_t(x_{t,h}) \neq \pi^\star(x_{t,h})\} \tag{72}$$

$$= \sum_{h=1}^{H} \mathbf{1}\{\mathtt{Query}(F_{t,h}(x_{t,h}), \vec{\Delta}_{t,h}(x_{t,h})) = 0, \pi_t(x_{t,h}) \neq \pi^\star(x_{t,h})\},$$

where in the last line follows from plugging in the query condition under which $Z_{t,h} = 0$. However note that for any $h \leq H$ for which $\mathtt{Query}(F_{t,h}(x_{t,h}), \vec{\Delta}_{t,h}(x_{t,h})) = 0$, by the definition of $\mathtt{Query}$ and the fact that $\pi_t(x_{t,h}) \neq \pi^\star(x_{t,h})$, there must exist some $m \in [M]$ such that

$$\|f_h^{\star,m}(x_{t,h}) - f_{t,h}^m(x_{t,h})\| > \Delta_{t,h}^m(x_{t,h}).$$

However, the above contradicts (71), and thus with probability at least $1 - \delta$,

$$\mathtt{T}_B = 0. \tag{73}$$

Plugging the above bounds on $\mathtt{T}_A$ and $\mathtt{T}_B$ in (70), we get that

$$R(\tau_t^{\pi^\star}) - R(\tau_t^{\pi_t}) \leq 2H \sum_{h=1}^{H} \mathbf{1}\{x_{t,h}^{\pi^\star} \in \mathtt{X}_\varepsilon\} + 2H \sum_{m=1}^{M} \sum_{h=1}^{H} Z_{t,h} \mathbf{1}\{\|f_{t,h}^m(x_{t,h}) - f_h^{\star,m}(x_{t,h})\| > \varepsilon\}. \tag{74}$$

**Step 2: Aggregating over all time steps.** Using the bound in (74) for each round $t$, we get that

$$\mathrm{Reg}_T = \sum_{t=1}^{T} \left( R(\tau_t^{\pi^\star}) - R(\tau_t^{\pi_t}) \right)$$

$$\leq 2H \sum_{h=1}^{H} \sum_{t=1}^{T} \mathbf{1}\{x_{t,h}^{\pi^\star} \in \mathtt{X}_\varepsilon\} + 2H \sum_{t=1}^{T} \sum_{m=1}^{M} \sum_{h=1}^{H} Z_{t,h} \mathbf{1}\{\|f_{t,h}^m(x_{t,h}) - f_h^{\star,m}(x_{t,h})\| > \varepsilon\}.$$

Using the fact that $\mathbf{1}\{a \geq b\} \leq a^2/b^2$ for any $a, b \geq 0$, and the definition of $T_{\varepsilon,h}$ in the above, we get that

$$\mathrm{Reg}_T \leq 2H \sum_{h=1}^{H} T_{\varepsilon,h} + 2H \sum_{t=1}^{T} \sum_{m=1}^{M} \sum_{h=1}^{H} Z_{t,h} \frac{\|f_{t,h}^m(x_{t,h}) - f_h^{\star,m}(x_{t,h})\|^2}{\varepsilon^2}$$

$$\leq 2H \sum_{h=1}^{H} T_{\varepsilon,h} + \frac{2H}{\varepsilon^2} \sum_{m=1}^{M} \sum_{h=1}^{H} \Psi_\delta^{\ell_\phi}(\mathcal{F}_h^m, T). \tag{75}$$

where the last line follows from using the bound in Lemma 21.

Plugging in the form of $\Psi_\delta^{\ell_\phi}(\mathcal{F}_h^m, T)$ and ignoring $\log$ factors and constants, we get that

$$\mathrm{Reg}_T \lesssim H \sum_{h=1}^{H} T_{\varepsilon,h} + \frac{H}{\lambda \varepsilon^2} \sum_{m=1}^{M} \sum_{h=1}^{H} \mathrm{Reg}^{\ell_\phi}(\mathcal{F}_h^m; T) + \frac{MH^2}{\lambda^2 \varepsilon^2} \log(1/\delta).$$

Notice that $\varepsilon$ is a free parameter above so the final bound follows by taking $\inf$ over all feasible $\varepsilon$.

### F.5.3 Total Number of Queries

Fix any $t \leq T$, and let $h_t$ denote the first time step at round $t$ for which $Z_{t,h_t} = 1$, if such a time-step exists (and is set to be $H + 1$ otherwise). We first observe that for all $h \leq h_t$, we have $\pi^\star(x_{t,h}) = \pi_t(x_{t,h})$. To see this, note that

$$Z_{t,h_t} \mathbf{1}\{\exists h < h_t : \pi^\star(x_{t,h}) \neq \pi_t(x_{t,h})\} \leq \sum_{h=1}^{h_t-1} Z_{t,h_t} \mathbf{1}\{\pi^\star(x_{t,h}) \neq \pi_t(x_{t,h})\}$$

$$\leq \sum_{h=1}^{h_t-1} Z_{t,h_t} \bar{Z}_{t,h} \mathbf{1}\{\pi^\star(x_{t,h}) \neq \pi_t(x_{t,h})\}$$

$$\leq \sum_{h=1}^{h_t-1} \bar{Z}_{t,h} \mathbf{1}\{\pi^\star(x_{t,h}) \neq \pi_t(x_{t,h})\}$$

where in second inequality above, we used the fact that $Z_{t,h} = 0$ (and thus $\bar{Z}_{t,h} = 1$) for all $h < h_t$, by the definition of $h_t$. Observe that the right hand side in the last inequality above is equivalent to the term (72) in the bound on $\mathtt{T}_B$ above (where sum is now till $h_t$ instead of $H$). Thus, using the bound in (73), we get that

$$Z_{t,h_t} \mathbf{1}\{\exists h < h_t : \pi^\star(x_{t,h}) \neq \pi_t(x_{t,h})\} = 0,$$

and thus

$$\pi^\star(x_{t,h}) = \pi_t(x_{t,h}) \qquad \text{for all } h \leq h_t. \tag{76}$$

Next, let $\varepsilon > 0$ be a free parameter, and note that plugging in the definition of $h_t$, we get that the total number of samples is bounded as:

$$N_T = \sum_{t=1}^{T} \sum_{h=1}^{H} Z_{t,h}$$

$$\leq H \sum_{t=1}^{T} Z_{t,h_t}$$

$$= H \sum_{t=1}^{T} Z_{t,h_t} \mathbf{1}\{x_{t,h_t} \in \mathtt{X}_\varepsilon\} + H \sum_{t=1}^{T} Z_{t,h_t} \mathbf{1}\{x_{t,h_t} \notin \mathtt{X}_\varepsilon\}$$

$$= H \sum_{t=1}^{T} Z_{t,h_t} \mathbf{1}\{x_{t,h_t} \in \mathtt{X}_\varepsilon\} + H \sum_{t=1}^{T} Z_{t,h_t} \mathbf{1}\{x_{t,h_t} \notin \mathtt{X}_\varepsilon, \|\Delta_{t,h_t}^m(x_{t,h_t})\|_\infty \leq \frac{\varepsilon}{4}\}$$

$$+ H \sum_{t=1}^{T} Z_{t,h_t} \mathbf{1}\{x_{t,h_t} \notin \mathtt{X}_\varepsilon, \|\Delta_{t,h_t}^m(x_{t,h_t})\|_\infty > \frac{\varepsilon}{4}\}$$

$$= \mathtt{T}_C + \mathtt{T}_D + \mathtt{T}_E,$$

where $\mathtt{T}_C$, $\mathtt{T}_D$ and $\mathtt{T}_E$ are the first, second and the third term respectively in the previous line. We bound them separately below.

- *Bound on* $\mathtt{T}_C$. Fix any $t \leq T$. Using the relation in (76), note that $\pi^\star(x_{t,h}) = \pi_t(x_{t,h})$ for all $h < h_t$. Thus, the corresponding trajectories would be identical till time step $h_t$, which implies that $x_{t,h_t} = x_{t,h_t}^{\pi^\star}$. Using this property in the $\mathtt{T}_C$, we get that

$$(\mathtt{T}_C)_t = H \sum_{t=1}^{T} Z_{t,h_t} \mathbf{1}\{x_{t,h_t} \in \mathtt{X}_\varepsilon\}$$

$$= H \sum_{t=1}^{T} Z_{t,h_t} \mathbf{1}\{x_{t,h_t}^{\pi^\star} \in \mathtt{X}_\varepsilon\}$$

$$\leq H \sum_{t=1}^{T} \sum_{h=1}^{H} Z_{t,h} \mathbf{1}\{x_{t,h}^{\pi^\star} \in \mathtt{X}_\varepsilon\}$$

$$= H \sum_{h=1}^{H} T_{\varepsilon,h},$$

where the last line plugs in the definition of $T_{\varepsilon,h}$.

- *Bound on* $\mathtt{T}_D$. First note that
$$(\mathtt{T}_D)_t = H\mathbf{1}\{\mathtt{Query}\big(F_{t,h_t}(x_{t,h_t}), \vec{\Delta}_{t,h_t}(x_{t,h_t})\big) = 1, \mathtt{Query}\big(F^\star(x_{t,h_t}), \varepsilon\vec{\mathbb{1}}\big) = 0, \sup_{m \in [M]} \Delta_t^m(x_{t,h_t}) \leq \varepsilon/4\}.$$

In the following, we will show that all the conditions in the above indicator can not hold simultaneously. First note that since $\mathtt{Query}\big(F_{t,h_t}(x_{t,h_t}), \vec{\Delta}_{t,h_t}(x_{t,h_t})\big) = 1$, there exists an $\widetilde{F}$ such that

$$\mathtt{SelectAction}(\widetilde{F}(x_{t,h_t}))) \neq \mathtt{SelectAction}(F_{t,h_t}(x_{t,h_t})) \tag{77}$$

and

$$\forall m \in [M]: \qquad \|\widetilde{F}(x_{t,h_t})[:,m] - F_{t,h_t}(x_{t,h_t})[:,m]\| \leq \Delta^m_{t,h_t}(x_{t,h_t}). \qquad (78)$$

On the other hand, recall that Lemma 21 implies that

$$\forall m \in [M]: \qquad \|F^\star(x_{t,h_t})[:,m] - F_{t,h_t}(x_{t,h_t})[:,m]\| \leq \Delta^m_{t,h_t}(x_{t,h_t}). \qquad (79)$$

Since, $\sup_m \Delta^m_{t,h_t}(x_{t,h_t}) \leq \varepsilon/4$, an application of Triangle inequality along with the bounds (78) and (79) imply that

$$\forall m \in [M]: \qquad \|F^\star(x_{t,h_t})[:,m] - \widetilde{F}(x_{t,h_t})[:,m]\| \leq 2\Delta^m_{t,h_t}(x_{t,h_t}) < \varepsilon. \qquad (80)$$

But the above contradicts the fact that $\mathtt{Query}(F^\star(x_{t,h_t}), \varepsilon\vec{\mathbb{1}}) = 0$ since both $\widetilde{F}$ and $F_t$ satisfy the norm constraints in the definition of $\mathtt{Query}$, but we can not simultaneously have that

$$\mathtt{SelectAction}(F^\star(x_{t,h_t}))) = \mathtt{SelectAction}(F_{t,h_t}(x_{t,h_t})) = \mathtt{SelectAction}(\widetilde{F}(x_{t,h_t})),$$

due to (77). Thus, we must have that

$$(\mathtt{T}_D)_t = 0.$$

- *Bound on* $\mathtt{T}_E$. We note that

$$\mathtt{T}_E \leq H \sum_{t=1}^{T} Z_{t,h_t} \mathbf{1}\{\|\vec{\Delta}_{t,h_t}(x_{t,h_t})\|_\infty > \varepsilon/4\}$$

$$= H \sum_{t=1}^{T} Z_{t,h_t} \mathbf{1}\{\exists m \in [M]: \Delta^m_{t,h_t}(x_{t,h_t}) > \varepsilon/4\}$$

$$\leq H \sum_{m=1}^{M} \sum_{t=1}^{T} Z_{t,h_t} \mathbf{1}\{\Delta^m_{t,h_t}(x_{t,h_t}) > \varepsilon/4\}$$

$$\leq H \sum_{h=1}^{H} \sum_{m=1}^{M} \sum_{t=1}^{T} Z_{t,h} \mathbf{1}\{\Delta^m_{t,h}(x_{t,h}) > \varepsilon/4\},$$

where the last line simply upper bound the term for $h_t$ by the corresponding terms for all $h \leq H$.

Using Lemma 22 to bound the term in the right hand side for each $m \in [M]$ and $h \leq H$, we get that

$$\mathtt{T}_E \leq \sum_{h=1}^{H} \sum_{m=1}^{M} \frac{320 H \Psi^{\ell_\phi}_\delta(\mathcal{F}^m_h, T)}{\varepsilon^2} \cdot \mathfrak{E}(\mathcal{F}^m_h, \frac{\varepsilon}{8}; f^{\star,m}_h).$$

Gathering the bound above, we get that

$$N_T \leq H \sum_{h=1}^{H} T_{\varepsilon,h} + \frac{320 H}{\varepsilon^2} \sum_{h=1}^{H} \sum_{m=1}^{M} \Psi^{\ell_\phi}_\delta(\mathcal{F}^m_h, T) \cdot \mathfrak{E}(\mathcal{F}^m_h, \frac{\varepsilon}{8}; f^{\star,m}_h).$$

Plugging in the form of $\Psi^{\ell_\phi}_\delta(\mathcal{F}^m_h, T)$ and ignoring log factors and constants, we get that

$$N_T \lesssim H \sum_{h=1}^{H} T_{\varepsilon,h} + \frac{H}{\lambda\varepsilon^2} \sum_{h=1}^{H} \sum_{m=1}^{M} \mathrm{Reg}^{\ell_\phi}(\mathcal{F}^m_h; T) \cdot \mathfrak{E}(\mathcal{F}^m_h, \varepsilon/8; f^{\star,m}_h) + \frac{MH^2}{\lambda^2\varepsilon^2} \log(1/\delta).$$

Notice that $\varepsilon$ is a free parameter above so the final bound follows by taking $\inf$ over all feasible $\varepsilon$.

