# OpenReview forum: "Selective Sampling and Imitation Learning via Online Regression"
_NeurIPS.cc/2023/Conference — NeurIPS 2023 poster_

### Official Review · Reviewer_mVoN · 2023-07-04

**Soundness:** 2 fair
**Presentation:** 2 fair
**Contribution:** 3 good
**Rating:** 6
**Confidence:** 3

**Summary:**

In this work, the authors make several contributions. Firstly, they propose a novel selective sampling algorithm (Algorithm 1) that utilizes a regression oracle and provide regret bounds and query complexity analysis. Secondly, they present a lower bound that demonstrates the unavoidable dependency of query complexity on the eluder dimension without additional assumptions. Thirdly, they introduce Algorithm 2, which is designed for to iid sample distributions and employs the disagreement coefficient as a substitute for the eluder dimension. For imitation learning, the authors prove an exponential gap in sample complexity between stochastic and deterministic expert demonstrations in the context of offline imitation learning. Lastly, they propose imitation learning algorithms that utilize regression oracles with selective sampling for both single and multiple stochastic expert settings.

**Strengths:**

1.This work provides valuable insights into the field of neural network-based imitation learning with selective sampling, laying the foundations for future advancements in practical and efficient algorithms.

2.The authors' contributions encompass both selective sampling and imitation learning, showcasing the potential for designing imitation learning algorithms that offer provable efficiency. Their research opens up exciting possibilities for further advancements in the field.

**Weaknesses:**

1.While this work presents self-contained proofs, it lacks a direct comparison of the imitation learning regret and query complexity of the proposed algorithm with Dagger and other existing algorithms. This omission makes it challenging to assess the efficiency and performance of the proposed approach accurately.

2.The experiment conducted on Cartpole may not provide sufficient insight and understanding of the algorithm's performance. Furthermore, the use of sophisticated selective sampling techniques may introduce significant computational overhead when applied to more complex experimental environments such as hopper and walker. This computational burden hinders the evaluation of the algorithm and makes it difficult to further explore its effectiveness in practical scenarios.

3.The usage of a deterministic Markov Decision Process (MDP) with a fixed starting state, which may not align with standard imitation learning problems. This restriction limits the applicability of the guarantees provided in this work to more general and diverse scenarios.

**Questions:**

Questions:
1.Lemma 18 does not appear to me a variant of the performance difference lemma, where in the second inequality of the equation after line 1075 the value on the right is already $2H$, the upper bound of the performance difference. Please explain if this implies the other results that applies lemma 18 are already greater than $2H$.

2.In line 281, for AggreVate, I believe it takes $O(\log(H))$ rounds for the probability of executing expert policy to be lower than $1/H^2$, where by the union bound we have the probability of executing the learner's policy for whole $H$ steps greater than $(H-1)/H$. If mixing with the noisy expert is the only issue, I don't agree that AggreVaTe style algorithm should sample $\Omega(2^H)$ samples if I am allowed to mix the proposed algorithm 2 with noisy expert policy as AggreVaTe-style in the roll out.

3.I am confused why in line 266 we need a large margin over state space for Dagger instead of the recoverability assumption. By the performance difference lemma, we only need the expert's advantage function to be bounded by some value smaller than $H$ for Dagger to achieve better regret guarantee than behavior cloning. Please feel free to let me know if you need any reference.

4.I am curious why the number of queries for Dagger is not a straight line, which may due to sampling trajectories instead. Please provide more experimental details including model and training details for more transparent presentation.

5.Does the analysis in this work generalizable to non-deterministic learner policies?

6.Is proposition 1 extendable to stochastic transition dynamic setting? I am afraid the stochastic transition dynamics will make any roll out policy hard to reach the designed leaf node.

7.Is line 5 in algorithm 1 easy to compute when the policy class is exponentially large?

8.Querying all $M$ experts and assigning $H \times M$ learner policies is quite computationally demanding, any method to reduce to learning only one policy under some assumption of realizability?


Minor suggestions:

1.In line 1006 it would be better to define $r(x,a) = 0$ for none leaf nodes.

2.In line 1022 use $ \forall \pi \in \Pi$ to avoid confusion.

3.In line 257, could you please clarify the definition of your regret, which is missing the expectation over transition dynamics. I assume you are in the deterministic transition setting. The state that would have been generated is a bit vague, please point out the deterministic MDP setting with unique starting state directly.

4.In line 256 'the learner proposes an action' is a bit vague, I suppose you are suggesting a deterministic learner's policy in stead of a stochastic random policy like a noisy expert. Could you please make it more clear by mentioning 'deterministic'?

5.In line 491, though using the last round policy as warm start may result in similar performance as training from random initialization. It would be beneficial to conduct one extra experiment comparing the result of using warm start and naively start from random initialization for each round, which would empirically justify this speedup.

6.In the Figure 2, Dagger takes around 13 iterations to converge, where it would be helpful to mark the exact number of annotations for Dagger to achieve near expert performance as a dotted line on the right of figure 2.

**Limitations:**

1.The lack of a direct theoretical comparison between the proposed algorithm's imitation learning regret and query complexity with Dagger and other preexisting algorithms hinders the interpretation of its efficiency. Comparative evaluations against existing algorithms would provide a clearer understanding of the algorithm's performance and effectiveness.

2.The experiment conducted solely on the Cartpole environment may not offer sufficient insights into the algorithm's behavior and capabilities. Additionally, the computational overhead associated with the sophisticated selective sampling technique might pose challenges when applying the algorithm to more complex environments like hopper and walker. This limitation makes it difficult to thoroughly evaluate the algorithm's performance in a broader range of scenarios.

3.The use of a deterministic Markov Decision Process (MDP) with a fixed starting state deviates from the standard settings in imitation learning problems. This restriction limits the guarantee and applicability of the proposed algorithm, as it may not generalize well to more diverse and realistic scenarios encountered in practice.

4.While Proposition 1 is supported by theoretical proof, it would be beneficial to include empirical observations to demonstrate its importance and practical relevance. Additionally, conducting experiments that involve behavior cloning would provide further insights and comparisons, enhancing the experimental evaluation and supporting the theoretical claims.

---

> ### Author Rebuttal · Authors · 2023-08-10
>
> Thank you for the detailed review and for your questions. Before providing the answers to your questions, we first recap our notation for the imitation learning setting which we will repeatedly rely upon in the rebuttal.
>
> - **Noisy Expert**: We assume that the learner interacts with an expert who gives noisy feedback. In particular, the expert has a score function $f^\star \in \mathcal{F}$, where $f^\star: \mathcal{X} \mapsto \Delta(K)$ and $K$ denotes the number of actions, using which, for any context $x$ the expert produces its label as
> $y \sim f^\star(x)$.
>
>
>      Importantly, we do not assume that $f^\star$ denotes the exact value function of the expert policy on the underlying MDP. However, there is an implicit assumption that comparing to the most likely preference of the expert (i.e. the deterministic policy $\pi^\star$) is reasonable, i.e. on a large part of the relevant state space, we have  $\arg\max\_{k \in K} f^\star(x)[k] = \arg\max\_{k \in K} Q^\star(x)[K]$. Such a relation can be obtained for example, when $f^\star$ is proportional to $Q^\star$, as is the case for our lower bound construction in proposition 1.
>
> - **Comparator Policy:**  We wish to compete with the deterministic policy $\pi^\star$ given by $\pi^\star(x) = \arg \max_{k \in [K]} f^\star(x)[k]$. When $f^\star$ aligns with the optimal value function, the deterministic policy $\pi^\star$ denotes the optimal policy for the MDP.
>
> The above captures the scenario where the expert has some intuition of the optimal action to take at any given state but may not be confident (hence, the noise).
>
>
> ### Lemma 18's Relation to Performance Difference Lemma
>
> Lemma 18 is similar in spirit to the performance difference lemma. In particular when $X = \emptyset$, we get that
>
>
> $R \leq 2H \sum_{i=1}^H 1\\{\pi\_2(x\_h^{\pi\_2}) \neq \pi\_1(x\_h^{\pi\_2}))\\}$
>
> The above is essentially the one-step disagreement of two policies on the trajectories generated according to $\pi\_2$, which is similar to the performance difference lemma (and hence the name). The $2H$ factor that the reviewer is alluding to appears multiplicatively with the other indicator terms that get converted to margin terms and the oracle regret terms in our final regret and query complexity bounds.
>
> ### AggreVate-style algorithms and Proposition 1
>
> We apologize we do not fully understand your question. Please let us know if  the following response does not fully answer your question.
>
> In our regret bound for imitation learning, we compare against the deterministic policy $\pi^\star(x) = \arg\max_{a} f^\star(x)$  whereas the expert gives noisy feedback sampled as $y \sim f^\star(x)$. For the lower-bound construction in Proposition 1:
>
> - Since the noisy expert samples its actions using $f^\star$, one can check that the value obtained by the noisy expert is  $O(1/2 + 2^{-H})$.
> - On the other hand, the value of the policy $\pi^\star$ is $3/4$.
>
> Thus, any aggravate style algorithm that directly imitates the noisy expert will obtain value  $O(1/2 + 2^{-H})$ unless it collects $2^H$ samples to observe the optimal action sequence in the tree (in our construction, such probability is exponentially small for the noisy expert). On the other hand, our algorithm explicitly plays deterministic policies given by $\mathsf{SelectAction}(f\_t(x\_t))$ which eventually converges to $\pi^\star(x\_t) = \mathsf{SelectAction}(f^\star(x\_t))$.
>
> ### Comparison to DAgger/Behavior Cloning
>
> **In terms of Regret:** DAgger, as well as Behavior cloning, aims to output a policy whose performance is comparable to that of the noisy expert from which the labels are sampled. Thus, the output policy may not have small regret against the comparator  $\pi^\star$ (defined above). As we discussed above in our answers above, the gap between the two policies could be significant.
>
> **In terms of Queries:** DAgger/Behavior Cloning are passive algorithms that query on every round, and do not do any active queries.
>
> ### Straight line for DAgger and requested experiment details
>
> In our cartpole experiments, similar to the standard experimental setups, we terminate the episode when either the pole is out of balance
> or the cart deviates too far from the origin (see lines 461-467 for more details). Since episodes terminate early in the initial few rounds of DAgger we get a slight flattening of the straight line shown in Figure 2.
>
> As described in lines 478 onwards in Appendix A, our policies are parameterized using neural networks (with a single hidden layer, and 4 neurons in each hidden layer). The rest of the experiment details are also listed there.
>
> ### Concern about large margin over state space for Dagger
>
>
> In our regret bound, we compare against the deterministic policy $\pi^\star$ (defined above), whereas DAgger can only compare to the expected performance of the noisy expert (i.e. the policy that samples from $f^\star$). Furthermore, the DAgger analysis does not aim to minimize the query complexity. In fact, in order to have a small query complexity with the DAgger-style of analysis, we would require a large margin of the expert policy over the state space that would be explored by the learner (this is because of the way DAgger uses performance difference lemma). On the other hand, our analysis only requires a large margin for the expert policy over the state space that is explored by the expert (instead of the learner). The latter could be much smaller.
>
> Finally, note that having a large margin of the expert policy over the entire state space, while being too much to ask for, would imply that the noisy policy and the deterministic policy would have comparable performance, and thus the two algorithms should perform similarly.
>
> ### Extension to Stochastic MDPs
>
> All our results can be trivially extended to stochastic MDPs. We provide a formal proof in Appendix D.4 on Page 45.
>
> ### Computationally Efficient Implementations
>
> Please find a shared response to all the reviewers above!

---

> > ### Author Response · Authors · 2023-08-10
> > **Rebuttal Continued**
> >
> > ### All results trivially extend to Stochastic MDPs
> >
> > All our results can be trivially extended to stochastic MDPs. We provide more discussion and proof in Appendix D.4 on Page 45.  For ease of analysis and presentation, we consider a scenario where the learner interacts with deterministic dynamics $\\{T\_{t, h}\\}\_{h \leq H}$ at time $t$ that could be adversarially chosen (and is unknown to the learner). However, because the dynamics can change adversarially, our assumptions are strictly weaker than assuming stochastic dynamics. In particular, in order to simulate stochastic dynamics, one can simply consider that for every round $t \in [T]$ a random seed $z\_t$ is chosen from a fixed unknown distribution. Given the seed $z\_t$, the adversarially chosen dynamics can simply be set as $\\{T\_{t, h}\\}\_{h \leq H} = T(; z\_t)$ where $T$ is some stochastic dynamics.
> >
> > ### Is proposition 1 extendable to a stochastic transition dynamic setting?
> >
> > The main contribution of the lower bound was to provide a lower bound for prior works on non-interactive imitation learning *when the expert provides noisy feedback*. On the other hand, our algorithm can solve the problem instance in the lower bound after collecting polynomially many samples.
> >
> > Since it is a lower bound, we only provided one instance which was a determinstic MDP. However, we believe that a similar lower bound could also be constructed for stochastic MDPs by changing the reward function on the leaf nodes. Note that a simpler stochastic MDP construction may be obtained by directly using a combination lock with rich observations as the underlying MDP.
> >
> > ### Does the analysis in this work generalizable to non-deterministic learner policies?
> >
> > Note that the policy $\pi^\star$ that we wish to compare to in our regret bounds is deterministic. Thus, we do not require stochasticity in the learner policies in order to obtain the optimal regret/query complexity bounds. However, our algorithm can be extended to work with stochastic policies parameterizations (e.g. neural networks with softmax) when they are sufficiently capable of representing close-to-deterministic policies (e.g. by tuning the scale parameter in softmax).

---

> > > ### Comment · Reviewer_mVoN · 2023-08-19
> > > **Thanks a lot for your explanation.**
> > >
> > > Dear Author,
> > >
> > > Thanks for addressing my questions and I am happy to increase my score to 6.
> > >
> > > Best regards,
> > > mVoN

---

### Official Review · Reviewer_hUJx · 2023-07-05

**Soundness:** 3 good
**Presentation:** 3 good
**Contribution:** 3 good
**Rating:** 7
**Confidence:** 2

**Summary:**

The paper proposed a selective sampling scheme on an online regression oracle over a function class $\mathcal{F}$ and time horizon $T$ for multiclass setting, which has the regret guarantee $Reg^{\ell_{\phi} } (\mathcal{F}, T)$, and we assume the label $y_t$ is sampled through a link function $\phi$. The algorithm actively acquire a label when the margin (the difference among top two action scores) are small, which resembles sampling by disagreement in active learning research.

Assuming the problem has a margin greater than some $\epsilon$ at all round and adversarially chosen context $x_t$, the sampling scheme achieve regret $\tilde{O}( Reg^{\ell_{\phi} } (\mathcal{F}, T) / \epsilon) $, and the labelling complexity as $\tilde{O} ( Reg^{\ell_{\phi} } (\mathcal{F}, T) / \epsilon^2 )$ ( also have dependence on eluder dimension of $\mathcal{F}$). In the case that the context $x_t$ is stochastically chosen, the algorithm achieves the same regret but fewer labels. The result can also be extended to imitation learning, where the regret bound and query number scales with episode number $H$ and teacher number $M$.


**Strengths:**

The paper is well-written and proposed a simple sampling rule based on the margin on the fly. The appendix was well managed and easy to follow.

The regret bound captures the difficulty of the problem and the regret of the online regression oracle, which is very general. The paper also shows lower bound on the tightness of the label complexity.

The proposed sampling rule was derived for both adversarial and stochastically chosen context, showing the later is improved in terms of labelling complexity. The sampling rule, and analysis can also be extended to imitation learning. Experimental results shows the trade off between convergence speed and labelling complexity.

**Weaknesses:**

Some possible extensions:

1. Since the experiment shows $\alpha$ in equation (7) was experimentally determined due to the complexity of finding the exact Lagrangian multiplier, whether theoretical bound can include the scenario: if $\Delta_t(x_t)$ can not be solved exactly. Instead we can only use an approximation denote as $\tilde{\Delta}_t(x) $ such that $| \tilde{\Delta}_t(x) - \Delta(x_t) | \le \delta $ for some threshold $\delta$. How does this error $\delta$ affects regret and sampling number.

2. For imitation learning multiple teacher, it looks like the regret and query number for Multiple teacher $M > 1$ (theorem 5) has a worse guarantee in terms of $M$ comparing to single teacher in theorem 4, due to $\sum_{m=1}^{M} $. This is counter intuitive in the sense that more resources was used. Whether the algorithm can be extended so we have regret and query number with $\min_{m \in [M] }$ instead of $\sum_{m=1}^{M}$. An example to illustrate current concern: in the single teacher setting, we use ${Oracle}_1$ has regret $R_1$. For the multiple teacher setting, we use ${Oracle}_1, {Oracle}_2, \cdots, {Oracle}_M$, which all have regret $R_1$, then multiple teacher setting is picking up $M R_1$ and single teacher is only $R_1$. Although they are using the same quality of teacher oracles.

**Questions:**

see above



**Limitations:**

The paper might have been limited by the submission page limit, which ends without an explicit discussion session. The paper already summarized its finding and significances. Currently limitation / improvement is addressed in the weakness session, a broader and domain specific limitation, future direction should be included if more space is permitted.

---

> ### Author Rebuttal · Authors · 2023-08-10
>
> Thank you for your valuable feedback. We will be sure to incorporate your suggestions in the final version of the paper. Please find our response to your concerns below.
>
>
> ### How does $\delta$ error propagate?
>
> Our algorithm can be easily modified if $\Delta\_t(x\_t)$ could only be solved approximately, up to an error tolerance $\delta$ (we assume that the learner knows $\delta$). In this case, we simply need to modify the query condition to be:
> $Z\_t = 1\\{\mathrm{Margin}(f\_t(x\_t)) \leq 2 \gamma (\Delta\_t(x\_t) + \delta)\\}$. Incorporating this change, the regret bound will remain unchanged. On the other hand, the query complexity bound will only change by a constant multiplicative factor in terms of $\epsilon$ for any $\epsilon > \delta$. This is because any $\epsilon > \delta$ can be eaten in the term $\epsilon/4 \gamma$ in the split in line 802 in the query complexity proof. Ignoring constant factors and for $\gamma = \lambda = 1$, the final bounds will be:
> - $\mathrm{Regret}:~~~~ \widetilde{O}(\inf\_{\epsilon > \delta} (T\_\epsilon + \frac{\mathrm{Reg}^{\mathrm{sq}}(\mathcal{F}, T)}{\epsilon}))$
> - $\mathrm{Query Complexity}: ~~~~ \widetilde{O}(\inf\_{\epsilon > \delta} (\epsilon T\_\epsilon + \frac{\mathrm{Reg}^{\mathrm{sq}}(\mathcal{F}, T) \mathfrak{E}(\mathcal{F}; \epsilon, f^\star)}{\epsilon^2}))$
>
> ### Dependence on $\sum\_{m=1}^M$ in the bounds for multiple experts
>
> The dependence on $M$ (or $\sum\_{m=1}^M$) in the regret/query complexity reflects the complexity of the aggregation function that is used to define the comparator policy by combining the advice of the given experts. In the case when the aggregator function always picks the output of a fixed expert (i.e. ignores everyone else throughout the interaction), our analysis can be easily extended to not have a sum corresponding to all the experts. On the other hand, for more complex aggregation functions, e.g. majority opinion, etc., we believe that some dependence on all the experts would be necessary as we are competing against a more complex comparator policy (that depends on all of the $M$ experts).
>
> > Concluding Discussions
>
> We will definitely include more discussion and a conclusion in the final version of the paper.

---

> > ### Comment · Reviewer_hUJx · 2023-08-13
> >
> > Dear Authors,
> >
> > Thank you for your several detailed examples on solving the label query parameter $\Delta_t(x_t)$, for elaborating on how $\delta$ error propagates. Thank you for explaining the dependence on $M$ for multiple expert setting.
> >
> > On top of the interesting theoretical bounds being presented given access to an oracle, my concerns has been addressed, hence I increased the score from 6 to 7.

---

### Official Review · Reviewer_A81c · 2023-07-06

**Soundness:** 3 good
**Presentation:** 2 fair
**Contribution:** 3 good
**Rating:** 5
**Confidence:** 3

**Summary:**

This paper studies selective sampling and imitation learning. In this setting a series of states arrives online, and a learner has the ability to choose to query expert labels/policy in order to correctly label the states/perform as well as an expert policy. The goal is to minimize the regret (i.e. the drop in performance relative to what the expert would achieve on the particular sequence) while also making as few queries to the expert labels/policy.

The authors design algorithms under the additional assumption that there is black-box access to an online regression oracle for the class of functions $\mathcal{F}$ under consideration. The algorithms achieve regret bounds that depend linearly on the regret achieved by the online regression oracle, plus the number of times that the expert-provided label function has a small-margin in the online sequence. These regret bounds are essentially best-possible considering the performance of the regression oracle. For the number of expert queries, the bounds for the algorithms grow proportional the product of the regret bounds of the oracle and the eluder dimension. The authors further show that the dependence on Eluder dimension is necessary.


**Strengths:**

The paper provides a clear general method to transform an online-regression oracle for a given function class $\mathcal{F}$ into an algorithm for active learning using the class $\mathcal{F}$. This method essentially maintains the regret of the original regression oracle (which has access to all the labels), while making as few oracle queries to the expert labels/policy as possible (without making further assumptions).

**Weaknesses:**

1. Both algorithms in the paper rely on solving an optimization problem (Equation (5) for Algorithm 1 and Equation (6) for Algorithm 2). It is not immediately clear how to use the online regression oracle to solve these problems. If they cannot be solved via the oracle, it seems likely that they are generally intractable for many function classes $\mathcal{F}$ of interest. This seems like an important issue for both algorithms, and significantly limits the utility of the method if an additional oracle is required to solve the problems in (5) and (6).

2. A minor point: the notation on line 9 of Algorithm 1 was initially confusing to me. Presumably the regression oracle is called with all previously queried labels, but the notation actually used was never introduced and so it is unclear what was actually meant.

**Questions:**

1. I don't immediately see why the optimization problem in Algorithm 1 Equation (5) is tractable, even given an online regression oracle for the function class $\mathcal{F}$. Am I missing something or is another assumption needed here (e.g. that we have a second oracle to solve the optimization problem in (5))? Alleviating my concerns about this issue would cause me to increase my score.

2. Can you clarify exactly which sequence is sent to the oracle in line 9 of algorithm 1?

**Limitations:**

Yes.

---

> ### Author Rebuttal · Authors · 2023-08-10
>
> Thank you for your valuable feedback. We will be sure to incorporate your suggestions in the final version of the paper. Please find our response to your concerns below.
>
>
>
> ### Solving optimization problem in equation (5) for Algorithm 1 and equation (6) for Algorithm 2
>
> Please find our shared response to all the reviewers above! We would love to discuss further with you if you have any further concerns.
>
>
> ### Sequence sent to the oracle
>
> We assume access to an online regression oracle which at the time $t$, has already seen the history $\{(x\_1, y\_1), \dots, (x\_{t-1}, y\_{t-1})\}$ and is required to predict a function $f_t \in \mathcal{F}$. After predicting $f\_t$, the oracle gets the datapoint $\{(x\_t, y\_t)\}$. Formally, the interaction is as follows:
>
> For $ t = 1, \dots, T:$
>
> 1. Predict $f\_t \in \mathcal{F}$.
> 2. Get sample $(x\_t, y\_t)$

---

> > ### Comment · Reviewer_A81c · 2023-08-16
> >
> > Thank you for your explanation, particularly of how to efficiently implement the optimization in equation (5) given access to a regression oracle. This would definitely be good to include in future versions of the paper. I will increase my score to 5.

---

> > > ### Author Response · Authors · 2023-08-16
> > > **Will do!**
> > >
> > > Thank you for your positive response to our rebuttal! We will definitely include a section on computationally efficient implementations in the revised version of our paper.

---

### Official Review · Reviewer_hxAC · 2023-07-26

**Soundness:** 3 good
**Presentation:** 4 excellent
**Contribution:** 3 good
**Rating:** 6
**Confidence:** 2

**Summary:**

The paper studies algorithms for selective sampling with applications to imitation learning.
In the former setting the authors propose an algorithm build around an online regression oracle and derive bounds on the overall regret and number of queries.
These are provided for both a general (potentially adversarial) setting as well as a potentially more favourable stochastic setting. A key component is the dependence on the eluder dimension of the regressor family and the authors show that a dependency on this is unavoidable by also providing matching lower bounds.

In the imitation learning setting the notion of an expert confidence *margin* is introduced. The authors argue that in an offline setting requirements for expert trajectories can grow exponentially w.r.t. to the length of the considered time horizon and consequently an online learning setup is required to achieve reasonable regret. Based on the algorithm from the selective sampling setting a matching formulation for imitation learning with corresponding regret derivations is presented. Finally, the notion of expert uncertainty encoded in margin scores allows for an extension to a multi-expert setting.

**Strengths:**

The paper is well structured and clearly elucidates the short-comings of prior work it addresses. The extensions proposed to overcome them are clearly motivated and the proposed algorithms appear sound.

**Weaknesses:**

Despite being motivated by making the algorithms more fit for practical applications, there is a rather limited experimental evaluation of the proposed algorithms. It would be interesting to compare their empirical performance to existing baselines such as DAgger.

**Questions:**

-

**Limitations:**

-

---

> ### Author Rebuttal · Authors · 2023-08-10
>
> Thank you for your valuable feedback. We will be sure to incorporate your suggestions in the final version of the paper. Please find our response to your concerns below.
>
> > Limited Experimental Evaluation
>
> Our motivation in this work is to theoretically understand regret and query complexity bounds for selective sampling (and imitation learning) with general function classes. Towards that end, our primary focus in the paper was to develop statistically efficient and general-purpose algorithms that build on online regression oracles w.r.t.~the underlying classes of interest. However, in order to show that our algorithm can be made practical we demonstrate experiments on simple toy problems, like cartpole (see Appendix A for more details). Making our algorithm practically applicable for large-scale and more complex imitation learning problems is a fascinating research direction, and is rich enough to form a paper of its own.
>
> **Regarding comparison to DAgger**: In our experiments in Appendix A, we provide a comparison to DAgger (called as the Passive algorithm in the figures and plots) and show that our algorithms can achieve comparable performance but with much fewer queries (See Figure 2 for comparison w.r.t. The query complexity and regret).

---

### Author Rebuttal · Authors · 2023-08-10

## Solving optimization problem in equation (5) for Algorithm 1 and equation (6) for Algorithm 2 computationally efficiently

Our work is theoretical in nature, and our focus in the paper is thus to understand the statistically efficient (query complexity), and to develop algorithms, for selective sampling and imitation learning with general model classes $\mathcal{F}$, given access to an online regression oracle w.r.t.~$\mathcal{F}$. However, in many cases, our algorithm (or its slight modification) can also be implemented efficiently. We describe some scenarios below:

- **1D-Linear models**: When $f: \mathcal{X} \mapsto \mathbb{R}$ is linear, the optimization objective in eqn (5) in Algorithm 1 (or eqn (6) in Algorithm 2) can be efficiently by instead solving the objectives $\Delta\_t^{(1)}(x\_t) = \max\_{f \in \mathcal{F}} f(x\_t) - f\_t(x\_t) \text{ s.t. } \sum\_{s=1}^{t-1} Z\_s |f(x\_s) - f\_s(x\_s)|^2 \leq \Psi$, and $\Delta^{(2)}\_t(x\_t) = \max\_{f \in \mathcal{F}} - (f(x\_t) - f\_t(x\_t)) \text{ s.t. } \sum\_{s=1}^{t-1} Z\_s |f(x\_s) - f\_s(x\_s)|^2 \leq \Psi$ and then picking the maximum absolute value. Both of these new objectives are linear functions, with convex constraints, and thus can be solved   efficiently using a standard solver (e.g. CvXOPT).

- **Differentiable parameterizations**: When class $\mathcal{F}$ could be parameterized in a differentiable way, e.g. a neural network, we can simply add the constraints as a penalty (with the appropriate multiplicative scale parameter) to convert eqn (5) into an unconstrained optimization problem w.r.t the parameters $\theta$ (of the differentiable parameterization) and then solve it using SGD algorithm. While this is only a heuristic, it works well in practice and is precisely what we do for our experiments in Appendix A with 2 layer neural networks.

### Efficient Implementation of eqn (5) via calls to a Regression Oracles w.r.t. $\mathcal{F}$.

 Below we discuss some scenarios and minor modifications of our algorithms under which the computation of $\Delta_t$ (as in eqn (5) in Algorithm 1 or eqn (6) in Algorithm 2) can be performed efficiently via calls to a regression oracle w.r.t.~$\mathcal{F}$. Suppose that $\mathcal{F}$ is closed under convexification. We consider two scenarios:

-  **Binary Actions Setting**: In this case, we can simply choose $\mathcal{F}$ to be a class of 1D functions, since we can model the expert using a function $f^\star: \mathcal{X} \mapsto \mathbb{R} \in \mathcal{F}$ where for any $x$, $f^\star(x)$ denotes the probability of choosing the first action. The probability of choosing the second action would then be $1 - f^\star(x)$. In this case, eqn (5) in Algorithm 1 simply reduces to

$\Delta'\_t(x\_t) = \max\_{f \in \mathcal{F}} |f(x\_t) - f\_t(x\_t)|\text{ s.t. }\sum_{s=1}^{t-1} Z\_s |f(x\_s) - f\_s(x\_s)|^2 \leq \Psi$.

The above can be implemented efficiently using the techniques from [1]. In particular, let $r\_{\max}$ denote the maximum value that $f(x)$ can take. We can solve the above objective using the BINSEARCH procedure in [1] where we perform a binary search over a weight parameter $w$, by solving for each $w$ the optimization problems:

$\arg\min\_{f \in \mathcal{F}} \left(w \cdot (f(x\_t) - f\_t(x\_t) - 2r\_{\max})^2 + \sum\_{s=1}^{t-1} Z\_s |f(x\_s) - f\_s(x\_s)|^2\right)$

and

$\arg\min\_{f \in \mathcal{F}} \left(w \cdot (f(x\_t) - f\_t(x\_t) +  2r\_{\max})^2 + \sum\_{s=1}^{t-1} Z\_s |f(x\_s) - f\_s(x\_s)|^2\right)$

both of which can be efficiently implemented using square loss regression oracles to  $\mathcal{F}$. We refer to [1] for the exact details.

- **Multiple Actions Setting:** Suppose that $\mathcal{F}$ is a product class of $\{\mathcal{F}\_k\}$ for different actions $k \in [K]$. We can get an oracle-efficient algorithm for a slight modification of equation (5), at a price of an extra multiplicative $K$ factor in the query complexity bound. Consider the $\Delta\_t(x\_t)$ given by
$ \max\_{f \in \mathcal{F}} \|f(x\_t) - f\_t(x\_t)\|\_\infty \text{ s.t. }\sum\_{s=1}^{t-1} Z\_s ||f(x\_s) - f\_s(x\_s)||^2\_2 \leq \Psi$, which is equal to
$ \max\_{k \in [K]} \max\_{f \in \mathcal{F}} |f\_k(x\_t) - f\_{t, k}(x\_t)|\text{ s.t. }\sum\_{s=1}^{t-1} Z\_s ||f(x\_s) - f\_s(x\_s)||^2\_2 \leq \Psi$, which can again be implemented via calls to a square loss regression oracle w.r.t.~$\mathcal{F}$ by using BINSEARCH procedure in [1] (similar to what we did for the Binary actions case above).


We will add more details on computational efficiency and practical implementations in the final version of the paper.

[1] "Practical Contextual Bandits with Regression Oracles", Foster et al. 2018

---

### Decision · Program_Chairs · 2023-09-21

**Decision:**

Accept (poster)

**Comment:**

The submitted paper was reviewed by 4 knowledgeable reviewers, all of whom recommended the acceptance of the paper. Initial concerns regarding the tractability of the optimization problems arising as part of the proposed approach and a few unclarities were successfully resolved by the authors in their rebuttal. There is a shared and standing concern regarding limited experimental validation but as the main contribution of the paper is theoretical this does not pose a major obstacle. Thus, in line with the reviewers' recommendations, I am recommending acceptance of the paper. Nevertheless, I would like to ask the authors to take several points into account when preparing the camera ready version of their paper:
* Please include the additional information you provided in the discussion with the reviewers in the paper. There were shared unclarities which should be resolved in the paper.
* If possible, extend the empirical evaluation to give a better sense of how well the proposed algorithm can work and how other algorithms can fail for the considered settings.
* Consider adjusting the paper structure. Currently the paper ends very abruptly with Section 4. Rearranging some material will result in an easier-to-read and digest paper.